# Meta Inverse Constrained Reinforcement Learning: Convergence Guarantee and Generalization Analysis

**Shicheng Liu & Minghui Zhu**
Department of Electrical Engineering
Pennsylvania State University
University Park, PA 16802, USA
{sfl5539,muz16}@psu.edu

## Abstract

This paper considers the problem of learning the reward function and constraints of an expert from few demonstrations. This problem can be considered as a meta-learning problem where we first learn meta-priors over reward functions and constraints from other distinct but related tasks and then adapt the learned meta-priors to new tasks from only few expert demonstrations. We formulate a bi-level optimization problem where the upper level aims to learn a meta-prior over reward functions and the lower level is to learn a meta-prior over constraints. We propose a novel algorithm to solve this problem and formally guarantee that the algorithm reaches the set of $\epsilon$-stationary points at the iteration complexity $O(\frac{1}{\epsilon^2})$. We also quantify the generalization error to an arbitrary new task. Experiments are used to validate that the learned meta-priors can adapt to new tasks with good performance from only few demonstrations.

## 1 Introduction

Inverse reinforcement learning (IRL) has been receiving substantial research efforts due to its effectiveness to recover a reward function from expert's demonstrations that can well explain the expert's behavior. In practical applications, however, constraints are ubiquitous and a reward function combined with a set of constraints can better explain complicated behaviors than a single reward function (Malik et al., 2021). Therefore, inverse constrained reinforcement learning (ICRL) is proposed to learn constraints from expert's demonstrations. Current state-of-the-arts on IRL (Fu et al., 2018; Imani & Ghoreishi, 2021) and ICRL (Scobee & Sastry, 2019) can either learn a reward function in unconstrained environments or infer constraints with access to the ground truth reward but cannot infer both. To solve this challenge, distributed ICRL (Liu & Zhu, 2022) is proposed to learn both the reward function and constraints of the expert. In this paper, we follow the definition of ICRL in (Liu & Zhu, 2022), which means learning both the reward function and constraints of the expert.

While the aforementioned literature can recover the reward function and constraints for single tasks, they typically need large amounts of expert demonstrations (Yu et al., 2019). When it comes to multiple related tasks that share common structural patterns, e.g., navigating to different locations in a common environment (Xu et al., 2019), it could be expensive and inefficient to collect enough demonstrations for each task and then learn the corresponding reward function and constraints separately. Meta-learning (Rajeswaran et al., 2019) has a potential to learn the reward functions and constraints efficiently from few demonstrations. It can exploit the structural similarity of a group of related tasks by learning meta-priors. The learned meta-priors allow for rapid adaptation to new related tasks from only limited data. Therefore, it motivates us to leverage meta-learning to infer the reward functions and constraints of the experts in new tasks from only few demonstrations.

**Related works**. IRL (Abbeel & Ng, 2004; Ziebart et al., 2008; Ziebart, 2010) and ICRL (Scobee & Sastry, 2019; Malik et al., 2021; Liu & Zhu, 2022) have shown great success in recovering the reward function and constraints from expert's demonstrations. However, when it comes to multiple related tasks, they all require large amounts of demonstrations for each task. Meta-learning (Finn

et al., 2017; Rajeswaran et al., 2019; Xu & Zhu, 2023b) provides a way to learn from limited data by learning the common structural patterns (i.e., meta-priors) of the related tasks and then optimizing for rapid adaptation to unseen tasks from only few data. It has achieved state-of-the-art performance in few-shot regression, classification (Finn et al., 2017), and reinforcement learning (Fallah et al., 2021a; Xu & Zhu, 2022). Recently, several meta IRL algorithms are proposed to recover reward functions from few demonstrations. In specific, (Yu et al., 2018; Xu et al., 2019) propose to learn a reward parameter initialization that can be adapted to new tasks via only one or few gradient descent step(s). (Yu et al., 2019; Seyed Ghasemipour et al., 2019) propose to learn a context-conditional model that, given a new task, can encode the task and output the corresponding reward parameters.

However, the existing works on meta IRL have two limitations. (i) They do not explicitly deal with constraints. Existing meta-learning algorithms can directly compute the gradient of the meta objective (i.e., hyper-gradient) when only reward functions are learned (Xu et al., 2019), but cannot compute the hyper-gradient when we also need to deal with constraints. (ii) They do not theoretically guarantee the proposed algorithms' convergence, and more importantly, adaptation performance (i.e., generalization error) to new tasks. This paper proposes the first theoretical framework and thereby an algorithm that can learn the reward function and constraints of a new task from only few demonstrations by first learning meta-priors over reward functions and constraints. While there is no meta IRL theoretical work, there are several theoretical works on other meta-learning problems. We discuss our distinctions from other related meta-learning theoretical works in Appendix A.14.

**Contribution statement**. Our contributions are threefold. First, we extend ICRL (Liu & Zhu, 2022) to a meta-learning setting where we learn meta-priors over reward functions and constraints in order to adapt to new related tasks from few demonstrations. We formulate a novel bi-level optimization problem to solve it. Second, we propose a novel "meta inverse constrained reinforcement learning" (M-ICRL) algorithm, that can efficiently compute the hyper-gradient, to solve the problem. Third, we provide the iteration complexity $O(\frac{1}{\epsilon^2})$ of the algorithm reaching the set of $\epsilon$-stationary points. More importantly, we quantify the generalization error to an arbitrary new task. It is shown that the generalization error can be sufficiently small if the new task is "close" to the training tasks.

## 2 PROBLEM FORMULATION

This section introduces the definition of a single task and then formulates the meta-learning problem.

### 2.1 SINGLE TASK: ICRL

In our problem, a single task $\mathcal{T}_i$ is an ICRL problem (Liu & Zhu, 2022) where a learner aims to learn the reward function and constraints of an expert from the expert's demonstrated trajectories. The expert's decision making is based on a constrained Markov decision process (CMDP). The task $\mathcal{T}_i$'s CMDP $(\mathcal{S}, \mathcal{A}, \gamma, P_0, P, r_i, c_i, b_i)$ is defined via state set $\mathcal{S}$, action set $\mathcal{A}$, discount factor $\gamma$, and initial state distribution $P_0$. The probability of state transition to $s'$ from $s$ by taking action $a$ is $P(s'|s, a)$. The reward and cost functions of the expert are $r_i, c_i : \mathcal{S} \times \mathcal{A} \to \mathbb{R}$. A trajectory of the CMDP is a state-action sequence $\zeta = s_0, a_0, s_1, a_1, \cdots$ and we use $P_\pi$ to denote the trajectory distribution generated by an arbitrary policy $\pi$ where the initial state is drawn from $P_0$. Define $J_{r_i}(\pi) \triangleq E_{\zeta \sim P_\pi}[\sum_{t=0}^{\infty} \gamma^t r_i(s_t, a_t)]$ as the expected cumulative reward under the policy $\pi$ and $J_{c_i}(\pi) \triangleq E_{\zeta \sim P_\pi}[\sum_{t=0}^{\infty} \gamma^t c_i(s_t, a_t)]$ as the expected cumulative cost. The expert's policy $\pi_i$ wants to maximize $J_{r_i}(\pi)$ subject to $J_{c_i}(\pi) \leq b_i$ where $b_i$ is a pre-defined budget. The expert can roll out $\pi_i$ to demonstrate a set of $D_i$ trajectories $\mathcal{D}_i = \{\zeta^j\}_{j=1}^{D_i}$ where $\zeta^j = s_0^j, a_0^j, s_1^j, a_1^j, \cdots$.

A learner observes $\mathcal{D}_i$ and aims to use parameterized models $r_\theta$ and $c_\omega$ with parameters $\theta$ and $\omega$ to learn the expert's reward function $r_i$ and cost function $c_i$ by solving the following ICRL problem:

$$\min_\theta \quad L_i(\theta, \omega^*(\theta)), \quad \text{s.t. } \omega^*(\theta) = \arg\min_\omega G_i(\omega; \theta). \tag{1}$$

The upper-level problem aims to learn a reward function $r_\theta$ that can minimize the expected negative log-likelihood $L_i(\theta, \omega) \triangleq -E_{\zeta \sim P_{\pi_i}}[\sum_{t=0}^{\infty} \gamma^t \log \pi_{\omega;\theta}(a_t|s_t)]$ where $\pi_{\omega;\theta}$ is the constrained soft Bellman policy (see the expression in Appendix A.2) (Liu & Zhu, 2022; 2024) under the reward function $r_\theta$ and cost function $c_\omega$. The constrained soft Bellman policy is an extension of soft Bellman policy (Ziebart et al., 2010; Zhou et al., 2017) to CMDPs. The soft Bellman policy is widely used in soft Q-learning (Haarnoja et al., 2017) and soft actor-critic (Haarnoja et al., 2018).

The lower-level **function** $G_i(\omega; \theta) \triangleq \max_\pi H(\pi) + J_{r_\theta}(\pi) - J_{c_\omega}(\pi) + J_{c_\omega}(\pi_i)$ can be regarded as an RL problem which aims to find the policy that maximizes the entropy-regularized cumulative reward-minus-cost (i.e., $H(\pi) + J_{r_\theta}(\pi) - J_{c_\omega}(\pi)$) where $H(\pi) \triangleq E_{\zeta \sim P_\pi}[-\sum_{t=0}^\infty \gamma^t \log \pi(a_t|s_t)]$ is the causal entropy. Note that the likelihood $L_i$ is defined on the expert's trajectory distribution $P_{\pi_i}$ while $H(\pi)$ is defined on the trajectory distribution of its current policy $\pi$. The last term $J_{c_\omega}(\pi_i)$ in $G_i$ is constant w.r.t. $\pi$. It is proved (Liu & Zhu, 2022) that the constrained soft Bellman policy is the optimal policy of the RL problem in $G_i(\omega; \theta)$, i.e., $\pi_{\omega;\theta} = \arg\max_\pi H(\pi) + J_{r_\theta}(\pi) - J_{c_\omega}(\pi)$. The lower-level **problem** $\min_\omega G_i(\omega; \theta)$ uses adversarial learning to find a cost function $c_\omega$ that makes the best policy (i.e., $\pi_{\omega;\theta}$) perform the worst and the last term $J_{c_\omega}(\pi_i)$ penalizes cost functions where the expert has high cumulative cost. We discuss the formulation of (1) in detail in Appendix A.1.

Since problem (1) is defined in expectation but the learner only observes $\mathcal{D}_i$, in practice, the learner solves an empirical problem defined on $\mathcal{D}_i$. Given a trajectory $\zeta^j = s_0^j, a_0^j, \cdots$, we define $\hat{J}_c(\zeta^j) \triangleq \sum_{t=0}^\infty \gamma^t c(s_t^j, a_t^j)$ as the empirical cumulative cost. Then the empirical problem the learner solves is $\min_\theta \hat{L}_i(\theta, \hat{\omega}^*(\theta), \mathcal{D}_i)$, s.t. $\hat{\omega}^*(\theta) = \arg\min_\omega \hat{G}_i(\omega; \theta, \mathcal{D}_i)$ where the $\hat{L}_i(\theta, \omega, \mathcal{D}_i) \triangleq -\frac{1}{D_i} \sum_{j=1}^{D_i} \sum_{t=0}^\infty \gamma^t \log \pi_{\omega;\theta}(a_t^j|s_t^j)$ and $\hat{G}_i(\omega; \theta, \mathcal{D}_i) \triangleq \max_\pi H(\pi) + J_{r_\theta}(\pi) - J_{c_\omega}(\pi) + \frac{1}{D_i} \sum_{j=1}^{D_i} \hat{J}_{c_\omega}(\zeta^j)$.

## 2.2 MULTIPLE TASKS: M-ICRL

ICRL in (1) can successfully recover the reward and cost functions of the expert (Liu & Zhu, 2022; 2024). However, it typically needs a large data set for each task when it comes to multiple related tasks. To learn the reward and cost functions from few demonstrations, we leverage meta-learning which optimizes for the ability to learn efficiently on new tasks. It is typically assumed in meta-learning that there is a set of $m$ training tasks $\{\mathcal{T}_i\}_{i=1}^m$ which share the CMDP. The difference of tasks is that each task $\mathcal{T}_i$ has its own reward function $r_i$, cost function $c_i$, and budget $b_i$. The goal of meta-learning is to optimize for meta-priors of reward and cost functions over the $m$ training tasks $\{\mathcal{T}_i\}_{i=1}^m$ such that the reward and cost functions adapted from the learned meta-priors have good performance on new tasks even if the new tasks only have limited data.

When it comes to meta-learning, two of the state-of-the-arts are model agnostic meta-learning (MAML) (Finn et al., 2017) and meta-learning with implicit gradients (iMAML) (Rajeswaran et al., 2019). MAML is simple and widely implemented in RL (Fallah et al., 2021a) and IRL (Yu et al., 2019), while iMAML shows better empirical performance (Rajeswaran et al., 2019) at the expense of heavier computation in the lower level since MAML only needs one gradient descent but iMAML needs to fully solve an optimization problem in the lower level. In M-ICRL, we aim to propose a problem formulation that utilizes the advantages of both methods.

The proposed problem formulation (2)-(3) has a bi-level structure (Ji et al., 2021; Xu & Zhu, 2023a) where we learn the reward meta-prior in the upper level and the cost meta-prior in the lower level.

$$\min_{\theta, \omega} \quad \frac{1}{m} \sum_{i=1}^m L_i(\varphi_i, \eta_i^*(\varphi_i, \omega)), \tag{2}$$

$$\text{s.t.} \quad \eta_i^*(\varphi_i, \omega) = \arg\min_\eta G_i(\eta; \varphi_i) + \frac{\lambda}{2} ||\eta - \omega||^2, \tag{3}$$

where $\varphi_i \triangleq \theta - \alpha \frac{\partial}{\partial \theta} L_i(\theta, \eta_i^*(\theta, \omega))$ is the task-specific reward adaptation and $\eta_i^*(\varphi_i, \omega)$ is the task-specific cost adaptation. Note that problem (2)-(3) reduces to the ICRL problem (1) if we only consider one task and do not perform meta-learning on reward parameter $\theta$ nor cost parameter $\omega$, i.e., $m = 1$, $\alpha = 0$, and $\lambda = 0$. In this case, we do not have task-specific adaptations $(\varphi_i, \eta_i^*)$. Problem (2)-(3) reduces to MAML if we only do meta-learning on the reward parameter $\theta$ and do not perform meta-learning on the cost parameter $\omega$, i.e., $\lambda = 0$. In this case, we only have the task-specific reward adaptation $\varphi_i$. Problem (2)-(3) reduces to iMAML (explained in Appendix A.4) if we only do meta-learning on the cost parameter $\omega$ and do not perform meta-learning on the reward parameter $\theta$, i.e., $\alpha = 0$. In this case, we only have the task-specific cost adaptation $\eta_i^*$.

Problem (2)-(3) can reduce to the MAML that only learns $\theta$ and the iMAML that only learns $\omega$. It utilizes iMAML but does not suffer from the extra computation burden usually caused by iMAML because ICRL in (1) is already a bi-level formulation and we need to fully solve the lower-level problem anyway. We do not use iMAML for $\theta$ because this will lead to a "three-level" problem.

## 3 THE PROPOSED ALGORITHM

This section proposes a novel algorithm that solves problem (2)-(3). Following (Fallah et al., 2020), we partition the data set $\mathcal{D}_i$ of each training task $\mathcal{T}_i$ into three subsets $\mathcal{D}_i^{\text{tr}}$, $\mathcal{D}_i^{\text{eval}}$, and $\mathcal{D}_i^{\text{h}}$ with sizes $D_i^{\text{tr}}$, $D_i^{\text{eval}}$, and $D_i^{\text{h}}$ respectively. The training set $\mathcal{D}_i^{\text{tr}}$ with limited data is used to compute the task-specific adaptations $\varphi_i$ and $\eta_i^*$, the evaluation set $\mathcal{D}_i^{\text{eval}}$ with abundant data is used to compute the hyper-gradients (i.e., the gradients of the upper-level loss function in (2) with respect to $\theta$ and $\omega$), and the set $\mathcal{D}_i^{\text{h}}$ is used to compute the second-order terms in the hyper-gradients.

For an arbitrary data set $\mathcal{D}$ with size $D$, we solve the empirical version (i.e., $\arg\min_\eta \hat{G}_i(\eta; \varphi_i, \mathcal{D}) + \frac{\lambda}{2}||\eta - \omega||^2$) of the lower-level problem (3) using $(K-1)$-step gradient descent $\hat{\eta}_i(\varphi_i, \omega, \mathcal{D}, k+1) = \hat{\eta}_i(\varphi_i, \omega, \mathcal{D}, k) - \tau[\nabla_\eta \hat{G}_i(\hat{\eta}_i(\varphi_i, \omega, \mathcal{D}, k); \varphi_i, \mathcal{D}) + \lambda(\hat{\eta}_i(\varphi_i, \omega, \mathcal{D}, k) - \omega)]$ where $\tau$ is the step size. We then use $\hat{\eta}_i(\varphi_i, \omega, \mathcal{D}, K)$ as an approximation of $\hat{\eta}_i^*(\varphi_i, \omega, \mathcal{D}) \triangleq \arg\min_\eta \hat{G}_i(\eta; \varphi_i, \mathcal{D}) + \frac{\lambda}{2}||\eta - \omega||^2$. We provide the expressions of all the gradients, including $\nabla_\eta \hat{G}_i$, in Appendix A.3.

---

**Algorithm 1** Meta inverse constrained reinforcement learning (M-ICRL)

---

**Input**: Initialized reward meta-prior $\theta(0)$ and cost meta-prior $\omega(0)$, task batch size $B$, step size $\alpha$
**Output**: Learned meta-prior $\theta(n)$ and cost meta-prior $\omega(n)$
1:  **for** $n = 0, 1, \cdots$ **do**
2:      Samples a batch of training tasks $\{\mathcal{T}_i\}_{i=1}^B$ with size $B$
3:      **for** all $\mathcal{T}_i$ **do**
4:          Samples the demonstration set $\mathcal{D}_i^{\text{tr}}$ to compute $\hat{\eta}_i(\theta(n), \omega(n), \mathcal{D}_i^{\text{tr}}, K)$ and $\hat{\varphi}_i(n) = \theta(n) - \alpha \frac{\partial}{\partial \theta} \hat{L}_i(\theta(n), \hat{\eta}_i(\theta(n), \omega(n), \mathcal{D}_i^{\text{tr}}, K), \mathcal{D}_i^{\text{tr}})$
5:          Samples the demonstration sets $\mathcal{D}_i^{\text{eval}}$ and $\mathcal{D}_i^{\text{h}}$
6:          $\nabla_{\theta,i}, \nabla_{\omega,i} = $ Hyper-gradient$(\theta(n), \omega(n), \hat{\varphi}_i(n), \mathcal{D}_i^{\text{tr}}, \mathcal{D}_i^{\text{eval}}, \mathcal{D}_i^{\text{h}})$
7:      **end for**
8:      $\theta(n+1) = \theta(n) - \frac{\alpha(n)}{B} \sum_{i=1}^B \nabla_{\theta,i}, \qquad \omega(n+1) = \omega(n) - \frac{\alpha(n)}{B} \sum_{i=1}^B \nabla_{\omega,i}$
9:  **end for**

---

At each iteration $n$ in Algorithm 1, the learner samples $B$ tasks from the set of training tasks $\{\mathcal{T}_i\}_{i=1}^m$. For each sampled training task $\mathcal{T}_i$, the learner first uses the training set $\mathcal{D}_i^{\text{tr}}$ to compute $\hat{\eta}_i$ and the task-specific reward adaptation $\hat{\varphi}_i$ (line 4). Then the learner uses the training set $\mathcal{D}_i^{\text{tr}}$, evaluation set $\mathcal{D}_i^{\text{eval}}$, and $\mathcal{D}_i^{\text{h}}$ to compute the hyper-gradients $\nabla_{\theta,i}$ and $\nabla_{\omega,i}$ (line 6). Finally, the learners utilizes stochastic gradient descent to update the reward and cost meta-priors (line 8).

The computation of the hyper-gradients is critical to Algorithm 1. In the following context, we first identify the difficulties of computing the hyper-gradients and then provide our solutions.

### 3.1 CHALLENGES OF COMPUTING THE HYPER-GRADIENTS

The hyper-gradients $\frac{\partial L_i(\varphi_i, \eta_i^*(\varphi_i, \omega))}{\partial \theta}$ and $\frac{\partial L_i(\varphi_i, \eta_i^*(\varphi_i, \omega))}{\partial \omega}$ of problem (2)-(3) are hard to compute. Take $\frac{\partial L_i(\varphi_i, \eta_i^*(\varphi_i, \omega))}{\partial \theta}$ as an example (the derivation of the hyper-gradients is in Appendix A.5):

$$\frac{\partial L_i(\varphi_i, \eta_i^*(\varphi_i, \omega))}{\partial \theta} = \left[ I - \alpha \frac{\partial^2}{\partial \theta^2} L_i(\theta, \eta_i^*(\theta, \omega)) \right] \cdot \left[ \nabla_{\varphi_i} L_i(\varphi_i, \eta_i^*(\varphi_i, \omega)) \right.$$

$$\left. - \nabla_{\varphi_i \eta}^2 G_i(\eta_i^*(\varphi_i, \omega); \varphi_i)[\nabla_{\eta\eta}^2 G_i(\eta_i^*(\varphi_i, \omega); \varphi_i) + \lambda I]^{-1} \nabla_\eta L_i(\varphi_i, \eta_i^*(\varphi_i, \omega)) \right].$$

(i) The second-order term $\frac{\partial^2}{\partial \theta^2} L_i(\theta, \eta_i^*(\theta, \omega))$ in the first bracket is intractable to compute since it requires to compute $\nabla_{\theta\theta}^2 \eta_i^*(\theta, \omega)$ which needs to calculate the gradient of an inverse-of-Hessian term $[\nabla_{\eta\eta}^2 G_i(\eta_i^*(\varphi_i, \omega); \varphi_i) + \lambda I]^{-1}$.
(ii) The inverse-of-Hessian $[\nabla_{\eta\eta}^2 G_i(\eta_i^*(\varphi_i, \omega); \varphi_i) + \lambda I]^{-1}$ in the second bracket is expensive to compute, especially when we use neural networks as parameterized models.
(iii) We cannot get $\eta_i^*$ but only its approximation since the optimization oracle is not guaranteed to find the exact optimal solution. This will cause errors when we compute the hyper-gradients.

### 3.2 MAIN IDEA TO SOLVE THE CHALLENGES

**Solution to challenge (i)** (Algorithm 2). The learner uses sampled data sets $\mathcal{D}_i^{\text{tr}}$, $\mathcal{D}_i^{\text{eval}}$, and $\mathcal{D}_i^{\text{h}}$ to approximate the hyper-gradients:

$$g_{\theta,i} \triangleq \left[ I - \alpha \frac{\partial^2}{\partial \theta^2} \hat{L}_i(\theta, \hat{\eta}_i^*(\theta, \omega, \mathcal{D}_i^{\text{tr}}), \mathcal{D}_i^{\text{h}}) \right] \Delta_{\theta,i}, \tag{4}$$

$$g_{\omega,i} \triangleq -\alpha \frac{\partial^2}{\partial \omega \partial \theta} \hat{L}_i(\theta, \hat{\eta}_i^*(\theta, \omega, \mathcal{D}_i^{\text{tr}}), \mathcal{D}_i^{\text{h}}) \Delta_{\theta,i} + \Delta_{\omega,i}, \tag{5}$$

where $\Delta_{\theta,i}$ and $\Delta_{\omega,i}$ are partial gradients of $\hat{L}_i(\hat{\varphi}_i, \hat{\eta}_i^*(\hat{\varphi}_i, \omega, \mathcal{D}_i^{\text{eval}}), \mathcal{D}_i^{\text{eval}})$ with respect to $\varphi_i$ and $\omega$. While the second-order terms (i.e., $\frac{\partial^2}{\partial \theta^2} \hat{L}_i$ and $\frac{\partial^2}{\partial \omega \partial \theta} \hat{L}_i$) in the hyper-gradients (4)-(5) are directly computed in many meta-learning works (Finn et al., 2017; Xu et al., 2019), in our case, it is prohibitively hard to compute. To compute the second-order terms, we need to calculate $\nabla^2 \hat{\eta}_i^*(\theta, \omega, \mathcal{D}_i^{\text{tr}})$ which needs to calculate the gradient of an inverse-of-Hessian term since $\nabla_\theta \hat{\eta}_i^*(\theta, \omega, \mathcal{D}_i^{\text{tr}}) = -\nabla_{\theta\eta}^2 \hat{G}_i(\hat{\eta}_i^*(\theta, \omega, \mathcal{D}_i^{\text{tr}}); \theta, \mathcal{D}_i^{\text{tr}})[\nabla_{\eta\eta}^2 \hat{G}_i(\hat{\eta}_i^*(\theta, \omega, \mathcal{D}_i^{\text{tr}}); \theta, \mathcal{D}_i^{\text{tr}}) + \lambda I]^{-1}$ and $\nabla_\omega \hat{\eta}_i^*(\theta, \omega, \mathcal{D}_i^{\text{tr}}) = \lambda[\nabla_{\eta\eta}^2 \hat{G}_i(\hat{\eta}_i^*(\theta, \omega, \mathcal{D}_i^{\text{tr}}); \theta, \mathcal{D}_i^{\text{tr}}) + \lambda I]^{-1}$ (derived in Appendix A.5). To tackle this challenge, we use the first-order approximation to approximate the products:

$$\frac{\partial^2}{\partial \theta^2} \hat{L}_i(\theta, \hat{\eta}_i^*(\theta, \omega, \mathcal{D}_i^{\text{tr}}), \mathcal{D}_i^{\text{h}}) \Delta_{\theta,i} \approx \frac{1}{2\delta} \left[ \frac{\partial}{\partial \theta} \hat{L}_i(\theta + \delta \Delta_{\theta,i}, \hat{\eta}_i^*(\theta + \delta \Delta_{\theta,i}, \omega, \mathcal{D}_i^{\text{tr}}), \mathcal{D}_i^{\text{h}}) \right.$$
$$\left. - \frac{\partial}{\partial \theta} \hat{L}_i(\theta - \delta \Delta_{\theta,i}, \hat{\eta}_i^*(\theta - \delta \Delta_{\theta,i}, \omega, \mathcal{D}_i^{\text{tr}}), \mathcal{D}_i^{\text{h}}) \right], \tag{6}$$

$$\frac{\partial^2}{\partial \omega \partial \theta} \hat{L}_i(\theta, \hat{\eta}_i^*(\theta, \omega, \mathcal{D}_i^{\text{tr}}), \mathcal{D}_i^{\text{h}}) \Delta_{\theta,i} \approx \frac{1}{2\delta} \left[ \frac{\partial}{\partial \omega} \hat{L}_i(\theta + \delta \Delta_{\theta,i}, \hat{\eta}_i^*(\theta + \delta \Delta_{\theta,i}, \omega, \mathcal{D}_i^{\text{tr}}), \mathcal{D}_i^{\text{h}}) \right.$$
$$\left. - \frac{\partial}{\partial \omega} \hat{L}_i(\theta - \delta \Delta_{\theta,i}, \hat{\eta}_i^*(\theta - \delta \Delta_{\theta,i}, \omega, \mathcal{D}_i^{\text{tr}}), \mathcal{D}_i^{\text{h}}) \right], \tag{7}$$

where $\delta$ is perturbation magnitude. In Algorithm 2, the learner first approximates the partial gradients $\Delta_{\theta,i}$ and $\Delta_{\omega,i}$ (line 1 in Algorithm 2), and then computes the first-order approximation (lines 2-4 in Algorithm 2). The output of Algorithm 2 is the approximation of the hyper-gradients (4)-(5).

**Solution to challenge (ii)** (Algorithm 3). The partial gradients of $\hat{L}_i(\hat{\varphi}_i, \hat{\eta}_i^*(\hat{\varphi}_i, \omega, \mathcal{D}_i^{\text{eval}}), \mathcal{D}_i^{\text{eval}})$ with respect to $\varphi_i$ and $\omega$ are respectively:

$$\Delta_{\theta,i} = \nabla_{\varphi_i} \hat{L}_i(\hat{\varphi}_i, \hat{\eta}_i^*(\hat{\varphi}_i, \omega, \mathcal{D}_i^{\text{eval}}), \mathcal{D}_i^{\text{eval}}) - \nabla_{\varphi_i \eta}^2 \hat{G}_i(\hat{\eta}_i^*(\hat{\varphi}_i, \omega, \mathcal{D}_i^{\text{eval}}); \hat{\varphi}_i, \mathcal{D}_i^{\text{eval}}) \cdot$$
$$[\lambda I + \nabla_{\eta\eta}^2 \hat{G}_i(\hat{\eta}_i^*(\hat{\varphi}_i, \omega, \mathcal{D}_i^{\text{eval}}); \hat{\varphi}_i, \mathcal{D}_i^{\text{eval}})]^{-1} \nabla_\eta \hat{L}_i(\hat{\varphi}_i, \hat{\eta}_i^*(\hat{\varphi}_i, \omega, \mathcal{D}_i^{\text{eval}}), \mathcal{D}_i^{\text{eval}}), \tag{8}$$
$$\Delta_{\omega,i} = \lambda[\lambda I + \nabla_{\eta\eta}^2 \hat{G}_i(\hat{\eta}_i^*(\hat{\varphi}_i, \omega, \mathcal{D}_i^{\text{eval}}); \hat{\varphi}_i, \mathcal{D}_i^{\text{eval}})]^{-1} \nabla_\eta \hat{L}_i(\hat{\varphi}_i, \hat{\eta}_i^*(\hat{\varphi}_i, \omega, \mathcal{D}_i^{\text{eval}}), \mathcal{D}_i^{\text{eval}}). \tag{9}$$

Note that the partial gradients (8)-(9) contain $[\lambda I + \nabla_{\eta\eta}^2 \hat{G}_i(\hat{\eta}_i^*(\hat{\varphi}_i, \omega, \mathcal{D}_i^{\text{eval}}); \hat{\varphi}_i, \mathcal{D}_i^{\text{eval}})]^{-1} \nabla_\eta \hat{L}_i(\hat{\varphi}_i, \hat{\eta}_i^*(\hat{\varphi}_i, \omega, \mathcal{D}_i^{\text{eval}}), \mathcal{D}_i^{\text{eval}})$ where the inverse-of-Hessian term is expensive to compute. Therefore, we solve the following optimization problem instead:

$$\min_x x^\top \left[ \lambda I + \nabla_{\eta\eta}^2 \hat{G}_i(\hat{\eta}_i^*(\hat{\varphi}_i, \omega, \mathcal{D}_i^{\text{eval}}); \hat{\varphi}_i, \mathcal{D}_i^{\text{eval}}) \right] x - [\nabla_\eta \hat{L}_i(\hat{\varphi}_i, \hat{\eta}_i^*(\hat{\varphi}_i, \omega, \mathcal{D}_i^{\text{eval}}), \mathcal{D}_i^{\text{eval}})]^\top x. \tag{10}$$

It is obvious that the optimal solution of problem (10) is $[\lambda I + \nabla_{\eta\eta}^2 \hat{G}_i(\hat{\eta}_i^*(\hat{\varphi}_i, \omega, \mathcal{D}_i^{\text{eval}}); \hat{\varphi}_i, \mathcal{D}_i^{\text{eval}})]^{-1} \cdot \nabla_\eta \hat{L}_i(\hat{\varphi}_i, \hat{\eta}_i^*(\hat{\varphi}_i, \omega, \mathcal{D}_i^{\text{eval}}), \mathcal{D}_i^{\text{eval}})$.

In Algorithm 3, the learner solves the problem (10) for $(\bar{K} - 1)$-step gradient descent to get an approximation $x(\bar{K})$ of the optimal solution of problem (10) (line 3 in Algorithm 3) and then use $x(\bar{K})$ to help approximate the partial gradients (8)-(9) (lines 5-6 in Algorithm 3).

**Solution to challenge (iii)**. We cannot get $\hat{\eta}_i^*(\theta, \omega, \mathcal{D}_i^{\text{tr}})$ but an approximation $\hat{\eta}_i(\theta, \omega, \mathcal{D}_i^{\text{tr}}, K)$. In practice, we use this approximation to substitute for $\hat{\eta}_i^*(\theta, \omega, \mathcal{D}_i^{\text{tr}})$ in (4)-(5). Similarly, we use the approximation $\hat{\eta}_i(\hat{\varphi}_i, \omega, \mathcal{D}_i^{\text{eval}}, K)$ to substitute for $\hat{\eta}_i^*(\hat{\varphi}_i, \omega, \mathcal{D}_i^{\text{eval}})$ in (8)-(9). To quantify the

approximation error of the hyper-gradients (4)-(5) caused by $||\hat{\eta}_i(\cdot, \cdot, \cdot, K) - \hat{\eta}_i^*(\cdot, \cdot, \cdot)||$, we exploit the Lipschitz continuity of the hyper-gradients with respect to $\eta$. In specific, we first prove the Lipschitz continuity of the partial gradients (8)-(9) w.r.t. $\eta$ in Appendix A.7 and then prove the Lipschitz continuity of the first-order approximation (6)-(7) w.r.t. $\eta$ in Appendix A.8. Then, we can see the Lipschitz continuity of the hyper-gradients (4)-(5) w.r.t. $\eta$.

---

**Algorithm 2** Hyper-gradient$(\theta, \omega, \hat{\varphi}_i, \mathcal{D}_i^{\text{tr}}, \mathcal{D}_i^{\text{eval}}, \mathcal{D}_i^{\text{h}})$

---

**Input**: Reward parameter $\theta$, cost parameter $\omega$, task-specific reward adaptation $\hat{\varphi}_i$, training set $\mathcal{D}_i^{\text{tr}}$, evaluation set $\mathcal{D}_i^{\text{eval}}$, the data set $\mathcal{D}_i^{\text{h}}$ to compute the second-order terms, perturbation $\delta$
**Output**: Approximate hyper-gradients $\hat{\Delta}_{\theta,i} - \alpha \nabla_{\theta,i}$, $\hat{\Delta}_{\omega,i} - \alpha \nabla_{\omega,i}$

1: $\hat{\Delta}_{\theta,i}, \hat{\Delta}_{\omega,i} = \texttt{Partial-gradient}(\hat{\varphi}_i, \omega, \mathcal{D}_i^{\text{eval}}, \mathcal{D}_i^{\text{eval}})$
2: $\hat{\Delta}_{\theta+,i}, \hat{\Delta}_{\omega+,i} = \texttt{Partial-gradient}(\theta + \delta \Delta_{\theta,i}, \omega, \mathcal{D}_i^{\text{tr}}, \mathcal{D}_i^{\text{h}})$
3: $\hat{\Delta}_{\theta-,i}, \hat{\Delta}_{\omega-,i} = \texttt{Partial-gradient}(\theta - \delta \Delta_{\theta,i}, \omega, \mathcal{D}_i^{\text{tr}}, \mathcal{D}_i^{\text{h}})$
4: $\nabla_{\theta,i} = (\hat{\Delta}_{\theta+,i} - \hat{\Delta}_{\theta-,i})/(2\delta), \quad \nabla_{\omega,i} = (\hat{\Delta}_{\omega+,i} - \hat{\Delta}_{\omega-,i})/(2\delta)$

---

**Algorithm 3** Partial-gradient$(\theta, \omega, \mathcal{D}_1, \mathcal{D}_2)$

---

**Input**: Reward parameter $\theta$, cost parameter $\omega$, data set $\mathcal{D}_1$, data set $\mathcal{D}_2$, step size $\beta$
**Output**: Approximate partial gradients $\hat{\Delta}_{\theta,i}, \hat{\Delta}_{\omega,i}$

1: Compute $\hat{\eta}_i(\theta, \omega, \mathcal{D}_1, K)$ and initialize $x(0)$
2: **for** $\bar{k} = 0, 1, \cdots, \bar{K} - 1$ **do**
3: $\quad x(\bar{k} + 1)$
$$= x(\bar{k}) - \beta \left( [\lambda I + \nabla_{\eta\eta}^2 \hat{G}_i(\hat{\eta}_i(\theta, \omega, \mathcal{D}_1, K); \theta, \mathcal{D}_1)] x(\bar{k}) - \nabla_\eta \hat{L}_i(\theta, \hat{\eta}_i(\theta, \omega, \mathcal{D}_1, K), \mathcal{D}_2) \right)$$
4: **end for**
5: $\hat{\Delta}_{\theta,i} = \nabla_\theta \hat{L}_i(\theta, \hat{\eta}_i(\theta, \omega, \mathcal{D}_1, K), \mathcal{D}_2) - \nabla_{\theta\eta}^2 \hat{G}_i(\hat{\eta}_i(\theta, \omega, \mathcal{D}_1, K); \theta, \mathcal{D}_1) x(\bar{K})$
6: $\hat{\Delta}_{\omega,i} = \lambda x(\bar{K})$

---

## 4 THEORETICAL ANALYSIS

This section has two parts: the first part provides the convergence guarantee of Algorithm 1 and the second part quantifies the generalization error to an arbitrary new task.

### 4.1 CONVERGENCE GUARANTEE

Compared to the standard stochastic gradient descent, the main difficulty of guaranteeing the convergence of Algorithm 1 lies in quantifying the approximation error of the hyper-gradients. The approximation error comes from three aspects which correspond to the three challenges in Subsection 3.1. (i) We cannot obtain the exact optimal solution $\hat{\eta}_i^*(\cdot, \cdot, \cdot)$ of the lower-level problem (3) but an approximation $\hat{\eta}_i(\cdot, \cdot, \cdot, K)$. (ii) We cannot compute the inverse-of-Hessian term $[\lambda I + \nabla_{\eta\eta}^2 \hat{G}_i]^{-1}$ but use an iterative method to approximate the product $[\lambda I + \nabla_{\eta\eta}^2 \hat{G}_i]^{-1} \nabla_\eta \hat{L}_i$ in Algorithm 3. This will result in the error between the approximate partial gradients $\hat{\Delta}_{\theta,i}, \hat{\Delta}_{\omega,i}$ (i.e., the output of Algorithm 3) and the real partial gradients $\Delta_{\theta,i}, \Delta_{\omega,i}$ in (8)-(9). (iii) We use the first-order approximation (6)-(7) in Algorithm 2 to approximate the real hyper-gradients (4)-(5).

In what follows, we first sequentially quantify the three approximation errors identified in the last paragraph and then analyze the convergence of Algorithm 1. We start with the following assumption.

**Assumption 1.** *(i) The parameterized reward function $r_\theta$ satisfies $|r_\theta(s, a)| \leq C_r$, $||\nabla_\theta r_\theta(s, a)|| \leq \bar{C}_r$, and $||\nabla_{\theta\theta}^2 r_\theta(s, a)|| \leq \tilde{C}_r$ for any $(s, a) \in \mathcal{S} \times \mathcal{A}$ and any $\theta$ where $C_r$, $\bar{C}_r$, and $\tilde{C}_r$ are positive constants; (ii) The parameterized cost function $c_\omega$ has similar properties with positive constants $C_c$, $\bar{C}_c$, and $\tilde{C}_c$; (iii) The third and fourth order gradients of the reward and cost functions with respect to their parameters are bounded for any $(s, a)$ and $(\theta, \omega)$.*

Note that Assumptions 1 (i) and (ii) are standard in RL (Wang et al., 2019; Kumar et al., 2019; Zhang et al., 2020; Zheng et al., 2023) and IRL (Guan et al., 2021). Assumption 1 (iii) is needed to exploit the Lipschitz continuity of the hyper-gradients. Moreover, the bounded third order gradients of the parameterized model in Assumption 1 (iii) are commonly assumed in meta RL (Fallah et al., 2021a).

**Approximation error (i)**. Proved in Appendix A.6, the function $G_i$ and its empirical approximation $\hat{G}_i$ using any data set are $C_{\nabla^2_{\eta\eta}G}$-smooth for any task $\mathcal{T}_i$ where $C_{\nabla^2_{\eta\eta}G}$ is a positive constant whose expression is in Appendix A.6. Therefore, the lower-level objective function in (3) becomes $(\lambda - C_{\nabla^2_{\eta\eta}G})$-strongly convex and $(\lambda + C_{\nabla^2_{\eta\eta}G})$-smooth if $\lambda > C_{\nabla^2_{\eta\eta}G}$. Choosing $\tau = \frac{1}{\lambda}$ and following the standard result for strongly-convex and smooth objective functions (Nesterov, 2003; Boyd & Vandenberghe, 2004), we have $||\hat{\eta}_i(\cdot,\cdot,\cdot,K) - \hat{\eta}_i^*(\cdot,\cdot,\cdot)|| \leq O((C_{\nabla^2_{\eta\eta}G}/\lambda)^K)$.

**Approximation error (ii)**. We next quantify the approximation error of the partial-gradients.

**Lemma 1.** *Suppose Assumption 1 holds and let $\beta = \frac{1}{\lambda}$ where $\lambda > C_{\nabla^2_{\eta\eta}G}$, then the outputs of Algorithm 3 satisfy:*

$$||\hat{\Delta}_{\theta,i} - \Delta_{\theta,i}|| \leq O\left(\left(\frac{C_{\nabla^2_{\eta\eta}G}}{\lambda}\right)^K + \left(\frac{C_{\nabla^2_{\eta\eta}G}}{\lambda}\right)^{\bar{K}}\right),$$

$$||\hat{\Delta}_{\omega,i} - \Delta_{\omega,i}|| \leq O\left(\left(\frac{C_{\nabla^2_{\eta\eta}G}}{\lambda}\right)^K + \left(\frac{C_{\nabla^2_{\eta\eta}G}}{\lambda}\right)^{\bar{K}}\right).$$

Lemma 1 shows that the approximation error of the partial gradients diminishes if we increase the iteration numbers of solving the lower-level problem and in Algorithm 3.

**Approximation error (iii)**. With the approximation error of the partial gradients, we can quantify the approximation error of the hyper-gradients.

**Lemma 2.** *Suppose the conditions in Lemma 1 hold, then the outputs of Algorithm 2 satisfy:*

$$||\hat{\Delta}_{\theta,i} - \alpha\nabla_{\theta,i} - g_{\theta,i}|| \leq O\left(\left(\frac{C_{\nabla^2_{\eta\eta}G}}{\lambda}\right)^K + \left(\frac{C_{\nabla^2_{\eta\eta}G}}{\lambda}\right)^{\bar{K}} + \delta\right),$$

$$||\hat{\Delta}_{\omega,i} - \alpha\nabla_{\omega,i} - g_{\omega,i}|| \leq O\left(\left(\frac{C_{\nabla^2_{\eta\eta}G}}{\lambda}\right)^K + \left(\frac{C_{\nabla^2_{\eta\eta}G}}{\lambda}\right)^{\bar{K}} + \delta\right).$$

Lemma 2 indicates that the approximation error of the hyper-gradients can be arbitrarily small if we solve the lower-level problem (3) for enough iterations, run Algorithm 3 for enough iterations, and choose sufficiently small $\delta$.

To reason about the convergence of Algorithm 1, we introduce $\epsilon$-approximate first order stationary point ($\epsilon$-FOSP) (Fallah et al., 2020): the variable $(\theta, \omega)$ is $\epsilon$-FOSP if $||\frac{1}{m}\sum_{i=1}^m \nabla L_i(\varphi_i, \eta_i^*(\varphi_i, \omega))||$ $\leq \epsilon$ where $\nabla L_i(\varphi_i, \eta_i^*(\varphi_i, \omega)) \triangleq [(\frac{\partial}{\partial\omega}L_i(\varphi_i, \eta_i^*(\varphi_i, \omega)))^\top, (\frac{\partial}{\partial\theta}L_i(\varphi_i, \eta_i^*(\varphi_i, \omega)))^\top]^\top$.

**Theorem 1** (Convergence of Algorithm 1). *Suppose the conditions in Lemma 2 hold. Let $\alpha \in [0, \frac{1}{\tilde{D}_\theta}]$ and $\alpha(n) = \frac{\bar{\alpha}}{(n+1)^\rho}$ where $\bar{\alpha} \in (0, \frac{2}{C_f+2}], \rho \in (\frac{1}{2}, 1)$, and $\tilde{D}_\theta$ and $C_f$ are positive constants whose existence is proved in Appendices A.8 and A.9 respectively. Then Algorithm 1 reaches the set of $\epsilon$-FOSP, i.e.,*

$$E\left[\left|\left|\frac{1}{m}\sum_{i=1}^m \nabla L_i(\varphi_i(n), \eta_i^*(\varphi_i(n), \omega(n)))\right|\right|\right],$$

$$\leq \epsilon + O\left(\left(\frac{C_{\nabla^2_{\eta\eta}G}}{\lambda}\right)^K + \left(\frac{C_{\nabla^2_{\eta\eta}G}}{\lambda}\right)^{\bar{K}} + \delta + \frac{1}{\min_i\{\sqrt{D_i^{\text{tr}}}\}_{i=1}^m}\right)$$

*after at most $N = \frac{\min\{C_1, C_2\}}{B\epsilon^2}$ iterations. The expressions of the positive constants $C_1$ and $C_2$ are in Appendix A.9.*

Theorem 1 shows that Algorithm 1 reaches the set of $\epsilon$-FOSP at the iteration complexity $O(\frac{1}{\epsilon^2})$. Moreover, to reduce $E[||\frac{1}{m}\sum_{i=1}^m \nabla L_i||]$ and reach the set of $\epsilon$-FOSP within fewer iterations in

Algorithm 1, we have the following choices: (i) increase the iteration number $K$ of solving the lower-level problem (3) and the iteration number $\bar{K}$ in Algorithm 3; (ii) choose smaller $\delta$ in the first-order approximations (6)-(7); (iii) sample larger size $B$ of training tasks at each iteration $n$ in Algorithm 1; (iv) choose larger size $D_i^{\text{tr}}$ of training data of each training task $\mathcal{T}_i$.

## 4.2 Generalization analysis

The goal of meta-learning is to learn good meta-priors such that the reward and cost functions adapted from the learned meta-priors can have good performance on new tasks. Theorem 1 shows that Algorithm 1 can find meta-priors $(\bar{\theta}, \bar{\omega})$ such that the average loss function of the $m$ training tasks can reach the set of $\epsilon$-FOSP. However, it does not provide insights into how the task-specific reward and cost adaptations $(\hat{\varphi}_{m+1}, \hat{\eta}_{m+1}^*(\hat{\varphi}_{m+1}, \bar{\omega}, \mathcal{D}_{m+1}))$, adapted from the learned meta-priors $(\bar{\theta}, \bar{\omega})$, perform on an arbitrary new task $\mathcal{T}_{m+1}$ where $\mathcal{D}_{m+1}$ is the small data set of the new task $\mathcal{T}_{m+1}$. Given that the loss function in our problem is the negative log-likelihood function $L_i$, we use $L_{m+1}(\theta, \omega)|_{\theta=\hat{\varphi}_{m+1}, \omega=\hat{\eta}_{m+1}^*(\hat{\varphi}_{m+1}, \bar{\omega}, \mathcal{D}_{m+1})}$ as the metric to reason about the performance of the task-specific adaptations $(\hat{\varphi}_{m+1}, \hat{\eta}_{m+1}^*(\hat{\varphi}_{m+1}, \bar{\omega}, \mathcal{D}_{m+1}))$ on an arbitrary new task $\mathcal{T}_{m+1}$.

We start our analysis with the definition of stationary state-action distribution. For a given policy $\pi$, the corresponding stationary state-action distribution is $\mu^\pi(s, a) \triangleq (1 - \gamma) \sum_{t=0}^\infty \gamma^t \mathbb{P}_t^\pi(s, a)$ where $\mathbb{P}_t^\pi(s, a)$ is the probability of policy $\pi$ visiting $(s, a)$ at time $t$. We then define the distance between two tasks $\mathcal{T}_i$ and $\mathcal{T}_j$ as $d(\mu^{\pi_i}, \mu^{\pi_j}) \triangleq \int_{s \in \mathcal{S}} \int_{a \in \mathcal{A}} |\mu^{\pi_i}(s, a) - \mu^{\pi_j}(s, a)| da ds$. Recall that $\pi_i$ is the expert's policy in task $\mathcal{T}_i$.

**Remark on the definition of the task distance**. *While it seems natural to use the distance between the reward functions and the distance between the cost functions to define the distance between different tasks, this kind of definition can cause ambiguity because different reward and cost functions may result in the same task. For example, in an unconstrained environment, multiplying the reward function by a constant does not change the task because this will lead to the same optimal policy.*

**Proposition 1.** *For any new task $\mathcal{T}_{m+1}$ and any parameters $(\theta, \omega)$, the following relation holds:* $||\frac{1}{m} \sum_{i=1}^m \nabla L_i(\theta, \omega) - \nabla L_{m+1}(\theta, \omega)|| \leq O\big(d(\frac{1}{m} \sum_{i=1}^m \mu^{\pi_i}, \mu^{\pi_{m+1}})\big)$.

**Theorem 2.** *For an arbitrary new task $\mathcal{T}_{m+1}$, the task-specific reward and cost adaptations $(\hat{\varphi}_{m+1}, \hat{\eta}_{m+1}^*(\hat{\varphi}_{m+1}, \bar{\omega}, \mathcal{D}_{m+1}))$ adapted from the learned meta-priors $(\bar{\theta}, \bar{\omega})$ have the property:*

$$E\left[||\nabla L_{m+1}(\theta, \omega)|_{\theta=\hat{\varphi}_{m+1}, \omega=\hat{\eta}_{m+1}^*(\hat{\varphi}_{m+1}, \bar{\omega}, \mathcal{D}_{m+1})}||\right],$$

$$\leq O\left(\epsilon + \frac{1}{m} \sum_{i=1}^m d(\mu^{\pi_i}, \mu^{\pi_{m+1}}) + d(\frac{1}{m} \sum_{i=1}^m \mu^{\pi_i}, \mu^{\pi_{m+1}})\right).$$

Theorem 2 shows that if the new task's stationary state-action distribution is sufficiently close to the training tasks', the task-specific adaptations $(\hat{\varphi}_{m+1}, \hat{\eta}_{m+1}^*)$ are near-stationary.

**Theorem 3.** *If the learned meta-priors $(\bar{\theta}, \bar{\omega})$ of Algorithm 1 satisfy $E[\frac{1}{m} \sum_{i=1}^m L_i(\bar{\varphi}_i, \eta_i^*(\bar{\varphi}_i, \bar{\omega}))]$ $- \min_{\theta, \omega} \frac{1}{m} \sum_{i=1}^m L_i(\varphi_i, \eta_i^*(\varphi_i, \omega)) \leq \epsilon$ where $\bar{\varphi}_i = \bar{\theta} - \alpha \frac{\partial}{\partial \theta} L_i(\bar{\theta}, \eta_i^*(\bar{\theta}, \bar{\omega}))$, then it holds that*

$$E[L_{m+1}(\hat{\varphi}_{m+1}, \hat{\eta}_{m+1}^*(\hat{\varphi}_{m+1}, \bar{\omega}, \mathcal{D}_{m+1}))] - \min_{\theta, \omega} L_{m+1}(\theta, \omega),$$

$$\leq \epsilon + O\left(\frac{1}{m} \sum_{i=1}^m d(\mu^{\pi_i}, \mu^{\pi_{m+1}}) + d(\frac{1}{m} \sum_{i=1}^m \mu^{\pi_i}, \mu^{\pi_{m+1}})\right).$$

Theorem 3 shows that if the learned meta-priors $(\bar{\theta}, \bar{\omega})$ are $\epsilon$-optimal and the new task is close to the training tasks, the task-specific adaptations $(\hat{\varphi}_{m+1}, \hat{\eta}_{m+1}^*)$ are near-optimal.

If the reward and cost functions are linear, we have the following stronger results:

**Theorem 4.** *If the expert's reward and cost functions and the parameterized reward and cost functions $r_\theta$, $c_\omega$ are linear, we have that (i) $E[|J_{r_{m+1}}(\pi_{\hat{\eta}_{m+1}^*; \hat{\varphi}_{m+1}}) - J_{r_{m+1}}(\pi_{m+1})|] \leq O(\epsilon + \frac{1}{m} \sum_{i=1}^m d(\mu^{\pi_i}, \mu^{\pi_{m+1}}) + d(\frac{1}{m} \sum_{i=1}^m \mu^{\pi_i}, \mu^{\pi_{m+1}}))$; (ii) $E[|J_{c_{m+1}}(\pi_{\hat{\eta}_{m+1}^*; \hat{\varphi}_{m+1}}) - J_{c_{m+1}}(\pi_{m+1})|] \leq O(\epsilon + \frac{1}{m} \sum_{i=1}^m d(\mu^{\pi_i}, \mu^{\pi_{m+1}}) + d(\frac{1}{m} \sum_{i=1}^m \mu^{\pi_i}, \mu^{\pi_{m+1}}))$.

Theorem 4 shows that (i) the cumulative reward difference and (ii) the cumulative cost difference between the adapted policy $\pi_{\hat{\varphi}_{m+1};\hat{\eta}^*_{m+1}}$ and the expert's policy $\pi_{m+1}$ on an arbitrary new task $\mathcal{T}_{m+1}$ can be sufficiently small if the new task is close to the training tasks.

## 5 EXPERIMENT

This section includes two classes of experiments to validate the effectiveness of M-ICRL. The first experiment is conducted on a physical drone and the second experiment is conducted in Mujoco. Due to space limit, the experiment details are included in Appendix B.

### 5.1 DRONE NAVIGATION WITH OBSTACLES

We conduct a navigation experiment on an AR. Drone 2.0 (Figure 1) where the drone (in the yellow box) needs to navigate to the destination (in the green box) while avoiding collision with the obstacles (in the red box). We use an indoor motion capture system "Vicon" to record the trajectories of the drone. For different tasks, we vary the locations of the goal and the obstacles. Given that there is no ground truth reward in this experiment, we use two metrics "constraint violation rate" (CVR) and "success rate" (SR) where CVR is the percentage of the learned policy colliding with any obstacle and SR is the percentage of the learned policy reaching the destination and avoiding obstacles. We use

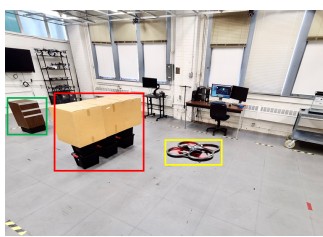

Figure 1: Drone navigation

50 training tasks and 10 test tasks where each test task has only one demonstration. We use three baselines for comparisons: **ICRL** (Liu & Zhu, 2022) which does not have meta-priors and directly learns from one demonstration without meta-priors, **ICRL(pre)** which naively pre-trains meta-priors by maximizing the likelihood across all the demonstrations of all the training tasks, **Meta-IRL** (Xu et al., 2019) which only learns a reward meta-prior using MAML. We include the experiment results in the second row of Table 1. The experiment details are included in Appendix B.

### 5.2 MUJOCO EXPERIMENT

We also conduct three experiments in Mujoco: Swimmer, HalfCheetah, and Walker. Given that Mujoco can output the ground truth reward, we use cumulative reward (CR) to replace the metric SR. Since there are no constraints in the original Mujoco environments, we add several constraints to the three Mujoco environments. The experiment details are in Appendix B.

Table 1: Experiment results.

| Task | Metric | M-ICRL | ICRL | ICRL(pre) | Meta-IRL | Expert |
|---|---|---|---|---|---|---|
| Drone | SR | $0.96 \pm 0.02$ | $0.62 \pm 0.07$ | $0.71 \pm 0.06$ | $0.45 \pm 0.10$ | $1.00 \pm 0.00$ |
| | CVR | $0.02 \pm 0.02$ | $0.16 \pm 0.10$ | $0.11 \pm 0.08$ | $0.33 \pm 0.12$ | $0.00 \pm 0.00$ |
| Swimmer | CR | $322.56 \pm 48.68$ | $76.44 \pm 18.26$ | $199.03 \pm 53.24$ | $113.66 \pm 32.51$ | $376.10 \pm 51.51$ |
| | CVR | $0.04 \pm 0.02$ | $0.22 \pm 0.13$ | $0.16 \pm 0.06$ | $0.35 \pm 0.18$ | $0.00 \pm 0.00$ |
| HalfCheetah | CR | $228.78 \pm 54.23$ | $60.74 \pm 32.63$ | $156.89 \pm 50.47$ | $108.05 \pm 36.89$ | $264.00 \pm 165.56$ |
| | CVR | $0.03 \pm 0.01$ | $0.28 \pm 0.19$ | $0.20 \pm 0.11$ | $0.31 \pm 0.10$ | $0.00 \pm 0.00$ |
| Walker | CR | $712.40 \pm 96.53$ | $144.79 \pm 66.37$ | $311.86 \pm 56.99$ | $165.86 \pm 70.08$ | $752.40 \pm 84.71$ |
| | CVR | $0.00 \pm 0.00$ | $0.26 \pm 0.18$ | $0.22 \pm 0.09$ | $0.42 \pm 0.26$ | $0.00 \pm 0.00$ |

From Table 1, we observe that M-ICRL achieves the best performance in all the four experiments. Meta-IRL has much higher constraint violation rate than the other three algorithms. This shows the benefits of learning both the reward function and constraints. ICRL(pre), which simply learns meta-priors across all the demonstrations of all the tasks, performs poorly. This illustrates the benefits of our meta-learning design for M-ICRL. We discuss the experiment results in detail in Appendix B.

## 6 Conclusion and future works

We propose M-ICRL, the first theoretical framework that can learn reward and cost functions of the expert from few demonstrations by first learning meta-priors from other related tasks. It is shown both theoretically and empirically that M-ICRL is effective to adapt to new tasks from few demonstrations. Despite its benefits, one limitation is that M-ICRL assumes that the states and actions are fully observable, however, this may not hold in some real-world problems due to practical issues such as noise. A future direction is to extend M-ICRL to partially observable MDPs (POMDPs).

## 7 Ackowledgement

This work is partially supported by the National Science Foundation through grants ECCS 1846706 and ECCS 2140175. We would like to thank the reviewers for their insightful and constructive suggestions.

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

# A APPENDIX

This section includes the proof and simulation details. At the beginning, we define the empirical cumulative reward. Given a trajectory $\zeta$, the empirical cumulative reward is defined as $\hat{J}_r(\zeta) \triangleq \sum_{t=0}^{\infty} \gamma^t r(s_t, a_t)$.

## A.1 THE DERIVATION OF PROBLEM (1)

The derivation of problem (1) is first introduced in (Liu & Zhu, 2022). For the sake of better understanding, we include it here. The fundamental idea of the ICRL problem (1) is to learn a reward function in the upper level and learn the corresponding policy and constraints in the lower level. In the following, we first introduce the lower-level problem and then introduce the bi-level problem formulation.

**The lower-level optimization problem**. Given a reward function $r_\theta$, the lower-level problem aims to learn the corresponding policy (and constraints). Therefore, we formulate the following constrained RL problem (11):

$$\max_{\pi} \ H(\pi) + J_{r_\theta}(\pi), \quad \text{s.t. } \mu_{c_i}(\pi) = \mu_{c_i}(\pi_i), \tag{11}$$

where $\mu_{c_i}(\pi) \triangleq E_{\zeta \sim P_\pi}[\sum_{t=0}^{\infty} \gamma^t \phi_{c_i}(s_t, a_t)]$ is the expected cumulative cost feature of policy $\pi$ and $\phi_{c_i}$ is the cost feature of task $\mathcal{T}_i$. Problem (11) aims to find a policy that maximizes the entropy-regularized cumulative reward $H(\pi) + J_{r_\theta}(\pi)$ subject to the constraint of cost expectation matching. The cost feature expectation matching follows the idea of "feature expectation matching" in (Abbeel & Ng, 2004; Ziebart et al., 2008; 2010).

We cannot directly solve problem (11) because it is non-convex. Therefore, we use dual methods and solve its dual problem. The dual problem of problem (11) is (Liu & Zhu, 2022)

$$\min \ G_i(\omega; \theta) = H(\pi_{\omega;\theta}) + J_{r_\theta}(\pi_{\omega;\theta}) + \omega^\top (\mu_{c_i}(\pi_i) - \mu_{c_i}(\pi)), \tag{12}$$

where $\omega$ is the dual variable and $\pi_{\omega;\theta}$ is the constrained soft Bellman policy. Since (Liu & Zhu, 2022) studies linear cost functions, $\omega^\top \mu_{c_i}(\pi) = J_{c_\omega}(\pi)$. Here, we extend the domain from linear cost functions to non-linear cost functions and directly use $J_{c_\omega}(\pi)$. We use the dual problem (12) as the lower-level problem in (1).

**The bi-level optimization problem**. (Liu & Zhu, 2022) has proved that the optimal solution of the primal problem (11) is $\pi_{\omega^*(\theta);\theta}$ where $\omega^*(\theta) = \arg\min_\omega G_i(\omega; \theta)$. The upper-level problem aims to learn a reward function $r_\theta$ such that the corresponding policy (i.e., the optimal solution of problem (11)) can minimize the negative log-likelihood of the expert's trajectories:

$$\min_{\theta} \ L_i(\theta, \omega^*(\theta)) \triangleq E_{\zeta \sim P_\pi}[-\sum_{t=0}^{\infty} \gamma^t \log \pi_{\omega^*(\theta);\theta}(a_t|s_t)],$$

$$\text{s.t.} \quad \omega^*(\theta) = \arg\min_\omega G_i(\omega; \theta).$$

Then we reach the problem formulation in (1).

## A.2 NOTIONS AND NOTATIONS

Define $J_{r_\theta}^\pi(s) \triangleq E_{\zeta \sim P_\pi}[\sum_{t=0}^{\infty} \gamma^t r_\theta(s_t, a_t)|s_0 = s]$ and $J_{r_\theta}^\pi(s, a) \triangleq E_{\zeta \sim P_\pi}[\sum_{t=0}^{\infty} \gamma^t r_\theta(s_t, a_t)|s_0 = s, a_0 = a]$. Similarly, we can define $J_{c_\omega}^\pi(s)$ and $J_{c_\omega}^\pi(s, a)$.

Define $H^\pi(s, a) \triangleq -E_{\zeta \sim P_\pi}[\sum_{t=0}^{\infty} \gamma^t \log \pi(a_t|s_t)|s_0 = s, a_0 = a]$ and $H^\pi(s) \triangleq -E_{\zeta \sim P_\pi}[\sum_{t=0}^{\infty} \gamma^t \log \pi(a_t|s_t)|s_0 = s]$.

The constrained soft Bellman policy (Liu & Zhu, 2022) is

$$\pi_{\omega;\theta}(a|s) = \frac{\exp(Q_{\omega;\theta}^{\text{soft}}(s, a))}{\exp(V_{\omega;\theta}^{\text{soft}}(s))},$$

$$Q_{\omega;\theta}^{\text{soft}}(s, a) = r_\theta(s, a) - c_\omega(s, a) + \gamma \int_{s' \in \mathcal{S}} P(s'|s, a) V_{\omega;\theta}^{\text{soft}}(s') ds',$$

$$V_{\omega;\theta}^{\text{soft}}(s) = \log\left(\int_{a\in\mathcal{A}} \exp(Q_{\omega;\theta}^{\text{soft}}(s,a))da\right).$$

We can approximate the constrained soft Bellman policy using soft Q-learning or soft actor-critic by treating $(r_\theta - c_\omega)$ as the reward function.

**Lemma 3.** *We have that*

$$\nabla_\theta \log \pi_{\omega;\theta}(a|s) = E_{\zeta\sim P_{\pi_{\omega;\theta}}}[\sum_{t=0}^{\infty} \gamma^t \nabla_\theta r_\theta(s_t, a_t)|s_0 = s, a_0 = a]$$

$$- E_{\zeta\sim P_{\pi_{\omega;\theta}}}[\sum_{t=0}^{\infty} \gamma^t \nabla_\theta r_\theta(s_t, a_t)|s_0 = s],$$

$$\nabla_\omega \log \pi_{\omega;\theta}(a|s) = E_{\zeta\sim P_{\pi_{\omega;\theta}}}[\sum_{t=0}^{\infty} \gamma^t \nabla_\omega c_\omega(s_t, a_t)|s_0 = s]$$

$$- E_{\zeta\sim P_{\pi_{\omega;\theta}}}[\sum_{t=0}^{\infty} \gamma^t \nabla_\omega c_\omega(s_t, a_t)|s_0 = s, a_0 = a].$$

*Proof.*

$$\nabla_\theta Q_{\omega;\theta}^{\text{soft}}(s,a) = \nabla_\theta r_\theta(s,a) + \gamma \int_{s'\in\mathcal{S}} P(s'|s,a)\nabla_\theta V_{\omega;\theta}^{\text{soft}}(s')ds',$$

$$= \nabla_\theta r_\theta(s,a) + \gamma \int_{s'\in\mathcal{S}} P(s'|s,a)\frac{\nabla_\theta \int_{a'\in\mathcal{A}} \exp(Q_{\omega;\theta}^{\text{soft}}(s',a'))da'}{\exp(V_{\omega;\theta}^{\text{soft}}(s'))}ds',$$

$$= \nabla_\theta r_\theta(s,a) + \gamma \int_{s'\in\mathcal{S}} P(s'|s,a)\int_{a'\in\mathcal{A}} \frac{\exp(Q_{\omega;\theta}^{\text{soft}}(s',a'))\nabla_\theta Q_{\omega;\theta}^{\text{soft}}(s',a')}{\exp(V_{\omega;\theta}^{\text{soft}}(s'))}da'ds',$$

$$\overset{(a)}{=} \nabla_\theta r_\theta(s,a) + \gamma \int_{s'\in\mathcal{S}} P(s'|s,a)\int_{a'\in\mathcal{A}} \pi_{\omega;\theta}(a|s)\nabla_\theta Q_{\omega;\theta}^{\text{soft}}(s',a')da'ds',$$

where $(a)$ follows the definition of $\pi_{\omega;\theta}$. Keep the expansion, we can see that

$$\nabla_\theta Q_{\omega;\theta}^{\text{soft}}(s,a) = E_{\zeta\sim P_{\pi_{\omega;\theta}}}[\sum_{t=0}^{\infty} \gamma^t \nabla_\theta r_\theta(s_t, a_t)|s_0 = s, a_0 = a],$$

$$\nabla_\theta V_{\omega;\theta}^{\text{soft}}(s) = E_{\zeta\sim P_{\pi_{\omega;\theta}}}[\sum_{t=0}^{\infty} \gamma^t \nabla_\theta r_\theta(s_t, a_t)|s_0 = s].$$

Therefore, we can see that

$$\nabla_\theta \log \pi_{\omega;\theta}(a|s) = \nabla_\theta Q_{\omega;\theta}^{\text{soft}}(s,a) - \nabla_\theta V_{\omega;\theta}^{\text{soft}}(s),$$

$$= E_{\zeta\sim P_{\pi_{\omega;\theta}}}[\sum_{t=0}^{\infty} \gamma^t \nabla_\theta r_\theta(s_t, a_t)|s_0 = s, a_0 = a] - E_{\zeta\sim P_{\pi_{\omega;\theta}}}[\sum_{t=0}^{\infty} \gamma^t \nabla_\theta r_\theta(s_t, a_t)|s_0 = s].$$

Similarly, we can get

$$\nabla_\omega \log \pi_{\omega;\theta}(a|s) = E_{\zeta\sim P_{\pi_{\omega;\theta}}}[\sum_{t=0}^{\infty} \gamma^t \nabla_\omega c_\omega(s_t, a_t)|s_0 = s]$$

$$- E_{\zeta\sim P_{\pi_{\omega;\theta}}}[\sum_{t=0}^{\infty} \gamma^t \nabla_\omega c_\omega(s_t, a_t)|s_0 = s, a_0 = a].$$

$\square$

**Lemma 4.** *We have the following relations:*

$$\nabla_\omega H^{\pi_{\omega;\theta}}(s,a)$$

$$= -\nabla_\omega \log \pi_{\omega;\theta}(a|s) + E_{\zeta \sim P_{\pi_{\omega;\theta}}}\left[\sum_{t=1}^\infty \gamma^t \nabla_\omega \log \pi_{\omega;\theta}(a_t|s_t)[H^{\pi_{\omega;\theta}}(s_t,a_t)-1]\Big| s_0=s, a_0=a\right],$$

$$\nabla_\omega H^{\pi_{\omega;\theta}}(s) = E_{\zeta \sim P_{\pi_{\omega;\theta}}}\left[\sum_{t=0}^\infty \gamma^t \nabla_\omega \log \pi_{\omega;\theta}(a_t|s_t)[H^{\pi_{\omega;\theta}}(s_t,a_t)-1]\Big| s_0=s\right],$$

$$\nabla_\omega J_{c_\omega}^{\pi_{\omega;\theta}}(s,a) = \nabla_\omega c_\omega(s,a)$$

$$+ E_{\zeta \sim P_{\pi_{\omega;\theta}}}\left[\sum_{t=1}^\infty \gamma^t [\nabla_\omega c_\omega(s_t,a_t) + \nabla_\omega \log \pi_{\omega;\theta}(a_t|s_t) \cdot J_{c_\omega}^{\pi_{\omega;\theta}}(s_t,a_t)]\Big| s_0=s, a_0=a\right],$$

$$\nabla_\omega J_{c_\omega}^{\pi_{\omega;\theta}}(s) = E_{\zeta \sim P_{\pi_{\omega;\theta}}}\left[\sum_{t=0}^\infty \gamma^t [\nabla_\omega c_\omega(s_t,a_t) + \nabla_\omega \log \pi_{\omega;\theta}(a_t|s_t) \cdot J_{c_\omega}^{\pi_{\omega;\theta}}(s_t,a_t)]\Big| s_0=s\right],$$

$$\nabla_\omega J_{r_\theta}^{\pi_{\omega;\theta}}(s,a) = E_{\zeta \sim P_{\pi_{\omega;\theta}}}\left[\sum_{t=1}^\infty \gamma^t \nabla_\omega \log \pi_{\omega;\theta}(a_t|s_t) \cdot J_{r_\theta}^{\pi_{\omega;\theta}}(s_t,a_t)\Big| s_0=s, a_0=a\right],$$

$$\nabla_\omega J_{r_\theta}^{\pi_{\omega;\theta}}(s) = E_{\zeta \sim P_{\pi_{\omega;\theta}}}\left[\sum_{t=0}^\infty \gamma^t \nabla_\omega \log \pi_{\omega;\theta}(a_t|s_t) \cdot J_{r_\theta}^{\pi_{\omega;\theta}}(s_t,a_t)\Big| s_0=s\right].$$

*Proof.*

$$\nabla_\omega H^{\pi_{\omega;\theta}}(s,a) = -\nabla_\omega \log \pi_{\omega;\theta}(a|s) + \gamma \int_{s'\in\mathcal{S}} P(s'|s,a) \cdot \nabla_\omega H^{\pi_{\omega;\theta}}(s')ds',$$

$$= -\nabla_\omega \log \pi_{\omega;\theta}(a|s) + \gamma \int_{s'\in\mathcal{S}} P(s'|s,a) \cdot \nabla_\omega \int_{a'\in\mathcal{A}} \pi_{\omega;\theta}(a'|s')H^{\pi_{\omega;\theta}}(s',a')da'ds',$$

$$= -\nabla_\omega \log \pi_{\omega;\theta}(a|s) + \gamma \int_{s'\in\mathcal{S}} P(s'|s,a) \int_{a'\in\mathcal{A}}\left[\nabla_\omega \pi_{\omega;\theta}(a'|s') \cdot H^{\pi_{\omega;\theta}}(s',a')\right.$$

$$+ \pi_{\omega;\theta}(a'|s') \cdot \nabla_\omega H^{\pi_{\omega;\theta}}(s',a')\Big]da'ds',$$

$$= -\nabla_\omega \log \pi_{\omega;\theta}(a|s) + \gamma \int_{s'\in\mathcal{S}} P(s'|s,a) \int_{a'\in\mathcal{A}} \pi_{\omega;\theta}(a'|s') \cdot \left[\nabla_\omega \log \pi_{\omega;\theta}(a'|s') \cdot H^{\pi_{\omega;\theta}}(s',a')\right.$$

$$+ \nabla_\omega H^{\pi_{\omega;\theta}}(s',a')\Big]da'ds'.$$

Keep the expansion, we have that

$$\nabla_\omega H^{\pi_{\omega;\theta}}(s,a)$$

$$= -\nabla_\omega \log \pi_{\omega;\theta}(a|s) + E_{\zeta \sim P_{\pi_{\omega;\theta}}}\left[\sum_{t=1}^\infty \gamma^t \nabla_\omega \log \pi_{\omega;\theta}(a_t|s_t)[H^{\pi_{\omega;\theta}}(s_t,a_t)-1]\Big| s_0=s, a_0=a\right],$$

$$\nabla_\omega H^{\pi_{\omega;\theta}}(s) = E_{\zeta \sim P_{\pi_{\omega;\theta}}}\left[\sum_{t=0}^\infty \gamma^t \nabla_\omega \log \pi_{\omega;\theta}(a_t|s_t)[H^{\pi_{\omega;\theta}}(s_t,a_t)-1]\Big| s_0=s\right].$$

Similarly, we can see that

$$\nabla_\omega J_{c_\omega}^{\pi_{\omega;\theta}}(s,a) = \nabla_\omega c_\omega(s,a) + \gamma \int_{s'\in\mathcal{S}} P(s'|s,a) \cdot \nabla_\omega J_{c_\omega}^{\pi_{\omega;\theta}}(s')ds',$$

$$= \nabla_\omega c_\omega(s,a) + \gamma \int_{s'\in\mathcal{S}} P(s'|s,a) \cdot \nabla_\omega \int_{a'\in\mathcal{A}} \pi_{\omega;\theta}(a'|s')J_{c_\omega}^{\pi_{\omega;\theta}}(s',a')da'ds',$$

$$= \nabla_\omega c_\omega(s,a) + \gamma \int_{s'\in\mathcal{S}} P(s'|s,a) \int_{a'\in\mathcal{A}} \pi_{\omega;\theta}(a'|s')\left[\nabla_\omega \log \pi_{\omega;\theta}(a'|s') \cdot J_{c_\omega}^{\pi_{\omega;\theta}}(s',a')\right.$$

$$+ \nabla_\omega J_{c_\omega}^{\pi_{\omega;\theta}}(s', a') \Big] da' ds'.$$

Keep the expansion, we can get

$$\nabla_\omega J_{c_\omega}^{\pi_{\omega;\theta}}(s, a) = \nabla_\omega c_\omega(s, a)$$

$$+ E_{\zeta \sim P_{\pi_{\omega;\theta}}} \left[ \sum_{t=1}^\infty \gamma^t [\nabla_\omega c_\omega(s_t, a_t) + \nabla_\omega \log \pi_{\omega;\theta}(a_t|s_t) \cdot J_{c_\omega}^{\pi_{\omega;\theta}}(s_t, a_t)] \Big| s_0 = s, a_0 = a \right],$$

$$\nabla_\omega J_{c_\omega}^{\pi_{\omega;\theta}}(s) = E_{\zeta \sim P_{\pi_{\omega;\theta}}} \left[ \sum_{t=0}^\infty \gamma^t [\nabla_\omega c_\omega(s_t, a_t) + \nabla_\omega \log \pi_{\omega;\theta}(a_t|s_t) \cdot J_{c_\omega}^{\pi_{\omega;\theta}}(s_t, a_t)] \Big| s_0 = s \right].$$

$$\nabla_\omega J_{r_\theta}^{\pi_{\omega;\theta}}(s, a) = \gamma \int_{s' \in \mathcal{S}} P(s'|s, a) \cdot \nabla_\omega J_{r_\theta}^{\pi_{\omega;\theta}}(s') ds',$$

$$= \gamma \int_{s' \in \mathcal{S}} P(s'|s, a) \cdot \nabla_\omega \int_{a' \in \mathcal{A}} \pi_{\omega;\theta}(a'|s') J_{r_\theta}^{\pi_{\omega;\theta}}(s', a') da' ds',$$

$$= \gamma \int_{s' \in \mathcal{S}} P(s'|s, a) \cdot \int_{a' \in \mathcal{A}} \left[ \nabla_\omega \pi_{\omega;\theta}(a'|s') \cdot J_{r_\theta}^{\pi_{\omega;\theta}}(s', a') + \pi_{\omega;\theta}(a'|s') \cdot \nabla_\omega J_{r_\theta}^{\pi_{\omega;\theta}}(s', a') \right] da' ds',$$

$$= \gamma \int_{s' \in \mathcal{S}} P(s'|s, a) \int_{a' \in \mathcal{A}} \pi_{\omega;\theta}(a'|s') \nabla_\omega \log \pi_{\omega;\theta}(a'|s') \cdot J_{r_\theta}^{\pi_{\omega;\theta}}(s', a') da' ds',$$

$$= E_{\zeta \sim P_{\pi_{\omega;\theta}}} \left[ \sum_{t=1}^\infty \gamma^t \nabla_\omega \log \pi_{\omega;\theta}(a_t|s_t) \cdot J_{r_\theta}^{\pi_{\omega;\theta}}(s_t, a_t) \Big| s_0 = s, a_0 = a \right].$$

Therefore, we can get that

$$\nabla_\omega J_{r_\theta}^{\pi_{\omega;\theta}}(s) = E_{\zeta \sim P_{\pi_{\omega;\theta}}} \left[ \sum_{t=0}^\infty \gamma^t \nabla_\omega \log \pi_{\omega;\theta}(a_t|s_t) \cdot J_{r_\theta}^{\pi_{\omega;\theta}}(s_t, a_t) \Big| s_0 = s \right].$$

$\square$

### A.3 EXPRESSIONS OF GRADIENTS

We provide the expressions of all the gradients below:

$$\nabla_\omega G_i(\omega; \theta) = \nabla_\omega J_{c_\omega}(\pi_i) - E_{\zeta \sim P_{\pi_{\omega;\theta}}} \left[ \sum_{t=0}^\infty \gamma^t \nabla_\omega c_\omega(s_t, a_t) \right],$$

$$\nabla_\omega \hat{G}_i(\omega; \theta, \mathcal{D}) = \frac{1}{D} \sum_{j=1}^D \nabla_\omega \hat{J}_{c_\omega}(\zeta^j) - E_{\zeta \sim P_{\pi_{\omega;\theta}}} \left[ \sum_{t=0}^\infty \gamma^t \nabla_\omega c_\omega(s_t, a_t) \right],$$

$$\nabla_\theta G_i(\omega; \theta) = E_{\zeta \sim P_{\pi_{\omega;\theta}}} \left[ \sum_{t=0}^\infty \gamma^t \nabla_\theta r_\theta(s_t, a_t) \right] - \nabla_\theta J_{r_\theta}(\pi_i),$$

$$\nabla_\theta \hat{G}_i(\omega; \theta, \mathcal{D}) = E_{\zeta \sim P_{\pi_{\omega;\theta}}} \left[ \sum_{t=0}^\infty \gamma^t \nabla_\theta r_\theta(s_t, a_t) \right] - \frac{1}{D} \sum_{j=1}^D \nabla_\theta \hat{J}_{r_\theta}(\zeta^j),$$

$$\nabla_{\omega\omega}^2 G_i(\omega; \theta) = \nabla_{\omega\omega}^2 \hat{G}_i(\omega; \theta, \mathcal{D}) = -E_{\zeta \sim P_{\pi_{\omega;\theta}}} \left[ \sum_{t=0}^\infty \gamma^t \left( \text{Cov}_\omega(s_t) + \nabla_{\omega\omega}^2 c_\omega(s_t, a_t) \right) \right],$$

$$\nabla_{\omega\theta}^2 G_i(\omega; \theta) = \nabla_{\omega\theta}^2 \hat{G}_i(\omega; \theta, \mathcal{D}) = -E_{\zeta \sim P_{\pi_{\omega;\theta}}} \left[ \sum_{t=0}^\infty \gamma^t \nabla_\theta \log \pi_{\omega;\theta}(a|s)(\nabla_\omega J_{c_\omega}^\pi(s, a)|_{\pi=\pi_{\omega;\theta}})^\top \right],$$

$$\nabla_\theta L_i(\theta, \omega) = E_{\zeta \sim P_{\pi_{\omega;\theta}}} \left[ \sum_{t=0}^\infty \gamma^t \nabla_\theta r_\theta(s_t, a_t) \right] - \nabla_\theta J_{r_\theta}(\pi_i),$$

$$\nabla_\theta \hat{L}_i(\theta, \omega, \mathcal{D}) = E_{\zeta \sim P_{\pi_{\omega;\theta}}} \left[ \sum_{t=0}^\infty \gamma^t \nabla_\theta r_\theta(s_t, a_t) \right] - \frac{1}{D} \sum_{j=1}^D \nabla_\theta \hat{J}_{r_\theta}(\zeta^j),$$

$$\nabla_\omega L_i(\theta,\omega) = \nabla_\omega J_{c_\omega}(\pi_i) - E_{\zeta\sim P_{\pi_{\omega;\theta}}}[\sum_{t=0}^\infty \gamma^t \nabla_\omega c_\omega(s_t,a_t)],$$

$$\nabla_\omega \hat{L}_i(\theta,\omega,\mathcal{D}) = \frac{1}{D}\sum_{j=1}^D \nabla_\omega \hat{J}_{c_\omega}(\zeta^j) - E_{\zeta\sim P_{\pi_{\omega;\theta}}}[\sum_{t=0}^\infty \gamma^t \nabla_\omega c_\omega(s_t,a_t)],$$

$$\nabla_{\omega\omega}^2 L_i(\theta,\omega) = \nabla_{\omega\omega}^2 \hat{L}_i(\theta,\omega,\mathcal{D}) = -E_{\zeta\sim P_{\pi_{\omega;\theta}}}\left[\sum_{t=0}^\infty \gamma^t\left(\text{Cov}_\omega(s_t) + \nabla_{\omega\omega}^2 c_\omega(s_t,a_t)\right)\right],$$

$$\nabla_{\omega\theta}^2 L_i(\theta,\omega) = \nabla_{\omega\theta}^2 \hat{L}_i(\theta,\omega,\mathcal{D}) = -E_{\zeta\sim P_{\pi_{\omega;\theta}}}\left[\sum_{t=0}^\infty \gamma^t \nabla_\theta \log \pi_{\omega;\theta}(a|s)(\nabla_\omega J_{c_\omega}^\pi(s,a)|_{\pi=\pi_{\omega;\theta}})^\top\right],$$

where the expression of $\text{Cov}_\omega(s)$ is in (14).

*Proof.* **Derivation of $\nabla_\omega G_i$ and $\nabla_\omega \hat{G}_i$.**

Since $\pi_{\omega;\theta} = \arg\max_\pi H(\pi) + J_{r_\theta}(\pi) - J_{c_\omega}(\pi)$, the following holds for any $(s,a) \in \mathcal{S}\times\mathcal{A}$:

$$\frac{\partial}{\partial\pi_t(a|s)}\left(H(\pi) + J_{r_\theta}(\pi) - J_{c_\omega}(\pi)\right)\bigg|_{\pi_t=\pi_{\omega;\theta}} = 0,$$

where we change policy to be time-dependent but force it to be stationary i.e., $\pi_t = \pi$. Therefore, we have that

$$\frac{\partial}{\partial\pi_t(a|s)}\left(H(\pi) + J_{r_\theta}(\pi) - J_{c_\omega}(\pi)\right)\bigg|_{\pi=\pi_{\omega;\theta}},$$

$$= -\gamma^t \mathbb{P}_t^{\pi_{\omega;\theta}}(s)(\log \pi_{\omega;\theta}(a|s) + 1) + \mathbb{P}_t^{\pi_{\omega;\theta}}(s)E_{\zeta\sim P_{\pi_{\omega;\theta}}}[\sum_{\tau=t+1}^\infty -\gamma^\tau \log \pi_{\omega;\theta}(a_\tau|s_\tau)|s_t=s,a_t=a]$$

$$+ \mathbb{P}_t^{\pi_{\omega;\theta}}(s)\left(\gamma^t r_\theta(s,a) - \gamma^t c_\omega(s,a) + E_{\zeta\sim P_{\pi_{\omega;\theta}}}[\sum_{\tau=t+1}^\infty \gamma^\tau(r_\theta(s_t,a_t) - c_\omega(s_t,a_t))|s_t=s,a_t=a]\right),$$

$$= \gamma^t \mathbb{P}_t^{\pi_{\omega;\theta}}(s)\left[H^{\pi_{\omega;\theta}}(s,a) - 1 + J_{r_\theta}^{\pi_{\omega;\theta}}(s,a) - J_{c_\omega}^{\pi_{\omega;\theta}}(s,a)\right] = 0. \tag{13}$$

Recall that $G_i(\omega;\theta) = H(\pi_{\omega;\theta}) + J_{r_\theta}(\pi_{\omega;\theta}) - J_{c_\omega}(\pi_{\omega;\theta}) + J_{c_\omega}(\pi_i)$. Therefore, we have that

$$\nabla_\omega G_i(\omega;\theta) = \int_{s_0\in\mathcal{S}} P_0(s_0)\cdot\left[\nabla_\omega H^{\pi_{\omega;\theta}}(s_0) + \nabla_\omega J_{r_\theta}^{\pi_{\omega;\theta}}(s_0) - \nabla_\omega J_{c_\omega}^{\pi_{\omega;\theta}}(s_0)\right]ds_0 + \nabla_\omega J_{c_\omega}(\pi_i),$$

$$\overset{(a)}{=} \int_{s_0\in\mathcal{S}} P_0(s_0)\cdot E_{\zeta\sim P_{\pi_{\omega;\theta}}}\left[\sum_{t=0}^\infty \gamma^t\left(\nabla_\omega \log \pi_{\omega;\theta}(a_t|s_t)\cdot\right.\right.$$

$$\left.\left.\left(H^{\pi_{\omega;\theta}}(s_t,a_t) - 1 + J_{r_\theta}^{\pi_{\omega;\theta}}(s_t,a_t) - J_{c_\omega}^{\pi_{\omega;\theta}}(s_t,a_t)\right) - \nabla_\omega c_\omega(s_t,a_t)\right)\right]ds_0 + \nabla_\omega J_{c_\omega}(\pi_i),$$

$$\overset{(b)}{=} \nabla_\omega J_\omega(\pi_i) - E_{\zeta\sim P_{\pi_{\omega;\theta}}}[\sum_{t=0}^\infty \gamma^t \nabla_\omega c_\omega(s_t,a_t)],$$

where $(a)$ follows Lemma 4 and $(b)$ follows (13). With similar derivation, we can get

$$\nabla_\omega \hat{G}_i(\omega;\theta,\mathcal{D}) = \frac{1}{D}\sum_{j=1}^D \nabla_\omega \hat{J}_{c_\omega}(\zeta^j) - E_{\zeta\sim P_{\pi_{\omega;\theta}}}[\sum_{t=0}^\infty \gamma^t \nabla_\omega c_\omega(s_t,a_t)].$$

**Derivation of $\nabla_{\omega\omega}^2 G_i$ and $\nabla_{\omega\omega}^2 \hat{G}_i$.**

We know that $\nabla_\omega J^\pi_{c_\omega}(s,a)|_{\pi=\pi_{\omega;\theta}} = \nabla_\omega E_{\zeta \sim P_{\pi_{\omega;\theta}}}[\sum_{t=0}^\infty \gamma^t \nabla_\omega c_\omega(s_t, a_t)|s_0 = s, a_0 = a]$ and $\nabla_\omega J^\pi_{c_\omega}(s)|_{\pi=\pi_{\omega;\theta}} = \nabla_\omega E_{\zeta \sim P_{\pi_{\omega;\theta}}}[\sum_{t=0}^\infty \gamma^t \nabla_\omega c_\omega(s_t, a_t)|s_0 = s]$.

$$\nabla^2_{\omega\omega} G_i(\omega;\theta) = -\nabla_\omega E_{\zeta \sim P_{\pi_{\omega;\theta}}}[\sum_{t=0}^\infty \gamma^t \nabla_\omega c_\omega(s_t, a_t)],$$

$$= -\int_{s_0 \in \mathcal{S}} P_0(s_0) \nabla_\omega \int_{a_0 \in \mathcal{A}} \pi_{\omega;\theta}(a_0|s_0) \cdot \nabla_\omega J^\pi_{c_\omega}(s_0, a_0)|_{\pi=\pi_{\omega;\theta}} da_0 ds_0,$$

$$= -\int_{s_0 \in \mathcal{S}} P_0(s_0) \int_{a_0 \in \mathcal{A}} \Big[ \nabla_\omega \pi_{\omega;\theta}(a_0|s_0) \cdot \nabla_\omega J^\pi_{c_\omega}(s_0, a_0)|_{\pi=\pi_{\omega;\theta}}$$

$$+ \pi_{\omega;\theta}(a_0|s_0) \cdot \nabla_\omega (\nabla_\omega J^\pi_{c_\omega}(s_0, a_0)|_{\pi=\pi_{\omega;\theta}}) \Big] da_0 ds_0,$$

$$= -\int_{s_0 \in \mathcal{S}} P_0(s_0) \int_{a_0 \in \mathcal{A}} \Big[ \nabla_\omega \pi_{\omega;\theta}(a_0|s_0) \cdot \nabla_\omega J^\pi_{c_\omega}(s_0, a_0)|_{\pi=\pi_{\omega;\theta}}$$

$$+ \pi_{\omega;\theta}(a_0|s_0) \cdot \nabla_\omega (\nabla_\omega c_\omega(s_0, a_0) + \int_{s_1 \in \mathcal{S}} P(s_1|s_0, a_0) \nabla_\omega J^\pi_{c_\omega}(s_1)|_{\pi=\pi_{\omega;\theta}} ds_1 \Big] da_0 ds_0.$$

Keep the expansion, we can get that

$$\nabla^2_{\omega\omega} G_i(\omega;\theta) = -\int_{s \in \mathcal{S}} \frac{\mu^{\pi_{\omega;\theta}}(s)}{1-\gamma} \int_{a \in \mathcal{A}} \pi_{\omega;\theta}(a|s) \nabla_\omega \log \pi_{\omega;\theta}(a|s) \nabla_\omega (J^\pi_{c_\omega}(s,a)|_{\pi=\pi_{\omega;\theta}})^\top da ds$$

$$- E_{\zeta \sim P_{\pi_{\omega;\theta}}}[\sum_{t=0}^\infty \gamma^t \nabla^2_{\omega\omega} c_\omega(s_t, a_t)].$$

Define the covariance

$$\text{Cov}_\omega(s) \triangleq \int_{a \in \mathcal{A}} \pi_{\omega;\theta}(a|s) \nabla_\omega \log \pi_{\omega;\theta}(a|s) (\nabla_\omega J^\pi_{c_\omega}(s,a)|_{\pi=\pi_{\omega;\theta}})^\top da,$$

$$\overset{(c)}{=} \int_{a \in \mathcal{A}} \pi_{\omega;\theta}(a|s) \Big[ \nabla_\omega J^\pi_{c_\omega}(s,a)|_{\pi=\pi_{\omega;\theta}} - \nabla_\omega J^\pi_{c_\omega}(s)|_{\pi=\pi_{\omega;\theta}} \Big] (\nabla_\omega J^\pi_{c_\omega}(s,a)|_{\pi=\pi_{\omega;\theta}})^\top da, \quad (14)$$

where $(c)$ follows Lemma 3. Therefore, we have that

$$\nabla^2_{\omega\omega} G_i(\omega;\theta) = -E_{\zeta \sim P_{\pi_{\omega;\theta}}} \Big[ \sum_{t=0}^\infty \gamma^t \Big( \text{Cov}_\omega(s_t) + \nabla^2_{\omega\omega} c_\omega(s_t, a_t) \Big) \Big],$$

and similarly we can get

$$\nabla^2_{\omega\omega} \hat{G}_i(\omega;\theta,\mathcal{D}) = -E_{\zeta \sim P_{\pi_{\omega;\theta}}} \Big[ \sum_{t=0}^\infty \gamma^t \Big( \text{Cov}_\omega(s_t) + \nabla^2_{\omega\omega} c_\omega(s_t, a_t) \Big) \Big].$$

**Derivation of $\nabla^2_{\omega\theta} G_i$ and $\nabla^2_{\omega\theta} \hat{G}_i$**

$$\nabla^2_{\omega\theta} G_i(\omega;\theta) = -\nabla_\theta E_{\zeta \sim P_{\pi_{\omega;\theta}}}[\sum_{t=0}^\infty \gamma^t \nabla_\omega c_\omega(s_t, a_t)],$$

$$= -\int_{s_0 \in \mathcal{S}} P_0(s_0) \nabla_\theta \int_{a_0 \in \mathcal{A}} \pi_{\omega;\theta}(a_0|s_0) \cdot \nabla_\omega J^\pi_{c_\omega}(s_0, a_0)|_{\pi=\pi_{\omega;\theta}} da_0 ds_0,$$

$$= -\int_{s_0 \in \mathcal{S}} P_0(s_0) \int_{a_0 \in \mathcal{A}} \Big[ \nabla_\theta \pi_{\omega;\theta}(a_0|s_0) \cdot \nabla_\omega J^\pi_{c_\omega}(s_0, a_0)|_{\pi=\pi_{\omega;\theta}}$$

$$+ \pi_{\omega;\theta}(a_0|s_0) \cdot \nabla_\theta (\nabla_\omega c_\omega(s_0, a_0) + \int_{s_1 \in \mathcal{S}} P(s_1|s_0, a_0) \nabla_\omega J^\pi_{c_\omega}(s_1)|_{\pi=\pi_{\omega;\theta}} ds_1 \Big] da_0 ds_0.$$

Keep the expansion, we can get

$$\nabla^2_{\omega\theta} G_i(\omega;\theta) = -\int_{s \in \mathcal{S}} \frac{\mu^{\pi_{\omega;\theta}}(s)}{1-\gamma} \int_{a \in \mathcal{A}} \pi_{\omega;\theta}(a|s) \nabla_\theta \log \pi_{\omega;\theta}(a|s) (\nabla_\omega J^\pi_{c_\omega}(s,a)|_{\pi=\pi_{\omega;\theta}})^\top da ds,$$

$$= -E_{\zeta \sim P_{\pi_{\omega;\theta}}}\left[\sum_{t=0}^{\infty}\gamma^t \nabla_\theta \log \pi_{\omega;\theta}(a|s)(\nabla_\omega J^\pi_{c_\omega}(s,a)|_{\pi=\pi_{\omega;\theta}})^\top\right],$$

and similarly we can get

$$\nabla^2_{\omega\theta}\hat{G}_i(\omega;\theta,\mathcal{D}) = -E_{\zeta \sim P_{\pi_{\omega;\theta}}}\left[\sum_{t=0}^{\infty}\gamma^t \nabla_\theta \log \pi_{\omega;\theta}(a|s)(\nabla_\omega J^\pi_{c_\omega}(s,a)|_{\pi=\pi_{\omega;\theta}})^\top\right].$$

**Derivation of $\nabla_\theta L_i$ and $\nabla_\theta \hat{L}_i$.**

$$\nabla_\theta L_i(\theta,\omega) = -E_{\zeta \sim P_\pi}[\sum_{t=0}^{\infty}\nabla_\theta \log \pi_{\omega;\theta}(a_t|s_t)],$$

$$= -E_{\zeta \sim P_\pi}\left[\sum_{t=0}^{\infty}\gamma^t\left(E_{\zeta \sim P_{\pi_{\omega;\theta}}}[\sum_{t=0}^{\infty}\gamma^t \nabla_\theta r_\theta(S_t,A_t)|S_0=s_t,A_0=a_t]\right.\right.$$

$$\left.\left. - E_{\zeta \sim P_{\pi_{\omega;\theta}}}[\sum_{t=0}^{\infty}\gamma^t \nabla_\theta r_\theta(S_t,A_t)|S_0=s_t]\right)\right],$$

$$= -E_{\zeta \sim P_\pi}\left[\sum_{t=0}^{\infty}\gamma^t\left(\nabla_\theta r_\theta(s_t,a_t) + E_{\zeta \sim P_{\pi_{\omega;\theta}}}[\sum_{t=1}^{\infty}\gamma^t \nabla_\theta r_\theta(S_t,A_t)|S_0=s_t,A_0=a_t]\right.\right.$$

$$\left.\left. - E_{\zeta \sim P_{\pi_{\omega;\theta}}}[\sum_{t=0}^{\infty}\gamma^t \nabla_\theta r_\theta(S_t,A_t)|S_0=s_t]\right)\right],$$

$$= -E_{\zeta \sim P_\pi}\left[\sum_{t=0}^{\infty}\gamma^t\left(\nabla_\theta r_\theta(s_t,a_t) + \gamma E_{\zeta \sim P_{\pi_{\omega;\theta}}}[\sum_{t=0}^{\infty}\gamma^t \nabla_\theta r_\theta(S_t,A_t)|S_0=s_{t+1}]\right.\right.$$

$$\left.\left. - E_{\zeta \sim P_{\pi_{\omega;\theta}}}[\sum_{t=0}^{\infty}\gamma^t \nabla_\theta r_\theta(S_t,A_t)|S_0=s_t]\right)\right],$$

$$= -E_{\zeta \sim P_\pi}\left[\sum_{t=0}^{\infty}\gamma^t \nabla_\theta r_\theta(s_t,a_t) - E_{\zeta \sim P_{\pi_{\omega;\theta}}}[\sum_{t=0}^{\infty}\gamma^t \nabla_\theta r_\theta(S_t,A_t)|S_0=s_0]\right],$$

$$= E_{\zeta \sim P_{\pi_{\omega;\theta}}}[\sum_{t=0}^{\infty}\gamma^t \nabla_\theta r_\theta(s_t,a_t)] - \nabla_\theta J_{r_\theta}(\pi_i).$$

Similarly, we can get

$$\nabla_\theta \hat{L}_i(\theta,\omega,\mathcal{D}) = E_{\zeta \sim P_{\pi_{\omega;\theta}}}[\sum_{t=0}^{\infty}\gamma^t \nabla_\theta r_\theta(s_t,a_t)] - \frac{1}{D}\sum_{j=1}^{D}\nabla_\theta \hat{J}_{r_\theta}(\zeta^j).$$

**Derivation of $\nabla_\omega L_i$ and $\nabla_\omega \hat{L}_i$.**

The derivation follows the similar steps of the derivation of $\nabla_\theta L_i$ and $\nabla_\theta \hat{L}_i$, thus we omit it.

**Derivation of $\nabla^2_{\omega\omega} L_i$ and $\nabla^2_{\omega\omega}\hat{L}_i$.**

We know that $\nabla_\omega L_i = \nabla_\omega G_i$ and thus $\nabla^2_{\omega\omega}L_i = \nabla^2_{\omega\omega}G_i$.

$\square$

## A.4 THE REDUCTION OF (2)-(3) TO IMAML

From A.3, we can see that $\nabla_\theta G_i(\omega;\theta) = \nabla_\theta L_i(\theta,\omega)$ and $\nabla_\omega G_i(\omega;\theta) = \nabla_\omega L_i(\theta,\omega)$. Therefore, we know that $L_i(\theta,\omega) = G_i(\omega;\theta) + c$ where $c$ is a constant, and thus $G_i$ can also serve as a negative

log-likelihood function. When $\alpha = 0$, problem (2)-(3) becomes:

$$\min_{\theta,\omega} \frac{1}{m} \sum_{i=1}^{m} L_i(\theta, \eta_i^*(\theta, \omega)), \quad \text{s.t. } \eta_i^*(\theta, \omega) = \arg\min_{\eta} G_i(\eta; \theta) + \frac{\lambda}{2} ||\eta - \omega||^2,$$

which is equivalent to

$$\min_{\theta,\omega} \frac{1}{m} \sum_{i=1}^{m} L_i(\theta, \eta_i^*(\theta, \omega)), \quad \text{s.t. } \eta_i^*(\theta, \omega) = \arg\min_{\eta} L_i(\theta; \eta) + \frac{\lambda}{2} ||\eta - \omega||^2.$$

If we ignore the reward parameter $\theta$, this is the standard formulation of iMAML (Rajeswaran et al., 2019) for the cost parameter $\omega$.

## A.5 THE DERIVATION OF THE HYPER-GRADIENTS

From implicit function theorem, we can get that

$$\nabla_\theta \eta_i^*(\theta, \omega) = -[\nabla_{\eta\eta}^2 G_i(\eta_i^*(\theta, \omega); \theta) + \lambda I]^{-1} \nabla_{\eta\theta}^2 G_i(\eta_i^*(\theta, \omega); \theta),$$
$$\nabla_\omega \eta_i^*(\theta, \omega) = \lambda[\nabla_{\eta\eta}^2 G_i(\eta_i^*(\theta, \omega); \theta) + \lambda I]^{-1}.$$

Therefore using the chain rule, we can derive the partial gradients:

$$\frac{\partial}{\partial \theta} L_i(\theta, \eta_i^*(\theta, \omega)) = \nabla_\theta L_i(\theta, \eta_i^*(\theta, \omega)) + [(\nabla_\eta L_i(\theta, \eta_i^*(\theta, \omega)))^\top \nabla_\theta \eta_i^*(\theta, \omega)]^\top,$$
$$= \nabla_\theta L_i(\theta, \eta_i^*(\theta, \omega)) - \nabla_{\theta\eta}^2 G_i(\eta_i^*(\theta, \omega); \theta)[\nabla_{\eta\eta}^2 G_i(\eta_i^*(\theta, \omega); \theta) + \lambda I]^{-1} \nabla_\eta L_i(\theta, \eta_i^*(\theta, \omega)),$$
$$\frac{\partial}{\partial \omega} L_i(\theta, \eta_i^*(\theta, \omega)) = [(\nabla_\eta L_i(\theta, \eta_i^*(\theta, \omega)))^\top \nabla_\omega \eta_i^*(\theta, \omega)]^\top,$$
$$= \lambda[\nabla_{\eta\eta}^2 G_i(\eta_i^*(\theta, \omega); \theta) + \lambda I]^{-1} \nabla_\eta L_i(\theta, \eta_i^*(\theta, \omega)).$$

With the partial gradients, we can derive the hyper-gradients:

$$\frac{\partial L_i(\varphi_i, \eta_i^*(\varphi_i, \omega))}{\partial \theta} = \frac{\partial \varphi_i}{\partial \theta} \cdot \frac{\partial}{\partial \varphi} L_i(\varphi_i, \eta_i^*(\varphi_i, \omega)),$$

$$= \left[ I - \alpha \frac{\partial^2}{\partial \theta^2} L_i(\theta, \eta_i^*(\theta, \omega)) \right] \cdot \frac{\partial}{\partial \varphi_i} L_i(\varphi_i, \eta_i^*(\varphi_i, \omega)),$$

$$= \left[ I - \alpha \frac{\partial^2}{\partial \theta^2} L_i(\theta, \eta_i^*(\theta, \omega)) \right] \cdot \left[ \nabla_\varphi L_i(\varphi_i, \eta_i^*(\varphi_i, \omega)) \right.$$

$$\left. - \nabla_{\theta\eta}^2 G_i(\eta_i^*(\varphi_i, \omega); \varphi_i)[\nabla_{\eta\eta}^2 G_i(\eta_i^*(\varphi_i, \omega); \varphi_i) + \lambda I]^{-1} \nabla_\eta L_i(\varphi_i, \eta_i^*(\varphi_i, \omega)) \right],$$

$$\frac{\partial L_i(\varphi_i, \eta_i^*(\varphi_i, \omega))}{\partial \omega} = [(\frac{\partial L_i(\varphi_i, \eta_i^*(\varphi_i, \omega))}{\partial \varphi})^\top \frac{\partial \varphi_i}{\partial \omega}]^\top + [(\frac{\partial L_i(\varphi_i, \eta_i^*(\varphi_i, \omega))}{\partial \eta})^\top \frac{\partial \eta_i^*(\varphi_i, \omega)}{\partial \omega}]^\top,$$

$$= -\alpha \frac{\partial^2}{\partial \omega \partial \theta} L_i(\theta, \eta_i^*(\theta, \omega)) \cdot \frac{\partial L_i(\varphi_i, \eta_i^*(\varphi_i, \omega))}{\partial \varphi}$$

$$+ \lambda[\nabla_{\eta\eta}^2 G_i(\eta_i^*(\theta, \omega); \theta) + \lambda I]^{-1} \nabla_\eta L_i(\theta, \eta_i^*(\theta, \omega)).$$

## A.6 THE SMOOTHNESS OF $G_i$ AND $\hat{G}_i$

From Lemma A.3, we know that

$$\nabla_{\omega\omega}^2 G_i(\omega; \theta) = \nabla_{\omega\omega}^2 \hat{G}_i(\omega; \theta, \mathcal{D}) = -E_{\zeta \sim P_{\pi_{\omega;\theta}}} \left[ \sum_{t=0}^{\infty} \gamma^t \left( \text{Cov}_\omega(s_t) + \nabla_{\omega\omega}^2 c_\omega(s_t, a_t) \right) \right],$$

$$\text{Cov}_\omega(s_t) = \int_{a \in \mathcal{A}} \pi_{\omega;\theta}(a|s) \left[ \nabla_\omega J_{c_\omega}^\pi(s, a)|_{\pi=\pi_{\omega;\theta}} - \nabla_\omega J_{c_\omega}^\pi(s)|_{\pi=\pi_{\omega;\theta}} \right] (\nabla_\omega J_{c_\omega}^\pi(s, a)|_{\pi=\pi_{\omega;\theta}})^\top da.$$

Assumption 1 (ii) assumes that $||\nabla_\omega c_\omega(s,a)|| \leq \bar{C}_c$ and $||\nabla^2_{\omega\omega} c_\omega(s,a)|| \leq \tilde{C}_c$ for any $(s,a) \in \mathcal{S} \times \mathcal{A}$. Therefore, we have that

$$||\nabla_\omega J^\pi_{c_\omega}(s,a)|_{\pi=\pi_{\omega;\theta}}|| \leq \frac{\bar{C}_c}{1-\gamma}, \qquad ||\nabla_\omega J^\pi_{c_\omega}(s,a)|_{\pi=\pi_{\omega;\theta}}|| \leq \frac{\tilde{C}_c}{1-\gamma},$$

$$||E_{\zeta \sim P_{\pi_{\omega;\theta}}}[\sum_{t=0}^\infty \gamma^t \nabla^2_{\omega\omega} c_\omega(s_t,a_t)]|| \leq \frac{\tilde{C}_c}{1-\gamma}.$$

Then, we know that

$$||\nabla^2_{\omega\omega} G_i(\omega;\theta)|| = ||\nabla^2_{\omega\omega} \hat{G}_i(\omega;\theta,\mathcal{D})|| \leq \frac{2\bar{C}_c}{1-\gamma} \cdot \frac{2\bar{C}_c}{1-\gamma} + \frac{2\tilde{C}_c}{1-\gamma} \triangleq C_{\nabla^2_{\eta\eta} G}.$$

Similarly, we can prove that there exists constants $C_{\nabla^2_{\eta\theta} G}$ and $C_{\nabla^2_{\theta\theta} G}$ such that $||\nabla^2_{\omega\theta} G_i(\omega;\theta)|| = ||\nabla^2_{\omega\theta} \hat{G}_i(\omega;\theta,\mathcal{D})|| \leq C_{\nabla^2_{\eta\theta} G}$ and $||\nabla^2_{\theta\theta} G_i(\omega;\theta)|| = ||\nabla^2_{\theta\theta} \hat{G}_i(\omega;\theta,\mathcal{D})|| \leq C_{\nabla^2_{\theta\theta} G}$.

From Appendix A.3, we can see that $\nabla_\omega \hat{L}_i = \nabla_\omega \hat{G}_i$ and $\nabla_\theta \hat{L}_i = \nabla_\theta \hat{G}_i$. Therefore, the second-order terms of $\hat{L}_i$ can also be bounded by these constants.

**Lemma 5.** *There are positive constant* $C_{\nabla^3_{\eta\eta\eta} G_i}$, $C_{\nabla^3_{\eta\eta\theta} G_i}$, *and* $C_{\nabla^3_{\eta\theta\theta} G_i}$ *such that* $||\nabla^3_{\eta\eta\eta} G_i(\eta;\theta)|| = ||\nabla^3_{\eta\eta\eta} \hat{G}_i(\eta;\theta,\mathcal{D})|| \leq C_{\nabla^3_{\eta\eta\eta} G}$, $||\nabla^3_{\eta\eta\theta} G_i(\eta;\theta)|| = ||\nabla^3_{\eta\eta\theta} \hat{G}_i(\eta;\theta,\mathcal{D})|| \leq C_{\nabla^3_{\eta\eta\theta} G}$, *and* $||\nabla^3_{\eta\theta\theta} G_i(\eta;\theta)|| = ||\nabla^3_{\eta\theta\theta} \hat{G}_i(\eta;\theta,\mathcal{D})|| \leq C_{\nabla^3_{\eta\theta\theta} G}$ *for any* $(\eta,\theta)$ *and any task* $\mathcal{T}_i$.

*Proof.* From Lemma A.3, we know that

$$\nabla^2_{\omega\omega} G_i(\omega;\theta) = \nabla^2_{\omega\omega} \hat{G}_i(\omega;\theta,\mathcal{D}) = -E_{\zeta \sim P_{\pi_{\omega;\theta}}} \left[ \sum_{t=0}^\infty \gamma^t \left( \text{Cov}_\omega(s_t) + \nabla^2_{\omega\omega} c_\omega(s_t,a_t) \right) \right],$$

$$\text{Cov}_\omega(s_t) = \int_{a \in \mathcal{A}} \pi_{\omega;\theta}(a|s) \left[ \nabla_\omega J^\pi_{c_\omega}(s,a)|_{\pi=\pi_{\omega;\theta}} - \nabla_\omega J^\pi_{c_\omega}(s)|_{\pi=\pi_{\omega;\theta}} \right] (\nabla_\omega J^\pi_{c_\omega}(s,a)|_{\pi=\pi_{\omega;\theta}})^\top da.$$

Now we take a look at the term $\text{Cov}_\omega(s_t) + \nabla^2_{\omega\omega} c_\omega(s_t,a_t)$.

$$||\text{Cov}_\omega(s_t) + \nabla^2_{\omega\omega} c_\omega(s_t,a_t)|| \leq ||\text{Cov}_\omega(s_t)|| + ||\nabla^2_{\omega\omega} c_\omega(s_t,a_t)||,$$

$$\leq ||\nabla_\omega J^\pi_{c_\omega}(s,a)|_{\pi=\pi_{\omega;\theta}} - \nabla_\omega J^\pi_{c_\omega}(s)|_{\pi=\pi_{\omega;\theta}}|| \cdot ||\nabla_\omega J^\pi_{c_\omega}(s,a)|_{\pi=\pi_{\omega;\theta}}|| + ||\nabla^2_{\omega\omega} c_\omega(s_t,a_t)||,$$

$$\leq \frac{2\bar{C}_c}{1-\gamma} \cdot \frac{\bar{C}_c}{1-\gamma} + \tilde{C}_c, \tag{15}$$

$$\nabla_\omega \text{Cov}_\omega(s_t)$$

$$= \nabla_\omega \int_{a \in \mathcal{A}} \pi_{\omega;\theta}(a|s) \left[ \nabla_\omega J^\pi_{c_\omega}(s,a)|_{\pi=\pi_{\omega;\theta}} - \nabla_\omega J^\pi_{c_\omega}(s)|_{\pi=\pi_{\omega;\theta}} \right] (\nabla_\omega J^\pi_{c_\omega}(s,a)|_{\pi=\pi_{\omega;\theta}})^\top da,$$

$$\leq \int_{a \in \mathcal{A}} \nabla_\omega \pi_{\omega;\theta}(a|s) \cdot \left[ \nabla_\omega J^\pi_{c_\omega}(s,a)|_{\pi=\pi_{\omega;\theta}} - \nabla_\omega J^\pi_{c_\omega}(s)|_{\pi=\pi_{\omega;\theta}} \right] (\nabla_\omega J^\pi_{c_\omega}(s,a)|_{\pi=\pi_{\omega;\theta}})^\top da \tag{16}$$

$$+ \int_{a \in \mathcal{A}} \pi_{\omega;\theta}(a|s) \nabla_\omega \left[ \nabla_\omega J^\pi_{c_\omega}(s,a)|_{\pi=\pi_{\omega;\theta}} - \nabla_\omega J^\pi_{c_\omega}(s)|_{\pi=\pi_{\omega;\theta}} \right] \cdot (\nabla_\omega J^\pi_{c_\omega}(s,a)|_{\pi=\pi_{\omega;\theta}})^\top da \tag{17}$$

$$+ \int_{a \in \mathcal{A}} \pi_{\omega;\theta}(a|s) \left[ \nabla_\omega J^\pi_{c_\omega}(s,a)|_{\pi=\pi_{\omega;\theta}} - \nabla_\omega J^\pi_{c_\omega}(s)|_{\pi=\pi_{\omega;\theta}} \right] \nabla_\omega (\nabla_\omega J^\pi_{c_\omega}(s,a)|_{\pi=\pi_{\omega;\theta}})^\top da. \tag{18}$$

Now, we bound each term (16)-(18).

First, we bound the term (16)

$$\int_{a \in \mathcal{A}} \nabla_\omega \pi_{\omega;\theta}(a|s) \cdot \left[ \nabla_\omega J^\pi_{c_\omega}(s,a)|_{\pi=\pi_{\omega;\theta}} - \nabla_\omega J^\pi_{c_\omega}(s)|_{\pi=\pi_{\omega;\theta}} \right] (\nabla_\omega J^\pi_{c_\omega}(s,a)|_{\pi=\pi_{\omega;\theta}})^\top da,$$

$$= \int_{a \in \mathcal{A}} \pi_{\omega;\theta}(a|s) \nabla_\omega \log \pi_{\omega;\theta}(a|s) \cdot$$
$$\left[ \nabla_\omega J_{c_\omega}^\pi(s,a)|_{\pi=\pi_{\omega;\theta}} - \nabla_\omega J_{c_\omega}^\pi(s)|_{\pi=\pi_{\omega;\theta}} \right] (\nabla_\omega J_{c_\omega}^\pi(s,a)|_{\pi=\pi_{\omega;\theta}})^\top da,$$
$$\stackrel{(a)}{=} \int_{a \in \mathcal{A}} \pi_{\omega;\theta}(a|s) \left[ E_{\zeta \sim P_{\pi_{\omega;\theta}}}[\sum_{t=0}^\infty \gamma^t \nabla_\omega c_\omega(s_t, a_t)|s_0 = s] \right.$$
$$\left. - E_{\zeta \sim P_{\pi_{\omega;\theta}}}[\sum_{t=0}^\infty \gamma^t \nabla_\omega c_\omega(s_t, a_t)|s_0 = s, a_0 = a] \right] \cdot$$
$$\left[ \nabla_\omega J_{c_\omega}^\pi(s,a)|_{\pi=\pi_{\omega;\theta}} - \nabla_\omega J_{c_\omega}^\pi(s)|_{\pi=\pi_{\omega;\theta}} \right] (\nabla_\omega J_{c_\omega}^\pi(s,a)|_{\pi=\pi_{\omega;\theta}})^\top da,$$

where $(a)$ follows Lemma 3. Therefore, we can see that

$$|| \int_{a \in \mathcal{A}} \nabla_\omega \pi_{\omega;\theta}(a|s) \cdot \left[ \nabla_\omega J_{c_\omega}^\pi(s,a)|_{\pi=\pi_{\omega;\theta}} - \nabla_\omega J_{c_\omega}^\pi(s)|_{\pi=\pi_{\omega;\theta}} \right] (\nabla_\omega J_{c_\omega}^\pi(s,a)|_{\pi=\pi_{\omega;\theta}})^\top da ||,$$
$$\leq \frac{2\bar{C}_c}{1-\gamma} \cdot \frac{2\bar{C}_c}{1-\gamma} \cdot \frac{\bar{C}_c}{1-\gamma}. \tag{19}$$

Second, we bound the term (17)

$$\nabla_\omega J_{c_\omega}^\pi(s,a)|_{\pi=\pi_{\omega;\theta}} = \nabla_\omega E_{\zeta \sim P_{\pi_{\omega;\theta}}}[\sum_{t=0}^\infty \gamma^t \nabla_\omega c_\omega(s_t, a_t)] = -\nabla_{\omega\omega}^2 G_i(\omega;\theta). \tag{20}$$

Therefore, $||\nabla_\omega J_{c_\omega}^\pi(s,a)|_{\pi=\pi_{\omega;\theta}}|| = ||\nabla_{\omega\omega}^2 G_i(\omega;\theta)|| \stackrel{(b)}{\leq} C_{\nabla_{\eta\eta}^2 G}$ where $(b)$ follows A.6. Similarly, we can see that $||\nabla_\omega J_{c_\omega}^\pi(s)|_{\pi=\pi_{\omega;\theta}}|| \leq C_{\nabla_{\eta\eta}^2 G}$. Then we have

$$|| \int_{a \in \mathcal{A}} \pi_{\omega;\theta}(a|s) \nabla_\omega \left[ \nabla_\omega J_{c_\omega}^\pi(s,a)|_{\pi=\pi_{\omega;\theta}} - \nabla_\omega J_{c_\omega}^\pi(s)|_{\pi=\pi_{\omega;\theta}} \right] \cdot (\nabla_\omega J_{c_\omega}^\pi(s,a)|_{\pi=\pi_{\omega;\theta}})^\top da ||,$$
$$\leq 2C_{\nabla_{\eta\eta}^2} G \cdot C_{\nabla_{\eta\eta}^2} G. \tag{21}$$

Third, we bound the term (18)

$$|| \int_{a \in \mathcal{A}} \pi_{\omega;\theta}(a|s) \left[ \nabla_\omega J_{c_\omega}^\pi(s,a)|_{\pi=\pi_{\omega;\theta}} - \nabla_\omega J_{c_\omega}^\pi(s)|_{\pi=\pi_{\omega;\theta}} \right] \nabla_\omega (\nabla_\omega J_{c_\omega}^\pi(s,a)|_{\pi=\pi_{\omega;\theta}})^\top da ||,$$
$$\leq \frac{2\bar{C}_c}{1-\gamma} \cdot C_{\nabla_{\eta\eta}^2 G}. \tag{22}$$

Therefore, we can see that

$$||\nabla_\omega \text{Cov}_\omega(s_t)|| \stackrel{(c)}{\leq} \frac{2\bar{C}_c}{1-\gamma} \cdot \frac{2\bar{C}_c}{1-\gamma} \cdot \frac{\bar{C}_c}{1-\gamma} + 2C_{\nabla_{\eta\eta}^2} G \cdot C_{\nabla_{\eta\eta}^2} G + \frac{2\bar{C}_c}{1-\gamma} \cdot C_{\nabla_{\eta\eta}^2 G}, \tag{23}$$

where $(c)$ follows (19)-(22).

Define $\text{Cov}_{\omega\omega}(s,a) \triangleq E_{\zeta \sim P_{\pi_{\omega;\theta}}}[\sum_{t=0}^\infty \gamma^t(\text{Cov}_\omega(s_t) + \nabla_{\omega\omega}^2 c_\omega(s_t, a_t))|s_0 = s, a_0 = a]$ and $\text{Cov}_{\omega\omega}(s) \triangleq E_{\zeta \sim P_{\pi_{\omega;\theta}}}[\sum_{t=0}^\infty \gamma^t(\text{Cov}_\omega(s_t) + \nabla_{\omega\omega}^2 c_\omega(s_t, a_t))|s_0 = s]$. We can see that

$$||\text{Cov}_{\omega\omega}(s,a)|| \stackrel{(d)}{\leq} \frac{1}{1-\gamma} \left( \frac{2\bar{C}_c^2}{(1-\gamma)^2} + \tilde{C}_c \right), \tag{24}$$

where $(d)$ follows (15).

Therefore,

$$\nabla_{\omega\omega}^2 G_i(\omega;\theta) = - \int_{s_0 \in \mathcal{S}} P_0(s_0) \nabla_\omega \int_{a_0 \in \mathcal{A}} \pi_{\omega;\theta}(a_0|s_0) \text{Cov}_{\omega\omega}(s_0, a_0) da_0 ds_0,$$

$$= -\int_{s_0 \in \mathcal{S}} P_0(s_0) \int_{a_0 \in \mathcal{A}} \Bigg[ \nabla_\omega \pi_{\omega;\theta}(a_0|s_0) \cdot \mathrm{Cov}_{\omega\omega}(s_0, a_0)$$

$$+ \pi_{\omega;\theta}(a_0|s_0) \nabla_\omega \mathrm{Cov}_{\omega\omega}(s_0, a_0) \Bigg] da_0 ds_0,$$

$$= -\int_{s_0 \in \mathcal{S}} P_0(s_0) \int_{a_0 \in \mathcal{A}} \Bigg[ \nabla_\omega \pi_{\omega;\theta}(a_0|s_0) \cdot \mathrm{Cov}_{\omega\omega}(s_0, a_0) + \pi_{\omega;\theta}(a_0|s_0) \cdot$$

$$\nabla_\omega \bigg( \mathrm{Cov}_\omega(s_0, a_0) + \nabla^2_{\omega\omega} c_\omega(s_0, a_0) + \gamma \int_{s_1 \in \mathcal{S}} P(s_1|s_0, a_0) \mathrm{Cov}_{\omega\omega}(s_1) ds_1 \bigg) \Bigg] da_0 ds_0.$$

Keep the expansion, we can see that

$$\nabla^2_{\omega\omega} G_i(\omega;\theta) = E_{\zeta \sim P_{\pi_{\omega;\theta}}} \Bigg[ \sum_{t=0}^\infty \gamma^t \bigg( \nabla_\omega \log \pi_{\omega;\theta}(a_t|s_t) \cdot \mathrm{Cov}_{\omega\omega}(s_t, a_t)$$

$$+ \nabla_\omega \mathrm{Cov}_\omega(s_t, a_t) + \nabla^3_{\omega\omega\omega} c_\omega(s_t, a_t) \bigg) \Bigg],$$

$$\overset{(e)}{\leq} \frac{1}{1-\gamma} \cdot \frac{2\bar{C}_c}{1-\gamma} \cdot \frac{1}{1-\gamma} \bigg( \frac{2\bar{C}_c^2}{(1-\gamma)^2} + \tilde{C}_c \bigg) + E_{\zeta \in \pi_{\omega;\theta}} \Big[ \nabla_\omega \mathrm{Cov}_\omega(s_t, a_t) + \nabla^3_{\omega\omega\omega} c_\omega(s_t, a_t) \Big],$$

$$\overset{(f)}{\leq} \frac{1}{1-\gamma} \cdot \frac{2\bar{C}_c}{1-\gamma} \cdot \frac{1}{1-\gamma} \bigg( \frac{2\bar{C}_c^2}{(1-\gamma)^2} + \tilde{C}_c \bigg)$$

$$+ \frac{2\bar{C}_c}{1-\gamma} \cdot \frac{2\bar{C}_c}{1-\gamma} \cdot \frac{\bar{C}_c}{1-\gamma} + 2C_{\nabla^2_{\eta\eta} G} \cdot C_{\nabla^2_{\eta\eta} G} + \frac{2\bar{C}_c}{1-\gamma} \cdot C_{\nabla^2_{\eta\eta} G} + E_{\zeta \in \pi_{\omega;\theta}} \Big[ \nabla^3_{\omega\omega\omega} c_\omega(s_t, a_t) \Big],$$

where $(e)$ follows Lemma 3 and (24) and $(f)$ follows (23). Assumption 1 (iii) assumes that $||\nabla^3_{\omega\omega\omega} c_\omega(s, a)||$ is bounded for any $(s, a) \in \mathcal{S}, \mathcal{A}$, therefore, there is a positive constant $C_{\nabla^3_{\eta\eta\eta} G_i}$ such that $||\nabla^3_{\eta\eta\eta} G(\eta;\theta)|| \leq C_{\nabla^3_{\eta\eta\eta} G}$ for any $(\eta, \theta)$ and any task $\mathcal{T}_i$. We can also get $||\nabla^3_{\eta\eta\eta} \hat{G}_i(\eta;\theta, \mathcal{D})|| \leq C_{\nabla^3_{\eta\eta\eta} G}$.

Similarly, we can prove the existence of $C_{\nabla^3_{\eta\eta\theta} G}$ and $C_{\nabla^3_{\eta\theta\theta} G}$.

**Lemma 6.** *There are positive constants $C_{\nabla^4_{\eta\eta\eta\eta} G}$, $C_{\nabla^4_{\eta\eta\eta\theta} G}$, $C_{\nabla^4_{\eta\eta\theta\theta} G}$, $C_{\nabla^4_{\eta\theta\theta\theta} G}$, and $C_{\nabla^4_{\theta\theta\theta\theta} G}$ such that $||\nabla^4_{\eta\eta\eta\eta} G_i(\eta;\theta)|| = ||\nabla^4_{\eta\eta\eta\eta} \hat{G}_i(\eta;\theta, \mathcal{D})|| \leq C_{\nabla^4_{\eta\eta\eta\eta} G}$, $||\nabla^4_{\eta\eta\eta\theta} G_i(\eta;\theta)|| = ||\nabla^4_{\eta\eta\eta\theta} \hat{G}_i(\eta;\theta, \mathcal{D})|| \leq C_{\nabla^4_{\eta\eta\eta\theta} G}$, $||\nabla^4_{\eta\eta\theta\theta} G_i(\eta;\theta)|| = ||\nabla^4_{\eta\eta\theta\theta} \hat{G}_i(\eta;\theta, \mathcal{D})|| \leq C_{\nabla^4_{\eta\eta\theta\theta} G}$, $||\nabla^4_{\eta\theta\theta\theta} G_i(\eta;\theta)|| = ||\nabla^4_{\eta\theta\theta\theta} \hat{G}_i(\eta;\theta, \mathcal{D})|| \leq C_{\nabla^4_{\eta\theta\theta\theta} G}$, and $||\nabla^4_{\theta\theta\theta\theta} G_i(\eta;\theta)|| = ||\nabla^4_{\theta\theta\theta\theta} \hat{G}_i(\eta;\theta, \mathcal{D})|| \leq C_{\nabla^4_{\theta\theta\theta\theta} G}$.*

*Proof.* We can derive the constants following the proof idea of Lemma 5 and thus we omit the proof. □

### A.7 PROOF OF LEMMA 1

Following the standard results for $(\lambda - C_{\nabla^2_{\eta\eta} G})$-strongly convex and $(\lambda + C_{\nabla^2_{\eta\eta} G})$-smooth objective functions (Nesterov, 2003; Boyd & Vandenberghe, 2004), we know that

$$||x(\bar{K}) - [\lambda I + \nabla^2_{\eta\eta} \hat{G}_i(\hat{\eta}_i(\hat{\varphi}_i, \omega, \mathcal{D}_i^{\mathrm{eval}}, K); \hat{\varphi}_i, \mathcal{D}_i^{\mathrm{eval}})]^{-1} \nabla_\eta \hat{L}_i(\hat{\varphi}_i, \hat{\eta}_i(\hat{\varphi}_i, \omega, \mathcal{D}_i^{\mathrm{eval}}, K), \mathcal{D}_i^{\mathrm{eval}})||,$$

$$\leq O\bigg( \big( \frac{C_{\nabla^2_{\eta\eta} G}}{\lambda} \big)^{\bar{K}} \bigg). \tag{25}$$

Define $\mathcal{H}(\eta;\varphi_i) \triangleq \lambda I + \nabla^2_{\eta\eta} \hat{G}_i(\eta;\varphi_i, \mathcal{D}_i^{\mathrm{eval}})$, therefore $\lambda - C_{\nabla^2_{\eta\eta} G} \leq ||\mathcal{H}(\eta;\varphi_i)|| \leq \lambda + C_{\nabla^2_{\eta\eta} G}$ and $||\nabla_\eta \mathcal{H}(\eta;\varphi_i)|| = ||\nabla^3_{\eta\eta\eta} \hat{G}_i(\eta;\varphi_i, \mathcal{D}_i^{\mathrm{eval}})|| \overset{(a)}{\leq} C_{\nabla^3_{\eta\eta\eta} G}$ where $(a)$ follows Lemma 5. Therefore,

$$||[\mathcal{H}(\eta_1, \varphi_i)]^{-1} - [\mathcal{H}(\eta_2, \varphi_i)]^{-1}||,$$

$$= ||[\mathcal{H}(\eta_2; \varphi_i)]^{-1} \{\mathcal{H}(\eta_2; \varphi_i) - \mathcal{H}(\eta_1; \varphi_i)\} [\mathcal{H}(\eta_1; \varphi_i)]^{-1}||,$$

$$\leq ||[\mathcal{H}(\eta_2; \varphi_i)]^{-1}|| \cdot ||[\mathcal{H}(\eta_1; \varphi_i)]^{-1}|| \cdot ||\mathcal{H}(\eta_2; \varphi_i) - \mathcal{H}(\eta_1; \varphi_i)||,$$

$$\leq \frac{C_{\nabla^3_{\eta\eta\eta}G}}{(\lambda - C_{\nabla^2_{\eta\eta}G})^2}||\eta_1 - \eta_2||. \tag{26}$$

We know that $||\nabla_{\eta\eta}\hat{L}_i(\theta, \eta, \mathcal{D})|| \overset{(b)}{=} ||\nabla^2_{\eta\eta}\hat{G}_i(\eta; \theta, \mathcal{D})|| \overset{(c)}{\leq} C_{\nabla^2_{\eta\eta}G}$ where $(b)$ follows A.3 and $(c)$ follows A.6, and $||\nabla_\eta \hat{L}_i(\theta, \eta, \mathcal{D})|| \overset{(d)}{\leq} \frac{\bar{C}_c}{1-\gamma}$ where $(d)$ follows A.3. Therefore we have the following

$$||[\mathcal{H}(\eta_1; \varphi_i)]^{-1}\nabla_\eta\hat{L}_i(\varphi_i, \eta_1, \mathcal{D}) - [\mathcal{H}(\eta_2; \varphi_i)]^{-1}\nabla_\eta\hat{L}_i(\varphi_i, \eta_2, \mathcal{D})||,$$

$$\leq ||[\mathcal{H}(\eta_1; \varphi_i)]^{-1}\nabla_\eta\hat{L}_i(\varphi_i, \eta_1, \mathcal{D}) - [\mathcal{H}(\eta_2; \varphi_i)]^{-1}\nabla_\eta\hat{L}_i(\varphi_i, \eta_1, \mathcal{D})||$$

$$+ ||[\mathcal{H}(\eta_2; \varphi_i)]^{-1}\nabla_\eta\hat{L}_i(\varphi_i, \eta_1, \mathcal{D}) - [\mathcal{H}(\eta_2; \varphi_i)]^{-1}\nabla_\eta\hat{L}_i(\varphi_i, \eta_2, \mathcal{D})||,$$

$$\leq ||[\mathcal{H}(\eta_1; \varphi_i)]^{-1} - [\mathcal{H}(\eta_2; \varphi_i)]^{-1}|| \cdot ||\nabla_\eta\hat{L}_i(\varphi_i, \eta_1, \mathcal{D})||$$

$$+ ||\nabla_\eta\hat{L}_i(\varphi_i, \eta_1, \mathcal{D}) - \nabla_\eta\hat{L}_i(\varphi_i, \eta_2, \mathcal{D})|| \cdot ||[\mathcal{H}(\hat{\eta}_i^*; \varphi_i)]^{-1}||,$$

$$\overset{(e)}{\leq} \frac{C_{\nabla^3_{\eta\eta\eta}G}\bar{C}_c}{(1-\gamma)(\lambda - C_{\nabla^2_{\eta\eta}G})^2}||\eta_1 - \eta_2|| + \frac{C_{\nabla^2_{\eta\eta}G}}{\lambda - C_{\nabla^2_{\eta\eta}G}}||\eta_1 - \eta_2||, \tag{27}$$

where $(e)$ follows (26). Therefore, we have that

$$||x(\bar{K}) - [\mathcal{H}(\hat{\eta}_i^*(\hat{\varphi}_i, \omega, \mathcal{D}_i^{\text{eval}}); \hat{\varphi}_i)]^{-1}\nabla_\eta\hat{L}_i(\hat{\varphi}_i, \hat{\eta}_i^*(\hat{\varphi}_i, \omega, \mathcal{D}_i^{\text{eval}}), \mathcal{D}_i^{\text{eval}})||,$$

$$\leq ||x(\bar{K}) - [\mathcal{H}(\hat{\eta}_i(\hat{\varphi}_i, \omega, \mathcal{D}_i^{\text{eval}}, K); \hat{\varphi}_i)]^{-1}\nabla_\eta\hat{L}_i(\hat{\varphi}_i, \hat{\eta}_i(\hat{\varphi}_i, \omega, \mathcal{D}_i^{\text{eval}}, K), \mathcal{D}_i^{\text{eval}})||$$

$$+ ||[\mathcal{H}(\hat{\eta}_i(\hat{\varphi}_i, \omega, \mathcal{D}_i^{\text{eval}}, K); \hat{\varphi}_i)]^{-1}\nabla_\eta\hat{L}_i(\hat{\varphi}_i, \hat{\eta}_i(\hat{\varphi}_i, \omega, \mathcal{D}_i^{\text{eval}}, K), \mathcal{D}_i^{\text{eval}})$$

$$- [\mathcal{H}(\hat{\eta}_i^*(\hat{\varphi}_i, \omega, \mathcal{D}_i^{\text{eval}}); \hat{\varphi}_i)]^{-1}\nabla_\eta\hat{L}_i(\hat{\varphi}_i, \hat{\eta}_i^*(\hat{\varphi}_i, \omega, \mathcal{D}_i^{\text{eval}}), \mathcal{D}_i^{\text{eval}})||,$$

$$\overset{(f)}{\leq} O\left( (\frac{C_{\nabla^2_{\eta\eta}G}}{\lambda})^{\bar{K}} + (\frac{C_{\nabla^2_{\eta\eta}G}}{\lambda})^K \right), \tag{28}$$

where $(f)$ follows (25) and (27).

From A.3, we know that

$$\nabla^2_{\omega\theta}G_i(\omega; \theta) = \nabla^2_{\omega\theta}\hat{G}_i(\omega; \theta, \mathcal{D}) = -E_{\zeta \sim P_{\pi_{\omega;\theta}}}\left[ \sum_{t=0}^\infty \gamma^t \nabla_\theta \log \pi_{\omega;\theta}(a|s)(\nabla_\omega J^\pi_{c_\omega}(s, a)|_{\pi=\pi_{\omega;\theta}})^\top \right].$$

Therefore, $||\nabla^2_{\omega\theta}\hat{G}_i(\omega; \theta, \mathcal{D})|| \overset{(g)}{\leq} \frac{1}{1-\gamma} \cdot \frac{2\bar{C}_r}{1-\gamma} \cdot C_{\nabla^2_{\eta\eta}G}$ where $(g)$ follows Lemma 3 and (20). $\qquad \square$

Then we have that

$$||\hat{\Delta}_{\theta,i} - \Delta_{\theta,i}|| \leq ||\nabla_\theta\hat{L}_i(\theta, \hat{\eta}_i(\theta, \omega, \mathcal{D}, K), \mathcal{D}) - \nabla_\theta\hat{L}_i(\theta, \hat{\eta}_i^*(\theta, \omega, \mathcal{D}), \mathcal{D})||$$

$$+ ||\nabla^2_{\theta\eta}\hat{G}_i(\hat{\eta}_i(\theta, \omega, \mathcal{D}, K); \theta, \mathcal{D})x(\bar{K})$$

$$- \nabla^2_{\theta\eta}\hat{G}_i(\hat{\eta}_i^*(\theta, \omega, \mathcal{D}); \theta, \mathcal{D})[\mathcal{H}(\hat{\eta}_i^*(\hat{\varphi}_i, \omega, \mathcal{D}); \hat{\varphi}_i)]^{-1}\nabla_\eta\hat{L}_i(\hat{\varphi}_i, \hat{\eta}_i^*(\hat{\varphi}_i, \omega, \mathcal{D}), \mathcal{D})||,$$

$$\overset{(h)}{\leq} \frac{\bar{C}_r}{1-\gamma}||\hat{\eta}_i(\theta, \omega, \mathcal{D}, K) - \hat{\eta}_i^*(\theta, \omega, \mathcal{D})||$$

$$+ ||\nabla^2_{\theta\eta}\hat{G}_i(\hat{\eta}_i(\theta, \omega, \mathcal{D}, K); \theta, \mathcal{D})x(\bar{K})$$

$$- \nabla^2_{\theta\eta}\hat{G}_i(\hat{\eta}_i(\theta, \omega, \mathcal{D}, K); \theta, \mathcal{D})[\mathcal{H}(\hat{\eta}_i^*(\hat{\varphi}_i, \omega, \mathcal{D}); \hat{\varphi}_i)]^{-1}\nabla_\eta\hat{L}_i(\hat{\varphi}_i, \hat{\eta}_i^*(\hat{\varphi}_i, \omega, \mathcal{D}), \mathcal{D})||$$

$$+ ||\nabla^2_{\theta\eta}\hat{G}_i(\hat{\eta}_i(\theta, \omega, \mathcal{D}, K); \theta, \mathcal{D})[\mathcal{H}(\hat{\eta}_i^*(\hat{\varphi}_i, \omega, \mathcal{D}); \hat{\varphi}_i)]^{-1}\nabla_\eta\hat{L}_i(\hat{\varphi}_i, \hat{\eta}_i^*(\hat{\varphi}_i, \omega, \mathcal{D}), \mathcal{D})$$

$$- \nabla^2_{\theta\eta}\hat{G}_i(\hat{\eta}_i^*(\theta, \omega, \mathcal{D}); \theta, \mathcal{D})[\mathcal{H}(\hat{\eta}_i^*(\hat{\varphi}_i, \omega, \mathcal{D}); \hat{\varphi}_i)]^{-1}\nabla_\eta\hat{L}_i(\hat{\varphi}_i, \hat{\eta}_i^*(\hat{\varphi}_i, \omega, \mathcal{D}), \mathcal{D})||,$$

$$\leq \frac{\bar{C}_r}{1-\gamma}||\hat{\eta}_i(\theta, \omega, \mathcal{D}, K) - \hat{\eta}_i^*(\theta, \omega, \mathcal{D})||$$

$$+ ||\nabla^2_{\theta\eta}\hat{G}_i(\hat{\eta}_i(\theta, \omega, \mathcal{D}, K); \theta, \mathcal{D})|| \cdot ||x(\bar{K}) - [\mathcal{H}(\hat{\eta}_i^*(\hat{\varphi}_i, \omega, \mathcal{D}); \hat{\varphi}_i)]^{-1} \cdot$$

$$\nabla_\eta \hat{L}_i(\hat{\varphi}_i, \hat{\eta}_i^*(\hat{\varphi}_i, \omega, \mathcal{D}), \mathcal{D})|| + ||\nabla_{\theta\eta}^2 \hat{G}_i(\hat{\eta}_i(\theta, \omega, \mathcal{D}, K); \theta, \mathcal{D}) - \nabla_{\theta\eta}^2 \hat{G}_i(\hat{\eta}_i^*(\theta, \omega, \mathcal{D}); \theta, \mathcal{D})||\cdot$$

$$||[\mathcal{H}(\hat{\eta}_i^*(\hat{\varphi}_i, \omega, \mathcal{D}); \hat{\varphi}_i)]^{-1} \nabla_\eta \hat{L}_i(\hat{\varphi}_i, \hat{\eta}_i^*(\hat{\varphi}_i, \omega, \mathcal{D}), \mathcal{D})||,$$

$$\overset{(i)}{\leq} \frac{\bar{C}_r}{1-\gamma}||\hat{\eta}_i(\theta, \omega, \mathcal{D}, K) - \hat{\eta}_i^*(\theta, \omega, \mathcal{D})|| + O\left((\frac{C_{\nabla_{\eta\eta}^2 G}}{\lambda})^{\bar{K}} + (\frac{C_{\nabla_{\eta\eta}^2 G}}{\lambda})^K\right)$$

$$+ ||\nabla_{\theta\eta}^2 \hat{G}_i(\hat{\eta}_i(\theta, \omega, \mathcal{D}, K); \theta, \mathcal{D}) - \nabla_{\theta\eta}^2 \hat{G}_i(\hat{\eta}_i^*(\theta, \omega, \mathcal{D}); \theta, \mathcal{D})||\cdot$$

$$||[\mathcal{H}(\hat{\eta}_i^*(\hat{\varphi}_i, \omega, \mathcal{D}); \hat{\varphi}_i)]^{-1} \nabla_\eta \hat{L}_i(\hat{\varphi}_i, \hat{\eta}_i^*(\hat{\varphi}_i, \omega, \mathcal{D}), \mathcal{D})||,$$

$$\overset{(j)}{\leq} \frac{\bar{C}_r}{1-\gamma}||\hat{\eta}_i(\theta, \omega, \mathcal{D}, K) - \hat{\eta}_i^*(\theta, \omega, \mathcal{D})|| + O\left((\frac{C_{\nabla_{\eta\eta}^2 G}}{\lambda})^{\bar{K}} + (\frac{C_{\nabla_{\eta\eta}^2 G}}{\lambda})^K\right)$$

$$+ C_{\nabla_{\eta\eta\theta}^3 G} \frac{1}{\lambda - C_{\nabla_{\eta\eta}^2 G}} \cdot \frac{\bar{C}_c}{1-\gamma}||\hat{\eta}_i(\theta, \omega, \mathcal{D}, K) - \hat{\eta}_i^*(\theta, \omega, \mathcal{D})||,$$

$$\leq O\left((\frac{C_{\nabla_{\eta\eta}^2 G}}{\lambda})^{\bar{K}} + (\frac{C_{\nabla_{\eta\eta}^2 G}}{\lambda})^K\right).$$

where $(h)$ follows A.3, $(i)$ follows A.6 and (28), and $(j)$ follows Lemma 5.

$$||\hat{\Delta}_{\omega,i} - \Delta_{\omega,i}|| = \lambda||x(\bar{K}) - [\mathcal{H}(\hat{\eta}_i^*(\theta, \omega, \mathcal{D}); \theta)]^{-1} \nabla_\eta \hat{L}_i(\theta, \hat{\eta}_i^*(\theta, \omega, \mathcal{D}), \mathcal{D})||,$$

$$\overset{(k)}{\leq} O\left((\frac{C_{\nabla_{\eta\eta}^2 G}}{\lambda})^{\bar{K}} + (\frac{C_{\nabla_{\eta\eta}^2 G}}{\lambda})^K\right),$$

where $(k)$ follows (28).

## A.8 Proof of Lemma 2

We first provide the approximation error of the first-order approximation and then provide the approximation error of the hyper-gradients.

**Claim 1.** *There are positive constants $\tilde{D}_\theta$, $D_\theta$ and $D_\omega$ such that $||\frac{\partial^2}{\partial\theta^2}L_i(\theta, \eta_i^*(\theta, \omega))|| = ||\frac{\partial^2}{\partial\theta^2}\hat{L}_i(\theta, \eta_i^*(\theta, \omega), \mathcal{D})|| \leq \tilde{D}_\theta$, $||\frac{\partial^3}{\partial\theta^3}L_i(\theta, \eta_i^*(\theta, \omega))|| = ||\frac{\partial^3}{\partial\theta^3}\hat{L}_i(\theta, \eta_i^*(\theta, \omega), \mathcal{D})|| \leq D_\theta$, and $||\frac{\partial^3}{\partial\omega\partial\theta\partial\theta}L_i(\theta, \eta_i^*(\theta, \omega))|| = ||\frac{\partial^3}{\partial\omega\partial\theta\partial\theta}\hat{L}_i(\theta, \eta_i^*(\theta, \omega), \mathcal{D})|| \leq D_\omega$.*

*Proof.* We first show that $||\frac{\partial^2}{\partial\theta^2}L_i(\theta, \eta_i^*(\theta, \omega))||$ is bounded and the boundedness of $||\frac{\partial^3}{\partial\theta^3}L_i(\theta, \eta_i^*(\theta, \omega))||$ follows the similar idea.

$$\frac{\partial^2}{\partial\theta^2}L_i(\theta, \eta_i^*(\theta, \omega)) = \nabla_{\theta\theta}^2 L_i(\theta, \eta_i^*(\theta, \omega)) + 2(\nabla_\theta \eta_i^*(\theta, \omega))^\top \nabla_{\eta\theta}^2 L_i(\theta, \eta_i^*(\theta, \omega))$$

$$+ (\nabla_{\theta\theta}^2 \eta_i^*(\theta, \omega))^\top \nabla_\eta L_i(\theta, \eta_i^*(\theta, \omega)) + (\nabla_\theta \eta_i^*(\theta, \omega))^\top \nabla_{\eta\eta}^2 L_i(\theta, \eta_i^*(\theta, \omega))\nabla_\theta \eta_i^*(\theta, \omega). \quad (29)$$

To show that $||\frac{\partial^2}{\partial\theta^2}L_i(\theta, \eta_i^*(\theta, \omega))||$ is bounded, we need to show that each term in (29) is bounded. From A.3, we know that $\nabla_\omega G_i = \nabla_\omega L_i$ and $\nabla_\theta G_i = \nabla_\theta L_i$, and thus $G_i = L_i + C$ where $C$ is a constant. Therefore, we know that $||\nabla_{\theta\theta}^2 L_i||$, $||\nabla_{\eta\theta}^2 L_i||$, and $||\nabla_{\eta\eta}^2 L_i||$ are bounded from Appendix A.6. Moreover, $||\nabla_\eta L_i|| \leq \frac{\bar{C}_c}{1-\gamma}$.

Then, we only need to show that $||\nabla_\theta \eta^*(\theta, \omega)||$ and $||\nabla_{\theta\theta}^2 \eta^*(\theta, \omega)||$ are bounded next. From A.5, we know that $\nabla_\theta \eta_i^*(\theta, \omega) = -[\nabla_{\eta\eta}^2 G_i(\eta_i^*(\theta, \omega); \theta) + \lambda I]^{-1} \nabla_{\eta\theta}^2 G_i(\eta_i^*(\theta, \omega); \theta)$. Therefore, $||\nabla_\theta \eta_i^*(\theta, \omega)|| \overset{(a)}{\leq} \frac{C_{\nabla_{\eta\theta}^2 G}}{\lambda - C_{\nabla_{\eta\eta}^2 G}}$ where $(a)$ follows A.6.

$$\nabla_{\theta\theta}^2 \eta_i^*(\theta, \omega) = -[\nabla_{\eta\eta}^2 G_i(\eta_i^*(\theta, \omega); \theta) + \lambda I]^{-1}[(\nabla_\theta \eta_i^*(\theta, \omega))^\top \nabla_{\eta\eta\eta}^3 G_i(\eta_i^*(\theta, \omega); \theta)$$

$$+ \nabla_{\eta\eta\theta}^3 G_i(\eta_i^*(\theta, \omega); \theta)] \cdot [\nabla_{\eta\eta}^2 G_i(\eta_i^*(\theta, \omega); \theta) + \lambda I]^{-1} \nabla_{\eta\theta}^2 G_i(\eta_i^*(\theta, \omega); \theta)$$

$$- [\nabla_{\eta\eta}^2 G_i(\eta_i^*(\theta, \omega); \theta) + \lambda I]^{-1} \nabla_{\eta\theta\theta}^3 G_i(\eta_i^*(\theta, \omega); \theta).$$

We can see that $||\nabla^2_{\theta\theta}\eta_i^*(\theta,\omega)||$ is bounded because all its terms are bounded. In specific, A.6 shows that the second-order terms of $G_i$ are bounded and Lemma 5 shows that the third-order terms of $G_i$ are bounded. Therefore, $\tilde{D}_\theta$ exists.

From A.3, we know that $\nabla_\omega G_i = \nabla_\omega L_i$ and $\nabla_\theta G_i = \nabla_\theta L_i$, and thus $G_i = L_i + C$ where $C$ is a constant. Therefore, we know that $||\nabla^3_{\eta\eta\eta}L_i||$, $||\nabla^3_{\eta\eta\theta}L_i||$, and $||\nabla^3_{\eta\theta\theta}L_i||$ are bounded from Lemma 5. Then we can get the boundedness of $||\frac{\partial^3}{\partial\theta^3}L_i(\theta,\eta_i^*(\theta,\omega))||$ once we prove the boundedness of $||\nabla^3_{\theta\theta\theta}\eta_i^*(\theta,\omega)||$.

$$
\nabla^3_{\theta\theta\theta}\eta_i^*(\theta,\omega) = -2[\nabla^2_{\eta\eta}G_i(\eta_i^*(\theta,\omega);\theta) + \lambda I]^{-1}[(\nabla_\theta\eta_i^*(\theta,\omega))^\top\nabla^3_{\eta\eta\eta}G_i(\eta_i^*(\theta,\omega);\theta)
$$
$$
+ \nabla^3_{\eta\eta\theta}G_i(\eta_i^*(\theta,\omega);\theta)]\cdot[\nabla^2_{\eta\eta}G_i(\eta_i^*(\theta,\omega);\theta) + \lambda I]^{-1}[(\nabla_\theta\eta_i^*(\theta,\omega))^\top\nabla^3_{\eta\eta\eta}G_i(\eta_i^*(\theta,\omega);\theta)
$$
$$
+ \nabla^3_{\eta\eta\theta}G_i(\eta_i^*(\theta,\omega);\theta)][\nabla^2_{\eta\eta}G_i(\eta_i^*(\theta,\omega);\theta) + \lambda I]^{-1}\nabla^2_{\eta\theta}G_i(\eta_i^*(\theta,\omega);\theta) - [\nabla^2_{\eta\eta}G_i(\eta_i^*(\theta,\omega);\theta)
$$
$$
+ \lambda I]^{-1}[(\nabla^2_{\theta\theta}\eta_i^*(\theta,\omega))^\top\nabla^3_{\eta\eta\eta}G_i(\eta_i^*(\theta,\omega);\theta) + (\nabla_\theta\eta_i^*(\theta,\omega))^\top\cdot[(\nabla_\theta\eta_i^*(\theta,\omega))^\top\cdot
$$
$$
\nabla^4_{\eta\eta\eta\eta}G_i(\eta_i^*(\theta,\omega);\theta) + \nabla^4_{\eta\eta\eta\theta}G_i(\eta_i^*(\theta,\omega);\theta) + (\nabla_\theta\eta_i^*(\theta,\omega))^\top\nabla^4_{\eta\eta\eta\theta}G_i(\eta_i^*(\theta,\omega);\theta)
$$
$$
+ \nabla^4_{\eta\eta\theta\theta}G_i(\eta_i^*(\theta,\omega);\theta)][\nabla^2_{\eta\eta}G_i(\eta_i^*(\theta,\omega);\theta) + \lambda I]^{-1}\nabla^2_{\eta\theta}G_i(\eta_i^*(\theta,\omega);\theta)
$$
$$
- [\nabla^2_{\eta\eta}G_i(\eta_i^*(\theta,\omega);\theta) + \lambda I]^{-1}[(\nabla_\theta\eta_i^*(\theta,\omega))^\top\nabla^3_{\eta\eta\eta}G_i(\eta_i^*(\theta,\omega);\theta) + \nabla^3_{\eta\eta\theta}G_i(\eta_i^*(\theta,\omega);\theta)]\cdot
$$
$$
[\nabla^2_{\eta\eta}G_i(\eta_i^*(\theta,\omega);\theta) + \lambda I]^{-1}\nabla^3_{\eta\theta\theta}G_i(\eta_i^*(\theta,\omega);\theta) - [\nabla^2_{\eta\eta}G_i(\eta_i^*(\theta,\omega);\theta) + \lambda I]^{-1}\cdot
$$
$$
[\nabla^4_{\eta\theta\theta\theta}G_i(\eta_i^*(\theta,\omega);\theta) + (\nabla_\theta\eta_i^*(\theta,\omega))^\top\nabla^4_{\eta\theta\theta\eta}G_i(\eta_i^*(\theta,\omega);\theta)].
$$

Even if the expression looks complicated, we can conclude that $||\nabla^3_{\theta\theta\theta}\eta_i^*(\theta,\omega)||$ is bounded because each term in the expression is bounded. In speicifc, the fourth-order terms of $G_i$ are bounded (Lemma 6) .Then $D_\theta$ exists and similarly $D_\omega$ exists. $\qquad\square$

Then, we have the following inequality:

$$
\frac{\partial}{\partial\theta}L_i(\theta + \delta\Delta_\theta, \eta_i^*(\theta + \delta\Delta_\theta, \omega))
$$
$$
\leq \frac{\partial}{\partial\theta}L_i(\theta, \eta_i^*(\theta,\omega)) + \frac{\partial^2}{\partial\theta^2}L_i(\theta, \eta_i^*(\theta,\omega))\delta\Delta_\theta + \frac{D_\theta}{2}||\delta\Delta_\theta||^2,
$$
$$
\frac{\partial}{\partial\theta}L_i(\theta - \delta\Delta_\theta, \eta_i^*(\theta - \delta\Delta_\theta, \omega))
$$
$$
\geq \frac{\partial}{\partial\theta}L_i(\theta, \eta_i^*(\theta,\omega)) - \frac{\partial^2}{\partial\theta^2}L_i(\theta, \eta_i^*(\theta,\omega))\delta\Delta_\theta - \frac{D_\theta}{2}||\delta\Delta_\theta||^2,
$$
$$
\Rightarrow \frac{\partial}{\partial\theta}L_i(\theta + \delta\Delta_\theta, \eta_i^*(\theta + \delta\Delta_\theta, \omega)) - \frac{\partial}{\partial\theta}L_i(\theta - \delta\Delta_\theta, \eta_i^*(\theta - \delta\Delta_\theta, \omega))
$$
$$
\leq \frac{2\partial^2}{\partial\theta^2}L_i(\theta, \eta_i^*(\theta,\omega))\delta\Delta_\theta + D_\theta||\delta\Delta_\theta||^2;
$$
$$
\frac{\partial}{\partial\theta}L_i(\theta + \delta\Delta_\theta, \eta_i^*(\theta + \delta\Delta_\theta, \omega))
$$
$$
\geq \frac{\partial}{\partial\theta}L_i(\theta, \eta_i^*(\theta,\omega)) + \frac{\partial^2}{\partial\theta^2}L_i(\theta, \eta_i^*(\theta,\omega))\delta\Delta_\theta - \frac{D_\theta}{2}||\delta\Delta_\theta||^2,
$$
$$
\frac{\partial}{\partial\theta}L_i(\theta - \delta\Delta_\theta, \eta_i^*(\theta - \delta\Delta_\theta, \omega))
$$
$$
\leq \frac{\partial}{\partial\theta}L_i(\theta, \eta_i^*(\theta,\omega)) - \frac{\partial^2}{\partial\theta^2}L_i(\theta, \eta_i^*(\theta,\omega))\delta\Delta_\theta + \frac{D_\theta}{2}||\delta\Delta_\theta||^2,
$$
$$
\Rightarrow \frac{\partial}{\partial\theta}L_i(\theta + \delta\Delta_\theta, \eta_i^*(\theta + \delta\Delta_\theta, \omega)) - \frac{\partial}{\partial\theta}L_i(\theta - \delta\Delta_\theta, \eta_i^*(\theta - \delta\Delta_\theta, \omega))
$$
$$
\geq \frac{2\partial^2}{\partial\theta^2}L_i(\theta, \eta_i^*(\theta,\omega))\delta\Delta_\theta - D_\theta||\delta\Delta_\theta||^2.
$$

Therefore, we can conclude that

$$
\left|\left|\frac{1}{2\delta}\left[\frac{\partial}{\partial\theta}L_i(\theta + \delta\Delta_\theta, \eta_i^*(\theta + \delta\Delta_\theta, \omega)) - \frac{\partial}{\partial\theta}L_i(\theta - \delta\Delta_\theta, \eta_i^*(\theta - \delta\Delta_\theta, \omega))\right]\right.\right.
$$

$$-\frac{\partial^2}{\partial\theta^2}L_i(\theta,\eta_i^*(\theta,\omega))\Delta_\theta\bigg|\bigg| \leq \frac{D_\theta\delta}{2}||\Delta_\theta||^2. \tag{30}$$

Similarly, we can find that the approximation error of $\frac{\partial^2}{\partial\omega\partial\theta}L_i(\theta,\eta_i^*(\theta,\omega))\Delta_\theta$ is upper bounded by $\frac{D_\omega\delta}{2}||\Delta_\theta||^2$.

$$||\hat{\Delta}_{\theta,i} - \alpha\nabla_{\theta,i} - g_{\theta,i}||$$
$$\leq ||\hat{\Delta}_{\theta,i} - \Delta_{\theta,i}|| + \alpha||\nabla_{\theta,i} - \frac{\partial^2}{\partial\theta^2}\hat{L}_i(\theta,\hat{\eta}_i^*(\theta,\omega,\mathcal{D}_i^{\mathrm{h}}))\Delta_{\theta,i}||,$$
$$\overset{(a)}{\leq} O\left((\frac{C_{\nabla_{\eta\eta}^2 G}}{\lambda})^{\bar{K}} + (\frac{C_{\nabla_{\eta\eta}^2 G}}{\lambda})^K\right) + \frac{\alpha D_\theta\delta}{2}||\Delta_{\theta,i}||^2,$$
$$= O\left((\frac{C_{\nabla_{\eta\eta}^2 G}}{\lambda})^{\bar{K}} + (\frac{C_{\nabla_{\eta\eta}^2 G}}{\lambda})^K + \delta\right),$$

where $(a)$ follows Lemma 1 and (30).

Similarly, we can get $||\hat{\Delta}_{\omega,i} - \alpha\nabla_{\omega,i} - g_{\omega,i}|| \leq O\left((\frac{C_{\nabla_{\eta\eta}^2 G}}{\lambda})^{\bar{K}} + (\frac{C_{\nabla_{\eta\eta}^2 G}}{\lambda})^K + \delta\right)$.

### A.9 PROOF OF THEOREM 1

We define a function $f_i(\theta,\omega)$ such that $\nabla_\theta f_i(\theta,\omega) = \hat{\Delta}_{\theta,i} - \alpha\nabla_{\theta,i}$ and $\nabla_\omega f_i(\theta,\omega) = \hat{\Delta}_{\omega,i} - \alpha\nabla_{\omega,i}$. Recall that

$$\hat{\Delta}_{\theta,i} = \nabla_\varphi\hat{L}_i(\hat{\varphi}_i,\hat{\eta}_i(\hat{\varphi}_i,\omega,\mathcal{D}_i^{\mathrm{eval}},K),\mathcal{D}_i^{\mathrm{eval}}) - \nabla_{\varphi\eta}^2\hat{G}_i(\hat{\eta}_i(\hat{\varphi}_i,\omega,\mathcal{D}_i^{\mathrm{eval}},K);\hat{\varphi}_i,\mathcal{D}_i^{\mathrm{eval}})\cdot$$
$$[\lambda I + \nabla_{\eta\eta}^2\hat{G}_i(\hat{\eta}_i(\hat{\varphi}_i,\omega,\mathcal{D}_i^{\mathrm{eval}},K);\hat{\varphi}_i,\mathcal{D}_i^{\mathrm{eval}})]^{-1}\nabla_\eta\hat{L}_i(\hat{\varphi}_i,\hat{\eta}_i(\hat{\varphi}_i,\omega,\mathcal{D}_i^{\mathrm{eval}},K),\mathcal{D}_i^{\mathrm{eval}}),$$
$$\hat{\Delta}_{\omega,i} = \lambda[\lambda I + \nabla_{\eta\eta}^2\hat{G}_i(\hat{\eta}_i(\hat{\varphi}_i,\omega,\mathcal{D}_i^{\mathrm{eval}});\hat{\varphi}_i,\mathcal{D}_i^{\mathrm{eval}},K)]^{-1}\nabla_\eta\hat{L}_i(\hat{\varphi}_i,\hat{\eta}_i(\hat{\varphi}_i,\omega,\mathcal{D}_i^{\mathrm{eval}},K),\mathcal{D}_i^{\mathrm{eval}}).$$

Therefore, we can see that $||\hat{\Delta}_{\theta,i}||$ and $||\hat{\Delta}_{\omega,i}||$ are bounded because the second-order terms of $G_i$ are bounded (A.6) and the first-order terms of $L_i$ are also bounded (see the expressions in A.3). Since $\Delta_{\theta,i}$ and $\Delta_{\omega,i}$ are first-order approximations using partial gradients, $||\Delta_{\theta,i}||$ and $||\Delta_{\omega,i}||$ are bounded. Therefore, there exists positive constants $C_{\nabla_\theta f}$ and $C_{\nabla_\omega f}$ such that $||\nabla_\theta f_i(\theta,\omega)|| \leq C_{\nabla_\theta f}$ and $||\nabla_\omega f_i(\theta,\omega)|| \leq C_{\nabla_\omega f}$.

There are $m$ training tasks and we denote the distribution of the training tasks by $P_\mathcal{T}$. We define $f(\theta,\omega) \triangleq E_{i\sim P_\mathcal{T}}[f_i(\theta,\omega)]$ and thus we have:

$$E_{i\sim P_\mathcal{T}}\left[\nabla_\theta f(\theta,\omega) - \nabla_\theta f_i(\theta,\omega)\right] = 0, \quad E_{i\sim P_\mathcal{T}}\left[||\nabla_\theta f(\theta,\omega) - \frac{1}{B}\sum_{i=1}^B\nabla_\theta f_i(\theta,\omega)||^2\right] \leq \frac{C_{\nabla_\theta f}^2}{B}, \tag{31}$$

$$E_{i\sim P_\mathcal{T}}\left[\nabla_\omega f(\theta,\omega) - \nabla_\omega f_i(\theta,\omega)\right] = 0, \quad E_{i\sim P_\mathcal{T}}\left[||\nabla_\omega f(\theta,\omega) - \frac{1}{B}\sum_{i=1}^B\nabla_\omega f_i(\theta,\omega)||^2\right] \leq \frac{C_{\nabla_\omega f}^2}{B}. \tag{32}$$

**Claim 2.** *The gradient $\nabla_\theta f_i(\theta,\omega)$ is $C_\theta$-Lipschitz continuous in $(\theta,\omega)$ and $\nabla_\omega f_i(\theta,\omega)$ is $C_\omega$-Lipschitz continuous in $(\theta,\omega)$ for any task $\mathcal{T}_i$. Thus $f_i$ is $C_f$-smooth.*

*Proof.* To prove the existence of $C_\theta$, it suffices to show that $||\nabla_{\theta\theta}^2 f_i(\theta,\omega)||$ and $||\nabla_{\theta\omega}^2 f_i(\theta,\omega)||$ are bounded. We know that $||\nabla_{\theta\theta}^2 f_i(\theta,\omega)|| \leq ||\frac{\partial}{\partial\theta}\hat{\Delta}_{\theta,i}|| + \alpha||\frac{\partial}{\partial\theta}\nabla_{\theta,i}||$. Since $\nabla_{\theta,i}$ is the first-order approximation, $||\frac{\partial}{\partial\theta}\nabla_{\theta,i}|| \leq ||\frac{\partial}{\partial\theta}\hat{\Delta}_{\theta,i}||/\delta$. Therefore, it suffices to show that $||\frac{\partial}{\partial\theta}\hat{\Delta}_{\theta,i}||$ is bounded.

$$\frac{\partial}{\partial\theta}\hat{\Delta}_{\theta,i},$$
$$= \nabla_{\theta\theta}^2\hat{L}_i(\theta,\hat{\eta}(\cdot,\cdot,\cdot,K)) - \nabla_{\theta\theta\eta}^3 G_i(\hat{\eta}(\cdot,\cdot,\cdot,K);\theta)x(\bar{K}) - \nabla_{\theta\eta}^2 G_i(\hat{\eta}(\cdot,\cdot,\cdot,K);\theta)\nabla_\theta x(\bar{K}).$$

We know that the second-order terms of $L_i$ and $G_i$ (Appendix A.6) and the third-order terms of $G_i$ are bounded (Lemma 5). Therefore, it suffices to show that $||x(\bar{K})||$ and $||\nabla_\theta x(\bar{K})||$ are bounded.

$$x(\bar{k}+1),$$
$$= x(\bar{k}) - \beta\left([\lambda I + \nabla^2_{\eta\eta}\hat{G}_i(\hat{\eta}_i(\theta,\omega,\mathcal{D}_1,K);\theta,\mathcal{D}_1)]x(\bar{k}) - \nabla_\eta\hat{L}_i(\theta,\hat{\eta}_i(\theta,\omega,\mathcal{D}_1,K),\mathcal{D}_2)\right)$$

We know that $||[\lambda I + \nabla^2_{\eta\eta}\hat{G}_i(\hat{\eta}_i(\theta,\omega,\mathcal{D}_1,K);\theta,\mathcal{D}_1)]||$ and $||\nabla_\eta\hat{L}_i(\theta,\hat{\eta}_i(\theta,\omega,\mathcal{D}_1,K),\mathcal{D}_2)||$ are bounded. Therefore, $||x(\bar{K})||$ is bounded because $\bar{K}$ is a finite number.

We define $g(\bar{k}) \triangleq [\lambda I + \nabla^2_{\eta\eta}\hat{G}_i(\hat{\eta}_i(\theta,\omega,\mathcal{D}_1,K);\theta,\mathcal{D}_1)]x(\bar{k}) - \nabla_\eta\hat{L}_i(\theta,\hat{\eta}_i(\theta,\omega,\mathcal{D}_1,K),\mathcal{D}_2)$. Then, we know $||\nabla_\theta x(0)||$ and $||\nabla_\theta g(0)||$ are bounded. Suppose $||\nabla_\theta x(\bar{k})||$ and $||\nabla_\theta g(\bar{k})||$ are bounded, then $||\nabla_\theta x(\bar{k}+1)|| \leq ||\nabla_\theta x(\bar{k})|| + \beta||\nabla_\theta g(\bar{k})||$ is bounded and $||\nabla_\theta g(k+1)|| \leq ||\nabla^2_{\eta\eta\theta}\hat{G}_i(\hat{\eta}_i(\theta,\omega,\mathcal{D}_1,K);\theta,\mathcal{D}_1)|| \cdot ||x(\bar{k})|| + ||\lambda I + \nabla^2_{\eta\eta}\hat{G}_i(\hat{\eta}_i(\theta,\omega,\mathcal{D}_1,K);\theta,\mathcal{D}_1)|| \cdot ||\nabla_\theta x(\bar{k})|| + ||\nabla^2_{\eta\theta}\hat{L}_i(\theta,\hat{\eta}_i(\theta,\omega,\mathcal{D}_1,K),\mathcal{D}_2)||$ is bounded because the second-order terms of $L_i$ and $G_i$ are bounded (Appendix A.6) and the third-order terms of $G_i$ are bounded (Lemma 5). By induction and given that $\bar{K}$ is a finite number, we know that $||\nabla_\theta x(\bar{K})||$ is bounded.

Therefore, $C_\theta$ exists and similarly we can prove that $C_\omega$ exists. Thus, we can conclude the existence of $C_f$. $\qquad\square$

From Claim 2, we can also see that $f$ is $C_f$-smooth. Therefore

$$f(\theta(n+1),\omega(n+1)) \leq f(\theta(n),\omega(n)) + [\nabla_\theta f(\theta(n),\omega(n))]^\top[\theta(n+1)-\theta(n)]$$
$$+ [\nabla_\omega f(\theta(n),\omega(n))]^\top[\omega(n+1)-\omega(n)] + \frac{C_f}{2}[||\theta(n+1)-\theta(n)||^2 + ||\omega(n+1)-\omega(n)||^2],$$
$$= f(\theta(n),\omega(n)) - \frac{\alpha(n)}{B}\sum_{i=1}^{B}[\nabla_\theta f(\theta(n),\omega(n))]^\top\nabla_\theta f_i(\theta(n),\omega(n))$$
$$- \frac{\alpha(n)}{B}\sum_{i=1}^{B}[\nabla_\omega f(\theta(n),\omega(n))]^\top\nabla_\omega f_i(\theta(n),\omega(n))$$
$$+ \frac{C_f(\alpha(n))^2}{2}\left[||\frac{1}{B}\sum_{i=1}^{B}\nabla_\theta f_i(\theta(n),\omega(n))||^2 + ||\frac{1}{B}\sum_{i=1}^{B}\nabla_\omega f_i(\theta(n),\omega(n))||^2\right].$$

Take expectation over training task distribution $P_{\mathcal{T}}$ on both sides, we get that

$$E[f(\theta(n+1),\omega(n+1))],$$
$$\leq E[f(\theta(n),\omega(n))] + E\left\{E_{i\sim P_{\mathcal{T}}}\left[-\alpha(n)\left(||\nabla_\theta f(\theta(n),\omega(n))||^2 + ||\nabla_\omega f(\theta(n),\omega(n))||^2\right)\right.\right.$$
$$+ \frac{C_f(\alpha(n))^2}{2}\left(||\nabla_\theta f(\theta(n),\omega(n))||^2 + ||\nabla_\omega f(\theta(n),\omega(n))||^2\right) + \frac{C_f(\alpha(n))^2}{2}\left(||\nabla_\theta f(\theta(n),\right.$$
$$\left.\left.\left.\omega(n)) - \frac{1}{B}\sum_{i=1}^{B}\nabla_\theta f_i(\theta(n),\omega(n))||^2 + ||\nabla_\omega f(\theta(n),\omega(n)) - \frac{1}{B}\sum_{i=1}^{B}\nabla_\omega f_i(\theta(n),\omega(n))||^2\right)\right]\right\},$$
$$\overset{(a)}{\leq} E[f(\theta(n),\omega(n))] + \left(\frac{C_f(\alpha(n))^2}{2} - \alpha(n)\right)E[||\nabla f(\theta(n),\omega(n))||^2]$$
$$+ \frac{C_f(\alpha(n))^2}{2B}(C^2_{\nabla_\theta f} + C^2_{\nabla_\omega f}),$$
$$\Rightarrow \sum_{n=0}^{N-1}\left(\alpha(n) - \frac{C_f(\alpha(n))^2}{2}\right)E[||\nabla f(\theta(n),\omega(n))||^2],$$
$$\leq E[f(\theta(0),\omega(0)) - f(\theta(N),\omega(N))] + \sum_{n=0}^{N-1}\frac{C_f(\alpha(n))^2}{2B}(C^2_{\nabla_\theta f} + C^2_{\nabla_\omega f}),$$

$$\Rightarrow \sum_{n=0}^{N-1} \bar{\alpha}\alpha(n)E[||\nabla f(\theta(n),\omega(n))||^2] \leq \sum_{n=0}^{N-1}\left(\alpha(n) - \frac{C_f(\alpha(n))^2}{2}\right)E[||\nabla f(\theta(n),\omega(n))||^2],$$

$$\leq E[f(\theta(0),\omega(0)) - f(\theta(N),\omega(N))] + \sum_{n=0}^{N-1}\frac{C_f(\alpha(n))^2}{2B}(C_{\nabla_\theta f}^2 + C_{\nabla_\omega f}^2),$$

$$\Rightarrow \sum_{n=0}^{N-1}\frac{\bar{\alpha}^2}{N^\rho}E[||\nabla f(\theta(n),\omega(n))||^2] \leq \sum_{n=0}^{N-1}\bar{\alpha}\alpha(n)E[||\nabla f(\theta(n),\omega(n))||^2],$$

$$\leq E[f(\theta(0),\omega(0)) - f(\theta(N),\omega(N))] + \sum_{n=0}^{N-1}\frac{C_f(\alpha(n))^2}{2B}(C_{\nabla_\theta f}^2 + C_{\nabla_\omega f}^2),$$

$$\Rightarrow \frac{1}{N}\sum_{n=0}^{N-1}E[||\nabla f(\theta(n),\omega(n))||^2],$$

$$\leq \frac{1}{\bar{\alpha}^2 N^{1-\rho}}E\left[f(\theta(0),\omega(0)) - f(\theta(N),\omega(N))\right] + \frac{1}{\bar{\alpha}^2 N}\sum_{n=0}^{N-1}\frac{C_f(\alpha(n))^2}{2B}(C_{\nabla_\theta f}^2 + C_{\nabla_\omega f}^2).$$

where $(a)$ follows (31)-(32) and $\bar{\alpha} \leq \frac{2}{2+C_f}$. Therefore, we have that

$$\frac{1}{N}\sum_{n=0}^{N-1}E[||\nabla f(\theta(n),\omega(n))||] \leq \sqrt{\frac{1}{N}\sum_{n=0}^{N-1}E||\nabla f(\theta(n),\omega(n))||^2} \leq \sqrt{\frac{C_1}{N^{1-\rho}} + \frac{C_2}{BN}}, \quad (33)$$

where $C_1 \triangleq \frac{1}{\bar{\alpha}^2}E\left[f(\theta(0),\omega(0)) - f(\theta(N),\omega(N))\right]$ and $C_2 \triangleq \frac{1}{\bar{\alpha}^2}\sum_{n=0}^{N-1}\frac{C_f(\alpha(n))^2}{2}(C_{\nabla_\theta f}^2 + C_{\nabla_\omega f}^2)$. Note that $\sum_{n=0}^\infty(\alpha(n))^2$ is finite because $\alpha(n) \propto \frac{1}{(n+1)^\rho}$ and $\rho \in (\frac{1}{2}, 1)$.

We have the convergence of $f$, the next step is to quantify $||E[\nabla_\theta f] - E_{i\sim P_\mathcal{T}}[\frac{\partial}{\partial\theta}L_i(\varphi_i, \eta_i^*(\varphi_i,\omega))]||$ and $||E[\nabla_\omega f] - E_{i\sim P_\mathcal{T}}[\frac{\partial}{\partial\omega}L_i(\varphi_i, \eta_i^*(\varphi_i,\omega))]||$.

**Claim 3.** *There exist positive constants $\bar{D}_\theta$ and $\bar{D}_\omega$ such that $||\frac{\partial}{\partial\theta}L_i(\varphi_i, \eta_i^*(\varphi_i,\omega))|| \leq \bar{D}_\theta$ and $||\frac{\partial}{\partial\omega}L_i(\varphi_i, \eta_i^*(\varphi_i,\omega))|| \leq \bar{D}_\omega$.*

*Proof.* From the proof of Claim 1, we know that $||\frac{\partial}{\partial\theta}L_i(\theta, \eta_i^*(\theta,\omega))||$ is bounded. From Appendix A.3 we know $\nabla_{\theta\theta}L_i = \nabla_{\theta\theta}G_i$ and from Appendix A.6 we know $||\nabla_{\theta\theta}G_i||$ is bounded, therefore $||\frac{\partial}{\partial\theta}L_i(\varphi_i, \eta_i^*(\varphi_i,\omega))|| \leq ||I - \alpha\frac{\partial^2}{\partial\theta\partial\theta}L_i(\theta, \eta_i^*(\theta,\omega))|| \cdot ||\frac{\partial}{\partial\varphi}L_i(\varphi_i, \eta_i^*(\varphi_i,\omega))||$ is bounded. Thus, $\bar{D}_\theta$ exists and similarly we can prove that $\bar{D}_\omega$ exists. $\qquad\square$

$$E\left[\left|\left|E_{\mathcal{D}_i^{\text{train}},\mathcal{D}_i^{\text{eval}},\mathcal{D}_i^{\text{h}}}\left[\nabla_\theta f(\theta(n),\omega(n)) - E_{i\sim P_\mathcal{T}}[\frac{\partial}{\partial\theta}L_i(\varphi_i(n), \eta_i^*(\varphi_i(n),\omega(n)))]\right]\right|\right|\right],$$

$$\leq E\left[\left|\left|E_{\mathcal{D}_i^{\text{train}},\mathcal{D}_i^{\text{eval}},\mathcal{D}_i^{\text{h}}}E_{i\sim P_\mathcal{T}}[\nabla_\theta f_i(\theta(n),\omega(n)) - g_{\theta,i}]\right|\right|\right.$$

$$+ \left|\left|E_{\mathcal{D}_i^{\text{train}},\mathcal{D}_i^{\text{eval}},\mathcal{D}_i^{\text{h}}}E_{i\sim P_\mathcal{T}}[g_{\theta,i} - \frac{\partial}{\partial\theta}L_i(\varphi_i(n), \eta_i^*(\varphi_i(n),\omega(n)))]\right|\right|\right],$$

$$\overset{(b)}{\leq} O\left(\left(\frac{C_{\nabla_{\eta\eta}^2}G}{\lambda}\right)^K + \left(\frac{C_{\nabla_{\eta\eta}^2}G}{\lambda}\right)^{\bar{K}} + \delta\right)$$

$$+ E\left[\left|\left|E_{\mathcal{D}_i^{\text{train}},\mathcal{D}_i^{\text{eval}},\mathcal{D}_i^{\text{h}}}E_{i\sim P_\mathcal{T}}[g_{\theta,i} - \frac{\partial}{\partial\theta}L_i(\varphi_i(n), \eta_i^*(\varphi_i(n),\omega(n)))]\right|\right|\right],$$

$$\overset{(c)}{\leq} O\left(\left(\frac{C_{\nabla_{\eta\eta}^2}G}{\lambda}\right)^K + \left(\frac{C_{\nabla_{\eta\eta}^2}G}{\lambda}\right)^{\bar{K}} + \delta\right) + \frac{\alpha\bar{D}_\theta^2}{\min_i\{\sqrt{D_i^{\text{tr}}}\}_{i=1}^m}, \quad (34)$$

where $(b)$ follows Lemma 2 and $(c)$ follows Lemma 5.10 in (Fallah et al., 2020) if $\alpha \in [0, \frac{1}{\bar{D}_\theta}]$. Similarly, we can get

$$E\left[\left|\left|E_{\mathcal{D}_i^{\text{train}}, \mathcal{D}_i^{\text{eval}}, \mathcal{D}_i^{\text{h}}}E_{i \sim P_\mathcal{T}}[\nabla_\omega f_i(\theta(n), \omega(n)) - \frac{\partial}{\partial \omega}L_i(\varphi_i(n), \eta_i^*(\varphi_i(n), \omega(n)))]\right|\right|\right],$$
$$\leq O\left(\left(\frac{C_{\nabla_{\eta\eta}^2}G}{\lambda}\right)^K + \left(\frac{C_{\nabla_{\eta\eta}^2}G}{\lambda}\right)^{\bar{K}} + \delta\right) + \frac{\alpha \bar{D}_\omega^2}{\min_i\{\sqrt{D_i^{\text{tr}}}\}_{i=1}^m}. \tag{35}$$

With (34) and (35), we have that

$$E\left[\left|\left|E_{\mathcal{D}_i^{\text{train}}, \mathcal{D}_i^{\text{eval}}, \mathcal{D}_i^{\text{h}}}E_{i \sim P_\mathcal{T}}[\nabla f_i(\theta(n), \omega(n)) - \nabla L_i(\varphi_i(n), \eta_i^*(\varphi_i(n), \omega(n)))]\right|\right|\right],$$
$$\leq O\left(\left(\frac{C_{\nabla_{\eta\eta}^2}G}{\lambda}\right)^K + \left(\frac{C_{\nabla_{\eta\eta}^2}G}{\lambda}\right)^{\bar{K}} + \delta\right) + \frac{\alpha(\bar{D}_\theta^2 + \bar{D}_\omega^2)}{\min_i\{\sqrt{D_i^{\text{tr}}}\}_{i=1}^m}. \tag{36}$$

Therefore, we have that

$$E\left[\left|\left|\frac{1}{m}\sum_{i=1}^m \nabla L_i(\varphi_i(n), \eta_i^*(\varphi_i(n), \omega(n)))\right|\right|\right] = E\left[\left|\left|E_{i \sim P_\mathcal{T}}[\nabla L_i(\varphi_i(n), \eta_i^*(\varphi_i(n), \omega(n)))]\right|\right|\right],$$

$$\leq E\left[E_{\mathcal{D}_i^{\text{train}}, \mathcal{D}_i^{\text{eval}}, \mathcal{D}_i^{\text{h}}}[||\nabla f(\theta(n), \omega(n))||]\right]$$

$$+ E\left[\left|\left|E_{\mathcal{D}_i^{\text{train}}, \mathcal{D}_i^{\text{eval}}, \mathcal{D}_i^{\text{h}}}E_{i \sim P_\mathcal{T}}[\nabla_\omega f_i(\theta(n), \omega(n)) - \frac{\partial}{\partial \omega}L_i(\varphi_i(n), \eta_i^*(\varphi_i(n), \omega(n)))]\right|\right|\right],$$

$$\overset{(d)}{\leq} E\left[E_{\mathcal{D}_i^{\text{train}}, \mathcal{D}_i^{\text{eval}}, \mathcal{D}_i^{\text{h}}}||\nabla f(\theta(n), \omega(n))||\right]$$

$$+ O\left(\left(\frac{C_{\nabla_{\eta\eta}^2}G}{\lambda}\right)^K + \left(\frac{C_{\nabla_{\eta\eta}^2}G}{\lambda}\right)^{\bar{K}} + \delta\right) + \frac{\alpha(\bar{D}_\theta^2 + \bar{D}_\omega^2)}{\min_i\{\sqrt{D_i^{\text{tr}}}\}_{i=1}^m},$$

$$\leq \frac{1}{N}\sum_{n=1}^{N-1} E\left[E_{\mathcal{D}_i^{\text{train}}, \mathcal{D}_i^{\text{eval}}, \mathcal{D}_i^{\text{h}}}[||\nabla f(\theta(n), \omega(n))||]\right]$$

$$+ O\left(\left(\frac{C_{\nabla_{\eta\eta}^2}G}{\lambda}\right)^K + \left(\frac{C_{\nabla_{\eta\eta}^2}G}{\lambda}\right)^{\bar{K}} + \delta\right) + \frac{\alpha(\bar{D}_\theta^2 + \bar{D}_\omega^2)}{\min_i\{\sqrt{D_i^{\text{tr}}}\}_{i=1}^m},$$

$$\overset{(e)}{\leq} \sqrt{\frac{C_1}{N^{1-\rho}} + \frac{C_2}{BN}} + \frac{\alpha(\bar{D}_\theta^2 + \bar{D}_\omega^2)}{\min_i\{\sqrt{D_i^{\text{tr}}}\}_{i=1}^m} + O\left(\left(\frac{C_{\nabla_{\eta\eta}^2}G}{\lambda}\right)^K + \left(\frac{C_{\nabla_{\eta\eta}^2}G}{\lambda}\right)^{\bar{K}} + \delta\right),$$

$$\leq \epsilon + O\left(\left(\frac{C_{\nabla_{\eta\eta}^2}G}{\lambda}\right)^K + \left(\frac{C_{\nabla_{\eta\eta}^2}G}{\lambda}\right)^{\bar{K}} + \delta + \frac{1}{\min_i\{\sqrt{D_i^{\text{tr}}}\}_{i=1}^m}\right),$$

where $(d)$ follows (36) and $(e)$ follows (33). To find the iteration number $N$, we have that

$$\sqrt{\frac{C_1}{N^{1-\rho}} + \frac{C_2}{BN}} \leq \epsilon,,$$
$$\Rightarrow \epsilon^2 \geq \frac{C_1}{N^{1-\rho}} + \frac{C_2}{BN} \geq \frac{C_1}{BN^{1-\rho}} + \frac{C_2}{BN} \geq \frac{\min\{C_1, C_2\}}{BN},$$
$$N \geq \frac{\min\{C_1, C_2\}}{B\epsilon^2}.$$

### A.10 PROOF OF PROPOSITION 1

From Appendix A.3, we know that

$$\nabla_\theta L_i(\theta, \omega) = E_{\zeta \sim P_{\pi_{\omega;\theta}}}\left[\sum_{t=0}^\infty \gamma^t \nabla_\theta r_\theta(s_t, a_t)\right] - \nabla_\theta J_{r_\theta}(\pi_i),$$

$$\nabla_\omega L_i(\theta, \omega) = \nabla_\omega J_{c_\omega}(\pi_i) - E_{\zeta \sim P_{\pi_{\omega;\theta}}} \Big[\sum_{t=0}^\infty \gamma^t \nabla_\omega c_\omega(s_t, a_t)\Big].$$

Note that $\nabla_\theta J_{r_\theta}(\pi_i) = E_{\zeta \sim P_\pi}[\sum_{t=0}^\infty \gamma^t \nabla_\theta r_\theta(s_t, a_t)] = \frac{1}{1-\gamma} \int_{s \in \mathcal{S}} \int_{a \in \mathcal{A}} \mu^{\pi_i}(s, a) \nabla_\theta r_\theta(s, a) da ds$
and $\nabla_\omega J_{c_\omega}(\pi_i) = E_{\zeta \sim P_\pi}[\sum_{t=0}^\infty \gamma^t \nabla_\omega c_\omega(s_t, a_t)] = \frac{1}{1-\gamma} \int_{s \in \mathcal{S}} \int_{a \in \mathcal{A}} \mu^{\pi_i}(s, a) \nabla_\omega c_\omega(s, a) da ds$.

Therefore, we have that

$$||\frac{1}{m} \sum_{i=1}^m \nabla L_i(\theta, \omega) - \nabla L_{m+1}(\theta, \omega)||,$$

$$\leq ||\frac{1}{m} \sum_{i=1}^m \nabla_\theta L_i(\theta, \omega) - \nabla_\theta L_{m+1}(\theta, \omega)|| + ||\frac{1}{m} \sum_{i=1}^m \nabla_\omega L_i(\theta, \omega) - \nabla_\omega L_{m+1}(\theta, \omega)||,$$

$$\leq ||\frac{1}{m} \sum_{i=1}^m \nabla_\theta J_{r_\theta}(\pi_i) - \nabla_\theta J_{r_\theta}(\pi_{m+1})|| + ||\frac{1}{m} \sum_{i=1}^m \nabla_\theta J_{c_\omega}(\pi_i) - \nabla_\theta J_{c_\omega}(\pi_{m+1})||,$$

$$\leq \frac{1}{1-\gamma} \Big|\Big|\int_{s \in \mathcal{S}} \int_{a \in \mathcal{A}} \Big(\frac{1}{m} \sum_{i=1}^m \mu^{\pi_i}(s, a) - \mu^{\pi_{m+1}}(s, a)\Big) \nabla_\theta r_\theta(s, a) da ds \Big|\Big|$$

$$+ \frac{1}{1-\gamma} \Big|\Big|\int_{s \in \mathcal{S}} \int_{a \in \mathcal{A}} \Big(\frac{1}{m} \sum_{i=1}^m \mu^{\pi_i}(s, a) - \mu^{\pi_{m+1}}(s, a)\Big) \nabla_\omega c_\omega(s, a) da ds \Big|\Big|$$

$$\overset{(a)}{\leq} \frac{1}{1-\gamma} \Big|\int_{s \in \mathcal{S}} \int_{a \in \mathcal{A}} \Big(\frac{1}{m} \sum_{i=1}^m \mu^{\pi_i}(s, a) - \mu^{\pi_{m+1}}(s, a)\Big) da ds \Big|(\bar{C}_r + \bar{C}_c),$$

$$\leq \frac{\bar{C}_r + \bar{C}_c}{1-\gamma} \int_{s \in \mathcal{S}} \int_{a \in \mathcal{A}} \Big|\frac{1}{m} \sum_{i=1}^m \mu^{\pi_i}(s, a) - \mu^{\pi_{m+1}}(s, a)\Big| da ds,$$

$$= \frac{\bar{C}_r + \bar{C}_c}{1-\gamma} d(\frac{1}{m} \sum_{i=1}^m \mu^{\pi_i}, \mu^{\pi_{m+1}}) = O\big(d(\frac{1}{m} \sum_{i=1}^m \mu^{\pi_i}, \mu^{\pi_{m+1}})\big),$$

where $(a)$ follows Assumption 1 (i) (ii).

## A.11 PROOF OF THEOREM 2

From Theorem 1, we know that we can find meta-priors $(\bar{\theta}, \bar{\omega})$ such that

$$E\Big[\Big|\Big|\frac{1}{m} \sum_{i=1}^m \nabla L_i(\bar{\varphi}_i, \eta_i^*(\bar{\varphi}_i, \bar{\omega}))\Big|\Big|\Big] \leq \epsilon,$$

where $\bar{\varphi}_i = \bar{\theta} - \alpha \frac{\partial}{\partial \theta} L_i(\bar{\theta}, \eta_i^*(\bar{\theta}, \bar{\omega}))$. From (4)-(5), we know that

$$\frac{\partial}{\partial \theta} L_i(\bar{\varphi}_i, \eta_i^*(\bar{\varphi}_i, \bar{\omega})) = \Big[I - \alpha \frac{\partial^2}{\partial \theta^2} L_i(\bar{\theta}, \eta_i^*(\bar{\theta}, \bar{\omega}))\Big] \cdot \frac{\partial}{\partial \varphi} L_i(\bar{\varphi}_i, \eta_i^*(\bar{\varphi}_i, \bar{\omega})),$$

$$\frac{\partial}{\partial \omega} L_i(\bar{\varphi}_i, \eta_i^*(\bar{\varphi}_i, \bar{\omega})) = -\alpha \frac{\partial^2}{\partial \omega \partial \theta} L_i(\bar{\theta}, \eta_i^*(\bar{\theta}, \bar{\omega})) \cdot \frac{\partial}{\partial \varphi} L_i(\bar{\varphi}_i, \eta_i^*(\bar{\varphi}_i, \bar{\omega})) + \frac{\partial}{\partial \omega} L_i(\bar{\varphi}_i, \eta_i^*(\bar{\varphi}_i, \bar{\omega})).$$

From Appendix A.6, we know that the second-order terms of $L_i$ are bounded. From the expression of $\nabla_\theta L_i$ in Appendix A.3, we can see that $||\nabla_\theta L_i||$ is bounded. Therefore, we know that

$$||\frac{\partial}{\partial \varphi} L_i(\bar{\varphi}_i, \eta_i^*(\bar{\varphi}_i, \bar{\omega}))|| = O(||\frac{\partial}{\partial \theta} L_i(\bar{\varphi}_i, \eta_i^*(\bar{\varphi}_i, \bar{\omega}))||),$$

$$||\frac{\partial}{\partial \omega} L_i(\bar{\varphi}_i, \eta_i^*(\bar{\varphi}_i, \bar{\omega}))|| = O(||\frac{\partial}{\partial \omega} L_i(\bar{\varphi}_i, \eta_i^*(\bar{\varphi}_i, \bar{\omega}))||).$$

Therefore, we can see that

$$||\nabla L_i(\theta, \eta_i^*(\theta, \omega))|_{\theta=\bar{\varphi}_i, \omega=\bar{\omega}}||^2 \leq ||\frac{\partial}{\partial \varphi} L_i(\bar{\varphi}_i, \eta_i^*(\bar{\varphi}_i, \bar{\omega}))||^2 + ||\frac{\partial}{\partial \omega} L_i(\bar{\varphi}_i, \eta_i^*(\bar{\varphi}_i, \bar{\omega}))||^2,$$

$$
\begin{aligned}
&= O(||\frac{\partial}{\partial\theta}L_i(\bar{\varphi}_i, \eta_i^*(\bar{\varphi}_i, \bar{\omega}))||^2 + ||\frac{\partial}{\partial\omega}L_i(\bar{\varphi}_i, \eta_i^*(\bar{\varphi}_i, \bar{\omega}))||^2),\\
&= O(||\nabla L_i(\bar{\varphi}_i, \eta_i^*(\bar{\varphi}_i, \bar{\omega}))||^2),\\
&\Rightarrow ||\nabla L_i(\theta, \eta_i^*(\theta,\omega))|_{\theta=\bar{\varphi}_i,\omega=\bar{\omega}}|| \le O(||\nabla L_i(\bar{\varphi}_i, \eta_i^*(\bar{\varphi}_i, \bar{\omega}))||),\\
&\Rightarrow E[||\frac{1}{m}\sum_{i=1}^m \nabla L_i(\theta, \eta_i^*(\theta,\omega))|_{\theta=\bar{\varphi}_i,\omega=\bar{\omega}}||] \le O(E[||\frac{1}{m}\sum_{i=1}^m \nabla L_i(\bar{\varphi}_i, \eta_i^*(\bar{\varphi}_i, \bar{\omega}))||]) \le O(\epsilon).
\end{aligned}
\tag{37}
$$

From Appendix A.6, we know that there exists a positive constant $C_{\nabla^2 L}$ such that $||\nabla^2 L_i(\theta,\omega)|| \le C_{\nabla^2 L}$. Therefore, we have that

$$
\begin{aligned}
&E[||\frac{1}{m}\sum_{i=1}^m \nabla L_i(\theta,\eta)|_{\theta=\bar{\varphi}_i,\eta=\eta_i^*(\bar{\varphi}_i,\bar{\omega})} - \nabla L_i(\theta,\eta)|_{\theta=\hat{\varphi}_{m+1},\eta=\hat{\eta}_{m+1}^*}||],\\
&= E[||\frac{1}{m}\sum_{i=1}^m \nabla L_i(\theta,\eta)|_{\theta=\bar{\varphi}_i,\eta=\eta_i^*(\bar{\varphi}_i,\bar{\omega})} - \nabla L_i(\theta,\eta)|_{\theta=\bar{\varphi}_{m+1},\eta=\eta_{m+1}^*(\bar{\varphi}_{m+1},\bar{\omega})}||],\\
&\le \frac{C_{\nabla^2 L}}{m}\sum_{i=1}^m E\left[||\bar{\varphi}_i - \bar{\varphi}_{m+1}|| + ||\eta_i^*(\bar{\varphi}_i,\bar{\omega}) - \eta_{m+1}^*(\bar{\varphi}_{m+1},\bar{\omega})||\right],\\
&\overset{(a)}{=} O(\frac{1}{m}\sum_{i=1}^m d(\mu^{\pi_i}, \mu^{\pi_{m+1}})),
\end{aligned}
\tag{38}
$$

where $(a)$ follows Claim 4.

**Claim 4.** $\frac{1}{m}\sum_{i=1}^m ||\bar{\varphi}_i - \bar{\varphi}_{m+1}|| \le O(\frac{1}{m}\sum_{i=1}^m d(\mu^{\pi_i}, \mu^{\pi_{m+1}}))$ and $\frac{1}{m}\sum_{i=1}^m ||\eta_i^*(\bar{\varphi}_i,\bar{\omega}) - \eta_{m+1}^*(\bar{\varphi}_{m+1},\bar{\omega})|| \le O(\frac{1}{m}\sum_{i=1}^m d(\mu^{\pi_i}, \mu^{\pi_{m+1}}))$.

*Proof.* Follow the proof of Claim 1, we can see that that there exists a positive constant $C_{\theta\eta}$ such that $||\frac{\partial^2}{\partial\theta\partial\eta}L_i(\theta,\eta_i^*(\theta,\omega))|| \le C_{\theta\eta}$. Moreover, we know that there exists a constant $\bar{C}$ such that $||\nabla^2_{\theta\eta}G_i[\nabla_{\eta\eta}G_i + \lambda I]^{-1}|| \le \bar{C}$ because the second-order terms of $G_i$ are bounded (Appendix A.6).

$$
\begin{aligned}
\frac{1}{m}\sum_{i=1}^m ||\bar{\varphi}_i - \bar{\varphi}_{m+1}|| &= \frac{\alpha}{m}\sum_{i=1}^m ||\frac{\partial}{\partial\theta}L_{m+1}(\theta,\eta_{m+1}^*(\theta,\omega)) - \frac{\partial}{\partial\theta}L_i(\theta,\eta_i^*(\theta,\omega))||,\\
&= \frac{\alpha}{m}\sum_{i=1}^m \Big[||\frac{\partial}{\partial\theta}L_{m+1}(\theta,\eta_{m+1}^*(\theta,\omega)) - \frac{\partial}{\partial\theta}L_{m+1}(\theta,\eta_i^*(\theta,\omega))||\\
&\quad + ||\frac{\partial}{\partial\theta}L_{m+1}(\theta,\eta_i^*(\theta,\omega)) - \frac{\partial}{\partial\theta}L_i(\theta,\eta_i^*(\theta,\omega))||\Big],\\
&\le \frac{\alpha C_{\theta\eta}}{m}\sum_{i=1}^m ||\eta_{m+1}^*(\theta,\omega)) - \eta_i^*(\theta,\omega))||\\
&\quad + \frac{\alpha}{m}\sum_{i=1}^m ||\frac{\partial}{\partial\theta}L_{m+1}(\theta,\eta_i^*(\theta,\omega)) - \frac{\partial}{\partial\theta}L_i(\theta,\eta_i^*(\theta,\omega))||,\\
&\le \frac{\alpha C_{\theta\eta}}{m}||\eta_{m+1}^*(\theta,\omega)) - \eta_i^*(\theta,\omega))|| + \frac{\alpha}{m}\sum_{i=1}^m \Big[||\nabla_\theta L_{m+1}(\theta,\eta_i^*(\theta,\omega))\\
&\quad - \nabla_\theta L_i(\theta,\eta_i^*(\theta,\omega))|| + \bar{C}||\nabla_\eta L_{m+1}(\theta,\eta_i^*(\theta,\omega)) - \nabla_\eta L_i(\theta,\eta_i^*(\theta,\omega))||\Big],\\
&\overset{(b)}{\le} \frac{\alpha C_{\theta\eta}}{m}\sum_{i=1}^m ||\eta_{m+1}^*(\theta,\omega) - \eta_i^*(\theta,\omega)|| + \frac{\alpha\bar{C}}{m}\sum_{i=1}^m d(\mu^{\pi_i}, \mu^{\pi_{m+1}}),
\end{aligned}
$$

where $(b)$ follows Proposition 1. Now, we bound the term $||\eta_{m+1}^*(\theta,\omega) - \eta_i^*(\theta,\omega)||$.

We know that

$$\nabla_\eta \left[ G_{m+1}(\eta^*_{m+1}(\theta,\omega);\theta) + \frac{\lambda}{2}||\eta^*_{m+1}(\theta,\omega) - \omega||^2 \right] = 0,$$

$$\Rightarrow \eta^*_{m+1}(\theta,\omega) = \frac{1}{\lambda}\left[\omega - \nabla_\eta G_{m+1}(\eta^*_{m+1}(\theta,\omega);\theta)\right].$$

Therefore, we can get that

$$||\eta^*_{m+1}(\theta,\omega) - \eta^*_i(\theta,\omega)|| = \frac{1}{\lambda}||\nabla_\eta G_{m+1}(\eta^*_{m+1}(\theta,\omega);\theta) - \nabla_\eta G_i(\eta^*_i(\theta,\omega);\theta)||,$$

$$= O(||\nabla_\eta J_{c_{\eta_{m+1}}}(\pi_{m+1}) - \nabla_\eta J_{c_{\eta_i}}(\pi_i)||),$$

$$\leq \bar{C}_c O(d(\mu^{\pi_i},\mu^{\pi_{m+1}})).$$

Therefore, we can see that $\frac{1}{m}\sum_{i=1}^m ||\bar{\varphi}_i - \bar{\varphi}_{m+1}|| \leq O(\frac{1}{m}\sum_{i=1}^m d(\mu^{\pi_i},\mu^{\pi_{m+1}}))$ and similarly we can get $\frac{1}{m}\sum_{i=1}^m ||\eta^*_i(\bar{\varphi}_i,\bar{\omega}) - \eta^*_{m+1}(\bar{\varphi}_{m+1},\bar{\omega})|| \leq O(\frac{1}{m}\sum_{i=1}^m d(\mu^{\pi_i},\mu^{\pi_{m+1}})).$ $\qquad\square$

Therefore, we have that

$$E[||\nabla L_{m+1}(\theta,\omega)|_{\theta=\hat{\varphi}_{m+1},\omega=\hat{\eta}^*_{m+1}}||],$$

$$\leq E\left[||\nabla L_{m+1}(\theta,\omega)|_{\theta=\hat{\varphi}_{m+1},\omega=\hat{\eta}^*_{m+1}} - \frac{1}{m}\sum_{i=1}^m \nabla L_i(\theta,\omega)|_{\theta=\hat{\varphi}_{m+1},\omega=\hat{\eta}^*_{m+1}}||\right.$$

$$+ ||\frac{1}{m}\sum_{i=1}^m \nabla L_i(\theta,\omega)|_{\theta=\bar{\varphi}_i,\omega=\eta^*_i(\bar{\varphi}_i,\bar{\omega})} - \nabla L_i(\theta,\omega)|_{\theta=\hat{\varphi}_{m+1},\omega=\hat{\eta}^*_{m+1}}||$$

$$+ ||\frac{1}{m}\sum_{i=1}^m \nabla L_i(\theta,\omega)|_{\theta=\bar{\varphi}_i,\omega=\eta^*_i(\bar{\varphi}_i,\bar{\omega})}||\Bigg],$$

$$\overset{(c)}{\leq} O(\frac{1}{m}\sum_{i=1}^m d(\mu^{\pi_i},\mu^{\pi_{m+1}}) + d(\frac{1}{m}\sum_{i=1}^m \mu^{\pi_i},\mu^{\pi_{m+1}}) + \epsilon),$$

where $(c)$ follows (38), (37), and Proposition 1.

## A.12 PROOF OF THEOREM 3

The proof is similar to the proof in Appendix A.11. The key step is to find the relation similar to Proposition 1 which is the following claim.

**Claim 5.** *For a given $(\theta,\omega)$, it holds that*

$$||\frac{1}{m}\sum_{i=1}^m L_i(\theta,\omega) - L_{m+1}(\theta,\omega)|| \leq O(d(\frac{1}{m}\sum_{i=1}^m \mu^{\pi_i},\mu^{\pi_{m+1}})).$$

*Proof.* Recall that

$$L_i(\theta,\omega) = -E_{\zeta\sim P_{\pi_i}}[\sum_{t=0}^\infty \log \pi_{\omega;\theta}(a_t|s_t)] = -\int_{s\in\mathcal{S}}\int_{a\in\mathcal{A}} \mu^{\pi_i}(s,a)\log \pi_{\omega;\theta}(a|s)dads.$$

Therefore, we have that

$$||\frac{1}{m}\sum_{i=1}^m L_i(\theta,\omega) - L_{m+1}(\theta,\omega)||,$$

$$\leq \int_{s\in\mathcal{S}}\int_{a\in\mathcal{A}} |\frac{1}{m}\sum_{i=1}^m \mu^{\pi_i}(s,a) - \mu^{\pi_{m+1}}(s,a)| \cdot ||\log \pi_{\omega;\theta}(a|s)||dads,$$

$$\overset{(a)}{\leq} O(d(\frac{1}{m}\sum_{i=1}^m \mu^{\pi_i},\mu^{\pi_{m+1}})),$$

where $(a)$ follows that fact that $(\theta,\omega)$ is fixed and thus there is a positive constant $D$ such that $||\log \pi_{\omega;\theta}(a|s)|| \leq D$ for any $(s,a) \in \mathcal{S} \times \mathcal{A}$. $\qquad\square$

**Remark 1.** *Note that Claim 5 does not hold for any $(s, a) \in \mathcal{S} \times \mathcal{A}$, it only holds for given (specific) $(\theta, \omega)$. However, this is enough because we only need to use Claim 5 at several specific points, e.g., $(\bar{\varphi}_i, \eta_i^*(\bar{\varphi}_i, \bar{\omega}))$ and $(\hat{\varphi}_{m+1}, \hat{\eta}_{m+1}^*)$.*

We can see that

$$E[L_{m+1}(\hat{\varphi}_{m+1}, \hat{\eta}_{m+1}^*(\hat{\varphi}_{m+1}, \bar{\omega}, \mathcal{D}_{m+1}))] - \min_{\theta, \omega} L_{m+1}(\theta, \omega),$$

$$= E[L_{m+1}(\bar{\varphi}_{m+1}, \eta_{m+1}^*(\bar{\varphi}_{m+1}, \bar{\omega}))] - \min_{\theta, \omega} L_{m+1}(\theta, \omega),$$

$$\leq E\left[L_{m+1}(\bar{\varphi}_{m+1}, \eta_{m+1}^*(\bar{\varphi}_{m+1}, \bar{\omega})) - \frac{1}{m} \sum_{i=1}^m L_i(\bar{\varphi}_{m+1}, \eta_{m+1}^*(\bar{\varphi}_{m+1}, \bar{\omega}))\right]$$

$$+ E\left[\frac{1}{m} \sum_{i=1}^m L_i(\bar{\varphi}_{m+1}, \eta_{m+1}^*(\bar{\varphi}_{m+1}, \bar{\omega})) - \frac{1}{m} \sum_{i=1}^m L_i(\bar{\varphi}_i, \eta_i^*(\bar{\varphi}_i, \bar{\omega}))\right]$$

$$+ E\left[\frac{1}{m} \sum_{i=1}^m L_i(\bar{\varphi}_i, \eta_i^*(\bar{\varphi}_i, \bar{\omega}))\right] - \frac{1}{m} \sum_{i=1}^m \min_{\theta, \omega} L_i(\varphi_i, \eta_i^*(\varphi_i, \omega))$$

$$+ \frac{1}{m} \sum_{i=1}^m \min_{\theta, \omega} L_i(\varphi_i, \eta_i^*(\varphi_i, \omega)) - \min_{\theta, \omega} L_{m+1}(\theta, \omega).$$

The first and fourth terms are bounded by $O(d(\frac{1}{m} \sum_{i=1}^m \mu^{\pi_i}, \mu^{\pi_{m+1}}))$ (Claim 5), and the third term is bounded by $\epsilon$. Now, we look at the second term. Since $||\nabla_\theta L_i(\theta, \omega)||$ and $||\nabla_\omega L_i(\theta, \omega)||$ are both bounded (see the expressions in Appendix A.3), there is a positive constant $C_{\nabla L}$ such that $||\nabla L(\theta, \omega)|| \leq C_{\nabla L}$.

$$E\left[\frac{1}{m} \sum_{i=1}^m L_i(\bar{\varphi}_{m+1}, \eta_{m+1}^*(\bar{\varphi}_{m+1}, \bar{\omega})) - \frac{1}{m} \sum_{i=1}^m L_i(\bar{\varphi}_i, \eta_i^*(\bar{\varphi}_i, \bar{\omega}))\right],$$

$$\leq \frac{C_{\nabla L}}{m} \sum_{i=1}^m \left[||\bar{\varphi}_{m+1} - \bar{\varphi}_i|| + ||\eta_{m+1}^*(\bar{\varphi}_{m+1}, \bar{\omega})) - \eta_i^*(\bar{\varphi}_i, \bar{\omega}))||\right],$$

$$\overset{(b)}{\leq} O(\frac{1}{m} \sum_{i=1}^m d(\mu^{\pi_i}, \mu^{\pi_{m+1}})),$$

where $(b)$ follows Claim 4. Therefore, we have that

$$E[L_{m+1}(\hat{\varphi}_{m+1}, \hat{\eta}_{m+1}^*(\hat{\varphi}_{m+1}, \bar{\omega}, \mathcal{D}_{m+1}))] - \min_{\theta, \omega} L_{m+1}(\theta, \omega),$$

$$\leq \epsilon + O(\frac{1}{m} \sum_{i=1}^m d(\mu^{\pi_i}, \mu^{\pi_{m+1}}) + d(\frac{1}{m} \sum_{i=1}^m \mu^{\pi_i}, \mu^{\pi_{m+1}}))$$

### A.13 PROOF OF THEOREM 4

If the reward functions and cost functions are linear, there are reward and cost feature vectors $\phi_r(\cdot, \cdot)$ and $\phi_c(\cdot, \cdot)$ defined over state-action space. The expert's reward function of the new task $\mathcal{T}_{m+1}$ are $r_{m+1} = \theta_E^\top \phi_r$ and $c_{m+1} = \omega_E^\top \phi_c$. The parameterized reward and cost functions are respectively $r_\theta = \theta^\top \phi_r$ and $c_\omega = \omega^\top (\phi_c)$. We define the expected cumulative reward feature as $\mu_r(\pi) \triangleq E_{\zeta \sim P_\pi}[\sum_{t=0}^\infty \gamma^t \phi_r(s_t, a_t)]$ and expected cumulative cost feature as $\mu_c(\pi) \triangleq E_{\zeta \sim P_\pi}[\sum_{t=0}^\infty \gamma^t \phi_c(s_t, a_t)]$. Therefore, from Appendix A.3, we can see that for any $(\theta, \omega)$ and any task $\mathcal{T}_i$:

$$\nabla_\theta L_i(\theta, \omega) = \mu_r(\pi_{\omega;\theta}) - \mu_r(\pi_i), \quad \nabla_\omega L_i(\theta, \omega) = \mu_c(\pi_i) - \mu_c(\pi_{\omega;\theta}).$$

Therefore, we know that

$$E[||\nabla_\theta L_{m+1}(\theta, \omega)|_{\theta=\hat{\varphi}_{m+1}, \omega=\hat{\eta}_{m+1}^*(\hat{\varphi}_{m+1}, \bar{\omega}, \mathcal{D}_{m+1})}||],$$

$$\leq E[||\nabla L_{m+1}(\theta, \omega)|_{\theta=\hat{\varphi}_{m+1}, \omega=\hat{\eta}_{m+1}^*(\hat{\varphi}_{m+1}, \bar{\omega}, \mathcal{D}_{m+1})}||],$$

$$\leq O(\epsilon + \frac{1}{m} \sum_{i=1}^{m} d(\mu^{\pi_i}, \mu^{\pi_{m+1}}) + d(\frac{1}{m} \sum_{i=1}^{m} \mu^{\pi_i}, \mu^{\pi_{m+1}})).$$

Thus we have that

$$E[|J_{r_{m+1}}(\pi_{\hat{\eta}_{m+1}^*; \hat{\varphi}_{m+1}}) - J_{r_{m+1}}(\pi_{m+1})|],$$
$$\leq E[||\theta_E|| \cdot ||\mu_r(\pi_{\hat{\eta}_{m+1}^*; \hat{\varphi}_{m+1}}) - \mu_r(\pi_{m+1})||],$$
$$= O\bigg( E[||\mu_r(\pi_{\hat{\eta}_{m+1}^*; \hat{\varphi}_{m+1}}) - \mu_r(\pi_{m+1})||] \bigg),$$
$$= O\bigg( E[||\nabla L_{m+1}(\theta, \omega)|_{\theta = \hat{\varphi}_{m+1}, \omega = \hat{\eta}_{m+1}^*(\hat{\varphi}_{m+1}, \bar{\omega}, \mathcal{D}_{m+1})}||] \bigg),$$
$$\leq O(\epsilon + \frac{1}{m} \sum_{i=1}^{m} d(\mu^{\pi_i}, \mu^{\pi_{m+1}}) + d(\frac{1}{m} \sum_{i=1}^{m} \mu^{\pi_i}, \mu^{\pi_{m+1}})).$$

Similarly, we can get

$$E[|J_{c_{m+1}}(\pi_{\hat{\eta}_{m+1}^*; \hat{\varphi}_{m+1}}) - J_{c_{m+1}}(\pi_{m+1})|] \leq O(\epsilon + \frac{1}{m} \sum_{i=1}^{m} d(\mu^{\pi_i}, \mu^{\pi_{m+1}}) + d(\frac{1}{m} \sum_{i=1}^{m} \mu^{\pi_i}, \mu^{\pi_{m+1}})).$$

### A.14 OUR DISTINCTIONS FROM RELATED META-LEARNING THEORETICAL WORKS

Although there is no theoretical work on meta IRL, there are theoretical works on general meta-learning problems that study convergence and generalization. In specific, (Fallah et al., 2020) studies the convergence of MAML, (Fallah et al., 2021b) studies the generalization of MAML, and (Denevi et al., 2019) studies the generalization of iMAML. In this section, we discuss our significant distinctions from these works from three perspectives: problem, algorithm, and theoretical analysis.

**Problem**: Neither MAML nor iMAML can be directly applied to solve our problem even if we reduce their general problem formulation to the context of IRL. Therefore, we propose the novel problem formulation (2)-(3) to learn both the reward and cost meta-priors.

**Algorithm**: A key step of both MAML and iMAML is to compute the hyper-gradient. In MAML, the hyper-gradient is assumed to be directly computed. In iMAML, they use conjugate gradient to help compute the hyper-gradient. In our case, we have an additional challenge to compute the hyper-gradient, i.e., Challenge (i) mentioned in Subsection 3.1, that does not occur in MAML or iMAML. To solve this new challenge, we design an additional algorithm (i.e., Algorithm 2) that uses the first-order approximation (6)-(7). Moreover, instead of using conjugate gradient to solve Challenge (ii) mentioned in Subsection 3.1, we use a new algorithm (Algorithm 3) to solve it. Compared to iMAML (Rajeswaran et al., 2019) that assumes to find a $\delta'$-approximate, Algorithm 3 enables us to guarantee the finite-time approximation error in Lemma 1 and Lemma 2.

**Theoretical analysis**: We first talk about the convergence guarantee and then talk about the generalization analysis. For the convergence guarantee, first, we have additional algorithms (i.e., Algorithms 2-3) to tackle the additional challenges and thus we need additional analysis (Subsections A.7-A.8) to justify Lemmas 1-2. Note that Lemmas 1-2 and the corresponding proof are novel compared to (Fallah et al., 2020; 2021b; Denevi et al., 2019). Moreover, for the proof of Theorem 1 in Subsection A.9, we use a novel technique to prove the convergence, i.e., we first construct a new function $f_i$ and provide the convergence of $f_i$. Then the convergence of $L_i$ can be derived by using the error between $f_i$ and $L_i$. This technique can simplify the analysis and is totally different from (Fallah et al., 2020).

For the generalization analysis, Proposition 1 and its proof is novel compared to (Fallah et al., 2021b; Denevi et al., 2019) since we leverage a unique property (i.e., distance between tasks and bounded gradient of the parameterized models $r_\theta$ and $c_\omega$) of our problem to prove this. Similarly, Theorems 2-3 require to leverage the special property of our problem to prove. Theorem 4 quantifies the cumulative reward and cost difference between the adapted policy and the expert policy, which is definitely novel.

### A.15 COMPARISON TO META CONSTRAINED REINFORCEMENT LEARNING (KHATTAR ET AL., 2023)

First, (Khattar et al., 2023) studies a different problem. (Khattar et al., 2023) studies meta constrained RL where the constraint (cost function) is given while we study meta ICRL where the cost function needs to be learned.

For dealing with the constraint, (Khattar et al., 2023) has a constrained RL problem to solve for each task. It uses a constrained RL algorithm called CRPO (Xu et al., 2021) to solve the constrained RL problem and get a corresponding task-specific policy. We have a cost learning problem in the lower level and we use gradient descent to obtain a corresponding task-specific cost adaptation. One similarity between ((Khattar et al., 2023) and our work is that we both do not require the exact task-specific adaptation and this makes the theoretical analysis of the meta learning performance more challenging.

For dealing with meta-learning, (Khattar et al., 2023) studies an online setting where at each online iteration, a new task is input and a corresponding task-specific policy adaptation is computed. At each online iteration, it updates the policy meta-prior by minimizing the KL divergence between the policy meta-prior and the current task-specific policy adaptation via one or multiple online gradient descent steps. In contrast, we utilize a bi-level optimization framework where we learn the meta-priors in the upper level such that the corresponding task-specific adaptations can maximize the likelihood of the demonstrations of each task. In order to optimize for the meta-priors, we need to compute the hyper-gradient which is very challenging in our case. We propose several novel approximation methods and algorithm designs to approximate the hyper-gradient. In conclusion, the meta-prior in our case is learned such that the task-specific adaptations adapted from the meta-prior have good performance on each specific task while the meta-prior in (Khattar et al., 2023) is learned such that the meta-prior is close to task-specific adaptations according to the metric of KL divergence.

## B EXPERIMENT DETAILS

This section includes the experiment details. It has two subsections where the first subsection includes the experiment details of the drone experiment and the second subsection includes the experiment details of the Mujoco experiment. We first explain the four baselines in detail.

- The baseline **ICRL** does not have meta-priors and directly learn from one trajectory from scratch.
- The baseline **ICRL(pre)** naively learns meta-priors across all demonstrations of all the training tasks. In specific, ICRL(pre) first solves $\min_\theta \frac{1}{m} \sum_{i=1}^{m} \hat{L}_i(\theta, \omega^*(\theta), \mathcal{D}_i)$, s.t. $\omega^*(\theta) = \arg\min_\omega \frac{1}{m} \sum_{i=1}^{m} \hat{G}_i(\omega; \theta, \mathcal{D}_i)$ where $\mathcal{D}_i \triangleq \{\mathcal{D}_i^{\text{tr}}, \mathcal{D}_i^{\text{eval}}, \mathcal{D}_i^{\text{h}}\}$. The obtained results of this problem (i.e., $\theta$ and $\omega^*(\theta)$) are the meta-priors of ICRL(pre). ICRL(pre) then uses these meta-priors as initializations to solve the problem $\min_\theta \hat{L}_{m+1}(\theta, \omega^*(\theta), \mathcal{D}_{m+1})$, s.t. $\omega^*(\theta) = \arg\min_\omega \hat{G}_{m+1}(\omega; \theta, \mathcal{D}_{m+1})$ for an arbitrary new task $\mathcal{T}_{m+1}$.
- The baseline **Meta-IRL** is from (Xu et al., 2019) which is a combination of maximum entropy IRL (Ziebart et al., 2008) and MAML (Finn et al., 2017).

### B.1 DRONE NAVIGATION WITH OBSTACLES

RL has been applied to many applications, including wireless network Huang et al. (2023) and motion planning Liu & Zhu (2024). Here, we study a motion planning problem. For the drone experiment, we cannot directly train the algorithm on the physical drone because this may cause damage to the drone. In specific, given learned reward and cost parameters $(\theta, \omega)$, we need to use soft Q learning or soft actor-critic to find the corresponding constrained soft Bellman policy. This RL step requires the drone to interact with the environment and thus improves its policy. During the learning process, the drone may inevitably execute some dangerous behaviors, such as colliding with obstacles or the wall. To avoid the damage of the drone, we build a simulator in Gazebo (Figure 2) that imitates the physical environment with the scale $1 : 1$. We train the algorithm on the simulated

drone in the simulator and the empirical results, i.e., CVR and SR, are counted in the simulator. Once we obtain a learned policy that has good performance in the simulator, we implement the policy on the physical drone.

**Discussion of the sim-to-real problem**. In some cases, the models that have good performance in the simulator may not have good performance in the real world due to the reason that the simulator cannot $100\%$ precisely imitate the physical world Sun et al. (2023); Wang & Cao (2024). However, in our case, the sim-to-real issue is not significant because of two reasons: (i) the simulated drone is built according to the dynamics of a real Ar. Drone 2.0 (Huang & Sturm, 2014); (ii) the states and actions are just the coordinates of the location and the heading direction of the drone instead of some low-level control such as the motor's velocity, etc. Given that Vicon can output precise pose of the physical drone and the simulator is built on the $1 : 1$ scale. If a learned trajectory can succeed in the simulator, it can succeed in the real world given that the low-level control of both the simulated and physical drones are given.

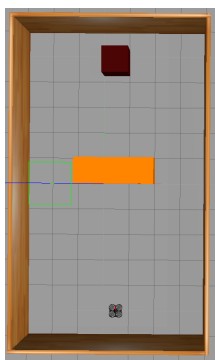

Figure 2: Simulator

In this experiment, the state of the drone is its 3-D coordinate $(x, y, z)$ and the action of the drone is also a 3-D coordinate $(dx, dy, dz)$ which captures the heading direction of the drone. We fix the length of each step as $0.1$ and thus the next state is $(x + \frac{dx}{10\sqrt{(dx)^2+(dy)^2+(dz)^2}}, y + \frac{dy}{10\sqrt{(dx)^2+(dy)^2+(dz)^2}}, z + \frac{dz}{10\sqrt{(dx)^2+(dy)^2+(dz)^2}})$. In this experiment, we do not need the drone to change its height so that we usually fix the value of $z$ and set $dz = 0$. The goal is an $1 \times 1$ square. Denote the coordinate of the center of the goal as $(x_{\text{goal}}, y_{\text{goal}})$, then for all the different tasks, $x_{\text{goal}} \in (0.5, 6.5)$ and $y_{\text{goal}} \in (10, 11)$. The obstacle is a $3 \times 1$ square. Denote the coordinate of the lower left end of the obstacle as $(x_{\text{obstacle}}, y_{\text{obstacle}})$, the for the different tasks, $x_{\text{obstacle}} \in (0, 4)$ and $y_{\text{obstacle}} \in (4, 5)$.

Note that we do not need features to help learn the reward function. Even if features can be learned Wu et al. (2024); Chen et al. (2023), the extra requirement of needing features can be impractical in various scenarios. We use neural networks to parameterize the reward and cost functions. In specific, the neural networks have two layers where the activation functions are relu and each layer has $64$ neurons. For each training task, the training set only has one demonstration and the evaluation set has 50 demonstrations. We set $\mathcal{D}_i^{\text{h}} = \mathcal{D}_i^{\text{eval}}$. The result in table 1 shows the mean and the standard deviation over the 10 training tasks.

## B.2 MUJOCO EXPERIMENT

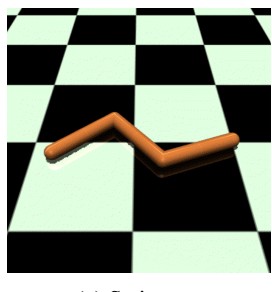
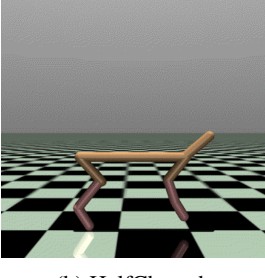
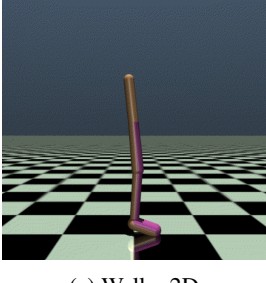

(a) Swimmer  (b) HalfCheetah  (c) Walker2D

Figure 3: The three Mujoco environments.

For all the three Mujoco experiments, we consider the random velocity task (Seyed Ghasemipour et al., 2019). In specific, the robots (i.e., swimmer, halfcheetah, and walker) need to reach and sustain at a target velocity. For different tasks, the target velocity is different. The design of the reward function is that the robots will receive reward $+1$ if they are at the target velocity and reward $0$ otherwise. The neural networks of all the three experiments have two hidden layers, and each layer has 256 neurons. The activation function of the first hidden layer is relu and the activation function of the second hidden layer is tanh. For all the three Mujoco experiments, we have 50

training tasks and 10 test tasks. Each test task only has one demonstration. For the training tasks, the training set has one demonstration and the evaluation set has 64 demonstrations. Similar to the drone experiment, we set $\mathcal{D}_i^{\mathrm{h}} = \mathcal{D}_i^{\mathrm{eval}}$.

The target velocity of the three experiments is designed in a similar way. In specific, the target velocity $v \in [0.5, 3.5]$. For a given target velocity $v$, the reward is $+1$ if the velocity is within $[v, v + 0.3]$ and 0 otherwise. For the constraint, the Swimmer experiment constrains all the states whose front tip angle is larger than $a_0$ where $a_0 \in [0.9, 1.2]$. The HalfCheetah experiment constrains all the states whose front tip height is larger than $h_0$ where $h_0 \in [0.5, 1.0]$. The Walker experiment constrains all the states whose top height is smaller than $h_1$ where $h_1 \in [0.6, 0.9]$.

**Discussion of the experiment results in Table 1**. From Table 1, we can observe that M-ICRL achieves similar SR/CR and CVR with the expert. ICRL has much worse SR/CR performance because it only has one demonstration for each test task, and it does not have meta-priors. Meta-IRL has the worst CVR performance because it only learns a reward function and it is difficult for a single reward function to capture the function of both the ground truth reward function and ground truth constraints. The bad CVR performance will also result in bad performance of SR and CVR even if Meta-IRL may estimate the ground truth reward function well, because the episode terminates if any constraint is violated. ICRL(pre) has the second-best performance because it has abundant training data and it learns both the reward function and constraints. However, ICRL(pre) has much worse performance than M-ICRL because ICRL(pre) only naively trains over all the data of all the training tasks.

