# OpenReview forum: "Meta Inverse Constrained Reinforcement Learning: Convergence Guarantee and Generalization Analysis"
_ICLR.cc/2024/Conference — ICLR 2024 poster_

### Official Review · Reviewer_Eje2 · 2023-10-15

**Soundness:** 3 good
**Presentation:** 3 good
**Contribution:** 2 fair
**Rating:** 6
**Confidence:** 3

**Summary:**

This paper addresses the challenge of inferring both reward and cost functions from expert demonstrations within a meta-learning environment. Specifically, the environment comprises a set of tasks, each with its unique rewards and constraints. To aid the inverse learning process, every task is paired with a demonstration dataset, capturing the actions of expert agents. The objective is to derive meta-priors for both reward functions and constraints that can be readily adapted to new tasks.

To tackle the meta Inverse Constrained Reinforcement Learning (ICRL) problem, the paper extends the bi-level optimization framework proposed by Liu & Zhu (2022). In this model, the upper-level problem focuses on reward learning, while the lower-level problem tackles constraint learning. Following this framework, the paper introduces an algorithm that estimates gradients at multiple levels. Due to the computational challenges associated with calculating the inverse-of-Hessian term (or hyper-gradients) and solving the exact lower-level problems, a set of approximation methods is proposed. The primary techniques include 1) a first-order approximation to second-order gradients, 2) substituting the computation of the inverse-of-Hessian term with an optimization problem solution, and 3) approximating the solution to the lower-level problem with multiple iterations of gradient descent.

To validate the convergence of the proposed algorithm, this paper extends the results of the ϵ-approximate first order stationary point (ϵ-FOSP) from Fallah et al. (2020) to its context. Specifically, it demonstrates that the norm of the empirical loss estimation is bounded upon convergence (with optimal estimate). Furthermore, this paper derives an upper bound on the empirical loss estimation for new tasks, reinforcing the generalization performance of the proposed method.

The empirical study primarily focuses on the physical drone setting and the Mujoco setting. These settings are used to substantiate the empirical performance of this method.

**Strengths:**

1. The paper is well-structured and written, with many details provided to aid readers in understanding the principal contributions and the main claims. The notations are carefully designed and well-defined.
2. The algorithm proposed is well presented and generally straightforward to understand, which speaks to the clarity and precision of the authors' exposition.
3. The theoretical results appear sound overall. The requisite assumptions underlying the main proof are well conveyed. References to key methods appear comprehensive, though it's not entirely clear whether the convergence and generalization outcomes are consistent with and comparable to the principal results under the meta Inverse Reinforcement Learning (IRL) or Reinforcement Learning (RL) frameworks. In RL, the analysis of regret or sample complexity is typically expected, yet the theoretical results in this paper predominantly cater to gradient-based methods.
4. The empirical results underscore the superior performance of the M-ICRL in comparison to other benchmarks. This highlights the practical efficacy of the proposed approach in real-world scenarios.

**Weaknesses:**

1.  **Novelty**. Liu & Zhu (2022) has proposed a bi-level optimization objective for ICRL. This paper expands upon that proposal, adapting the objective to fit within a meta-learning context. The primary innovation is the integration of a meta-prior into the initial objective (see objective (2)). The algorithm presented outlines a direct, gradient-based optimization method, which, unfortunately, is computationally infeasible under most circumstances. The core focus of this paper is on *mitigating this computational difficulty* through the implementation of three levels of approximation. Additionally, it explores the impact of these approximations on both convergence and generalization. However, it should be noted that this study deviates somewhat from the original goal of Meta ICRL. Moreover, the advancements it provides over Liu & Zhu (2022) could be considered somewhat marginal.

2. **Theoretical Contributions.** I have several concerns about the theoretical results.
- Theorem 1 sets an upper bound on the expected gradient of the exact optimal solution. When compared to other more frequently used results, such as the upper bound of regret or the sample complexity in the PAC (Probably Approximately Correct) analysis, this result doesn't appear particularly robust.
- It is surprising to note that the upper bound is raised to the power of $K$ (the number of gradient steps), rather than being linear or sublinear with respect to $K$. This implies that the approximation error could accumulate rapidly with each gradient step. Unfortunately, the paper does not provide a satisfactory explanation for this issue.
- I question the assertion that this upper bound can be regarded as an ϵ-approximate first-order stationary point. While the construction of ϵ being proportional to $\sqrt{1/N}$ is reasonable, the inclusion of an additional Big O term seems less comprehensible. It is my belief that the approximations exert a significant influence on convergence, which should not be overlooked.

3. **Bi-level approximation.** In Liu & Zhu's (2022), the bi-level optimization framework of ICRL is structured such that the upper-level problem was focused on reward learning, while the lower-level problem addressed constraint learning. In this current paper, the upper-level problem has been expanded to include the learning of meta-priors for both rewards and constraints ($\theta$ and $\omega$). This shift deviates from the original bi-level optimization framework of ICRL, raising some questions about its consistency. Moreover, the paper proposes that the task-specific reward adaptation is carried out via a gradient update. However, it is unclear whether this is a well-defined adaptation for specific tasks. There is a concern that this adaptation may be inadequate. It would be beneficial to provide additional supporting evidence. This could include referencing relevant papers where similar strategies have been successfully applied.

4. **The Audience of the current paper.** My final concern pertains to the intended audience of this paper. The manner in which the content is presented seems to diverge from the primary interests of the mainstream Reinforcement Learning (RL) and Machine Learning (ML) community. This paper primarily focuses on the approximation of computationally intractable gradients and the subsequent implications for convergence and generalization. While these advancements are valid, they may not be entirely consistent with the broader ML community. Furthermore, it's unclear how subsequent work can leverage and benefit from these methods. In my view, the style of this paper aligns more closely with the interests of the optimal control and robotics community. Subjectively speaking, ICLR may not be the most appropriate venue for this paper.

5. **Empirical Results** There are several concerns about the empirical results.
- The environmental parameters in the MuJoCo experiments raise some questions. The paper states (see Appendix B.2) that "The design of the reward function is such that the robots will receive a reward of +1 if they maintain the target velocity and a reward of 0 otherwise." and "For the constraint, the Swimmer experiment constrains all the states where the front tip angle exceeds a0, where a0 ∈ [0.9, 1.2]." These constraints do not appear to be significant. In particular, it's unclear why the agent would need to violate the constraint to maximize cumulative rewards, and whether maintaining the correct speed is a suitable reward for a MuJoCo task. More justification for these settings is required.

- The results for Meta-IRL are counterintuitive. One would expect that imitation learning methods like IRL, which do not model constraints, would yield higher reward and constraint violation rates. The presented results do not align with this understanding. Please provide an explanation.

- There's a considerable gap between the performance of the baselines and the expert performance. The lead of M-ICRL is substantial, suggesting that all the baselines fail in the task. Without careful design, the validity of these baselines could be called into question. More details or improvements on the baseline design are suggested.

**Questions:**

1. Line 4, Algorithm 1: It's unclear what the second $\mathcal{D}_i^{tr}$ in the gradient is intended for. It appears that the final parenthesis doesn't correspond to any elements in the algorithm.

2. Why does objective (1) incorporate a discounted log-likelihood? This suggests that the policy in later time steps has less impact on the likelihood objective. It raises the question: Would a Markov Decision Process (MDP) with a finite horizon be more consistent with the current objective?

3. The $\delta$ present in gradients (6) and (7) is not defined anywhere in the paper. It would be beneficial to provide an explicit definition or reference for this term.

---

> ### Author Response · Authors · 2023-11-17
> **Response to Reviewer Eje2**
>
> Thanks for your detailed and constructive reviews. We appreciate that you think this paper well-written and we believe that our discussion can lead to a better paper in general. Before addressing your comments, we would like to first discuss (i) the significant difference between RL and IRL, and thus why regret and sample complexity are not typically studied in IRL; (ii) the fundamental spirit of meta-learning.
>
> RL focuses on policy learning where the agent learns in an online style and requires sampling in the environment to collect the training data $(s,a,r)$. It is typical to use regret and sample complexity as the theoretical metric to analyze RL algorithms. However, IRL focuses on reward learning and the reward learning is a supervised learning problem where the training data is pre-collected and an offline optimization problem is formulated. Therefore, the regret is not a standard metric for IRL.
>
> Indeed, IRL also has a policy learning part, i.e, it usually needs to use RL to compute the policy corresponding to the learned reward function, and analyzing sample complexity is important to RL. However, IRL typically assumes the access to an RL oracle to get the policy corresponding to the learned reward function [D2,D4], so that how the policy is generated by the RL oracle is not the major concern in IRL. Therefore, it is not typical to use regret nor sample complexity to analyze IRL algorithms. In IRL, people focus on the reward learning part, i.e., how the offline optimization problem is solved. Therefore, analyzing the algorithm convergence is standard in IRL, either quantifying the loss function value or distance between the learned parameter and optimal parameter in convex cases [D2,D3] or quantifying the upper bound of the gradient norm in non-convex cases [D1,D5,D6]. We agree that evaluating the regret and sample complexity of the (RL) policy learning part in the IRL framework is interesting and significant, however, at current stage, the IRL theoretical studies focus more on the reward learning part instead of the (RL) policy learning part. In the future, we will explore more on the policy learning part.
>
> Now, we talk about the fundamental spirit of meta-learning. The standard supervised learning aims to learn good parameters from training data such that the model with the learned parameters can perform well on the test data. In contrast, meta-learning proposes to learn the ability to learn. It deals with multiple tasks. In specific, meta-learning wants to learn a meta-prior instead of specific parameters for specific tasks [D7,D8]. The meta-prior is learned from related tasks and is learned in the way that the task-specific adaptations adapted from the meta-prior can have good performance on the corresponding related tasks even if there is only limited data in the corresponding related tasks. Therefore, compared to standard supervised learning, meta-learning studies a different problem and requires special designs in order to learn the meta-priors.
>
> In our case, ICRL [D1] is just a supervised learning problem that optimizes for the reward and cost parameters for a single task. M-ICRL deals with multiple tasks. M-ICRL aims to learn the reward and cost meta-priors such that the task-specific reward and cost adaptations adapted from the reward and cost meta-priors can have good performance on specific tasks. Meta-learning [D7,D8,D9] usually has a bi-level learning structure where the upper level is to learn the meta-prior and the lower level aims to learn the task-specific adaptations. How to design the task-specific adaptation is important in meta-learning. A naive way is just solving the supervised learning problem on a specific task where the initialization is the meta-prior, and then use the result as the task-specific adaptation. However, this kind of design is problematic because we only have limited data to compute the task-specific adaptation, and fully solving the supervised learning problem can lead to overfitting. To solve this issue, the fundamental idea of meta-learning is to use regularization to avoid overfitting. Currently, there are two major types of designs. The first one, called MAML, is to compute the task-specific adaptation via one-step gradient descent [D7,D9]. This is how we compute the task-specific reward adaptation $\varphi_i=\theta-\alpha\frac{\partial}{\partial\theta}L_i(\theta,\eta_i^{\ast}(\theta,\omega))$. This method can be regarded as a regularization [D8] because it restricts the task-specific adaptation $\varphi_i$ from being far from the meta-prior $\theta$. The second one, called iMAML, is to directly use a regularization term [D8,D9]. This is how we compute the task-specific cost adaptation $\eta_i^{\ast}(\varphi_i,\omega)=\mathop{\arg\min}G_i(\eta;\varphi_i)+\frac{\lambda}{2}||\eta-\omega||^2$. Directly adding the regularization term usually has better empirical results than one-step gradient descent [D8], however,

---

> ### Author Response · Authors · 2023-11-17
> **Response to Reviewer Eje2 (continued)**
>
> it is more computationally expensive since it needs to solve a regularized optimization problem. Our problem formulation (2)-(3) embeds the spirit of both designs. Moreover, as mentioned in the last paragraph in Section 2.2, our design does not suffer the extra computation usually caused by iMAML because the original supervised learning problem (1) is already bi-level and we need to fully the lower-level problem anyway.
>
> In conclusion, meta-learning (M-ICRL) and supervised learning (ICRL) study totally different problems. Compared to ICRL, M-ICRL requires special designs in order to learn good meta-priors. In our work, we include special designs to solve meta-learning, and thus have a different structure from ICRL.
>
> We now address your comments.
>
> **Weakness 1**: [D1] has proposed a bi-level optimization objective for ICRL. This paper expands upon that proposal, adapting the objective to fit within a meta-learning context. The primary innovation is the integration of a meta-prior into the initial objective (see objective (2)). The algorithm presented outlines a direct, gradient-based optimization method, which, unfortunately, is computationally infeasible under most circumstances. The core focus of this paper is on mitigating this computational difficulty through the implementation of three levels of approximation. Additionally, it explores the impact of these approximations on both convergence and generalization. However, it should be noted that this study deviates somewhat from the original goal of Meta ICRL. Moreover, the advancements it provides over [D1] could be considered somewhat marginal.
>
> **Answer**: Thanks for mentioning the focus of this paper and the advancement over ICRL [D1].
>
> **For the comment on deviation from meta-learning**, we respectfully disagree with it. There are three major reasons.
>
> First, let us assume that we can obtain the exact hyper-gradient and ignore the approximation of the hyper-gradient. The other parts of this paper focus on the original goal of meta-learning. In specific, the original goal of meta-learning [D7,D8,D9], including meta-RL [D11], is to learn good meta-priors such that the corresponding task-specific adaptations can perform well on new tasks even if there is only limited data in new tasks. Along the same line, we extend the problem formulation of meta-learning to meta-ICRL using the bi-level optimization problem (2)-(3). More importantly, we theoretically analyze the convergence and generalization of the algorithm. The convergence and especially the generalization are the focus of meta-learning because the goal of meta-learning is to generalize to new tasks from limited data [D7,D8,D9]. In fact, these two properties have been widely studied in meta-learning [D8,D10,D12,D13], including meta-RL [D11].
>
> Second, the approximation part is essential to solve the problem (2)-(3) which models meta-ICRL. In specific, solving problem (2)-(3) requires to compute the hyper-gradient. However, it is challenging because the hyper-gradient includes the inverse-of-Hessian term and the gradient of the inverse-of-Hessian term (as detailed in Section 3.1). We propose novel approximation methods to address these challenges.
>
> Third, we would like to mention that recent papers [D8,D10,D11,D12,D13] perform theoretic analysis of meta-learning algorithms subject to approximation errors of the hyper-gradient caused by practical implementation. For example, [D10] studies the convergence of MAML [D7] and studies two methods to efficiently approximate the Hessian term in the hyper-gradient. It also quantifies the approximation error and convergence.
>
> **For the advancements over ICRL** [D1], we would like to notify that M-ICRL studies a totally different problem from ICRL, instead of advancing the methodology of ICRL. The focus of M-ICRL is the meta-learning part instead of the ICRL part. M-ICRL only utilizes ICRL as a task-specific solver, however, the focus of M-ICRL is how to learn good meta-priors and the generalization performance. In order to learn meta-priors, we propose special designs in (2)-(3) and these special designs introduce unique challenges in algorithm design and theoretical analysis. The unique challenges in algorithm design, as mentioned in Section 3.1, are the difficulties of approximating the hyper-gradients. The unique challenges in theoretical analysis have two parts. First, we study generalization while [D1] does not. Second, we need to quantify the approximation errors and their impact on the convergence while [D1] does not.

---

> ### Author Response · Authors · 2023-11-17
> **Response to Reviewer Eje2 (continued)**
>
> **Weakness 2**: Theorem 1 sets an upper bound on the expected gradient of the exact optimal solution. When compared to other more frequently used results, such as the upper bound of regret or the sample complexity in the PAC (Probably Approximately Correct) analysis, this result doesn't appear particularly robust.
>
> **Answer**: Thanks for mentioning the convergence. We agree that regret and sample complexity are typically analyzed in RL. However, as discussed above, current IRL theoretical studies focus on the reward learning part instead of the (RL) policy learning part. The reward learning is a supervised learning problem where the data is pre-collected and an offline optimization problem needs to be solved. Therefore, analyzing the algorithm convergence is standard in IRL, either quantifying the loss function value or distance between the learned parameter and optimal parameter in convex cases [D2,D3] or quantifying the upper bound of the gradient norm in non-convex cases [D1,D5,D6]. We agree that using PAC to analyze regret and sample complexity is significant and interesting for the (RL) policy learning part in the IRL framework, and we would like to explore more in the future.
>
> **Weakness 3**: It is surprising to note that the upper bound is raised to the power of $K$ (the number of gradient steps), rather than being linear or sublinear with respect to $K$. This implies that the approximation error could accumulate rapidly with each gradient step. Unfortunately, the paper does not provide a satisfactory explanation for this issue.
>
> **Answer**: The approximation error exponentially decreases rather than exponentially increases in our case. Note that $\frac{C_{\nabla^2_{\eta\eta}G}}{\lambda}\in (0,1)$, therefore the approximation error in Lemma 1 and Lemma 2 exponentially decreases when the number of $K$ increases, which is a virtue.
>
> **Weakness 4**: I question the assertion that this upper bound can be regarded as an $\epsilon$-approximate first-order stationary point. While the construction of $\epsilon$ being proportional to $\sqrt{1/N}$ is reasonable, the inclusion of an additional Big O term seems less comprehensible. It is my belief that the approximations exert a significant influence on convergence, which should not be overlooked.
>
> **Answer**: We agree that the approximation errors can exert a significant influence on convergence, however, this upper bound can be regarded as an $\epsilon$-approximate first-order stationary point for two reasons. First, all the terms in the big O can be arbitrarily small if we choose large enough $K$, $\bar{K}$, $D_i^{\text{tr}}$, and small enough $\delta$. Second, this is a standard practice in meta-learning convergence theory papers [D10,D11]. For example, [D10] also studies $\epsilon$-approximate first-order stationary point where the gradient norm is upper bounded by $\epsilon+O(\cdot)$ and the error terms in the big O can be arbitrarily small.
>
> **Weakness 5**: In [D1], the bi-level optimization framework of ICRL is structured such that the upper-level problem was focused on reward learning, while the lower-level problem addressed constraint learning. In this current paper, the upper-level problem has been expanded to include the learning of meta-priors for both rewards and constraints ($\theta$ and $\omega$). This shift deviates from the original bi-level optimization framework of ICRL, raising some questions about its consistency. Moreover, the paper proposes that the task-specific reward adaptation is carried out via a gradient update. However, it is unclear whether this is a well-defined adaptation for specific tasks. There is a concern that this adaptation may be inadequate. It would be beneficial to provide additional supporting evidence. This could include referencing relevant papers where similar strategies have been successfully applied.
>
> **Answer**: Thanks for mentioning the meta-prior and task-specific adaptation. We agree that there is a deviation from the original ICRL problem in terms of learning the cost meta-prior in the upper level, however, we still learn the task-specific cost adaptation $\eta_i^{\ast}$ in the lower level. In fact, this deviation is necessary and standard in meta-learning. Note that in our problem formulation (2)-(3), we learn the reward meta-prior $\theta$ and cost meta-prior $\omega$ in the upper level, and the task-specific cost adaptation $\eta_i^{\ast}$ in the lower-level. This is exactly the standard practice of meta-learning [D7,D8,D9] where the upper level is to learn the meta-prior and the lower level is to learn the task-specific adaptations. Here, we would like to emphasize again that meta-learning (M-ICRL) and supervised learning (ICRL) study different problems and thus it is expected that meta-learning problem has special designs.
>
> The task-specific reward adaptation $\varphi_i$ is computed via one-step gradient descent from the reward meta-prior $\theta$. The intuition behind this is

---

> ### Author Response · Authors · 2023-11-17
> **Response to Reviewer Eje2 (continued)**
>
> to avoid overfitting using early stop as a regularization (as detailed at the beginning of this response, before answering Weakness 1). This style of computing the task-specific adaptation is called MAML [D7]. It is first introduced in [D7] and then widely adopted in meta-RL [D11] and meta-IRL [D14,D15]. The convergence [D10] and generalization [D12] of MAML are both well studied. It is actually one of the two predominant meta-learning methods. The other one is iMAML [D8].
>
> **Weakness 6**: My final concern pertains to the intended audience of this paper. The manner in which the content is presented seems to diverge from the primary interests of the mainstream Reinforcement Learning (RL) and Machine Learning (ML) community. This paper primarily focuses on the approximation of computationally intractable gradients and the subsequent implications for convergence and generalization. While these advancements are valid, they may not be entirely consistent with the broader ML community. Furthermore, it's unclear how subsequent work can leverage and benefit from these methods. In my view, the style of this paper aligns more closely with the interests of the optimal control and robotics community. Subjectively speaking, ICLR may not be the most appropriate venue for this paper.
>
> **Answer**: Thanks for mentioning the intended audience of this paper. We believe that the paper studies the core problems of meta-learning. Please refer to our answer to the comment on deviation from meta-learning in weakness 1. Here we would like to further mention that the approximation of hyper-gradient is consistent with the interest in many other machine learning problems.
>
> In the IRL community, [D1,D5] explicitly formulate the IRL problem as a bi-level optimization problem and they all propose novel approximation methods to efficiently approximate the hyper-gradient (see the "local gradient approximation" in [D1] and Lemma 1 in [D5]). Since IRL usually has a bi-level learning structure where the upper level learns the reward function and the lower level learns the corresponding policy, it is of interest to efficiently compute the hyper-gradient (i.e., gradient of the upper-level problem) to update the learned reward function.
>
> In the bi-level optimization community [D21,D22,D23], approximating the hyper-gradient is one of the fundamental research interests since the hyper-gradient is needed to solve the upper-level problem and is difficult to compute in general. For example, the contribution of [D22] is "comprehensive convergence rate analysis for two popular algorithms
> respectively based on approximate implicit differentiation (AID) and iterative differentiation (ITD)" where AID and ITD are two methods to approximate the hyper-gradient.
>
> Moreover, other works can build on our paper since they can use some of our proposed approximation methods. For example, in our paper, we propose to solve an optimization problem in order to approximate the inverse-of-Hessian term. iMAML also needs to approximate an inverse-of-Hessian term, however, they just assume that they can approximate the inverse-of-Hessian term with $\delta$-accuracy without theoretical guarantees. In contrast, our work provides a principled way with theoretical guarantees to ensure that the approximation error can be arbitrarily small (proved in Lemma 1).
>
> **Weakness 7**: The environmental parameters in the MuJoCo experiments raise some questions. The paper states (see Appendix B.2) that "The design of the reward function is such that the robots will receive a reward of +1 if they maintain the target velocity and a reward of 0 otherwise." and "For the constraint, the Swimmer experiment constrains all the states where the front tip angle exceeds $a_0$, where $a_0 \in [0.9, 1.2]$." These constraints do not appear to be significant. In particular, it's unclear why the agent would need to violate the constraint to maximize cumulative rewards, and whether maintaining the correct speed is a suitable reward for a MuJoCo task. More justification for these settings is required.
>
> **Answer**: Thanks for mentioning the design of reward and constraint. For the design of reward, the motivation of maintaining the target velocity is that it has wide applications in the real world, such as autonomous vehicle driving at a constant speed [D19] and robotic dog skating at a constant velocity [D20].  Moreover, maintaining target velocity in Mujoco is a standard benchmark in meta-RL [D7,D16,D17] and meta-IRL [D18]. Since meta-learning needs multiple related tasks, maintaining different target velocity can be considered as multiple related task.
>
> For the design of the constraint in the Swimmer case, we use this kind of constraints because we empirically find that the angle of Swimmer's front tip will typically exceed 1.2 if there is no constraint. We choose this kind of constraint to show that our algorithm can indeed learn the constraint and avoid violating the constraint.

---

> > ### Author Response · Authors · 2023-11-17
> > **Response to Reviewer Eje2 (continued)**
> >
> > **Weakness 8**: The results for Meta-IRL are counterintuitive. One would expect that imitation learning methods like IRL, which do not model constraints, would yield higher reward and constraint violation rates. The presented results do not align with this understanding. Please provide an explanation.
> >
> > **Answer**: Meta-IRL in our experiment yields higher constraint violation rate (CVR) and lower cumulative reward (CR). It is intuitive that Meta-IRL has higher constraint violation because it does not model constraints. It has lower reward because, as mentioned in the last paragraph in Appendix B.2, the bad CVR performance will also result in bad performance of CR even if Meta-IRL may estimate the ground truth reward function well, because the episode will terminate if any constraint is violated. Since the trajectory is truncated if the constraint is violated, it has smaller cumulative reward as it has shorter length to accumulate reward.
> >
> > **Weakness 9**: There's a considerable gap between the performance of the baselines and the expert performance. The lead of M-ICRL is substantial, suggesting that all the baselines fail in the task. Without careful design, the validity of these baselines could be called into question. More details or improvements on the baseline design are suggested.
> >
> > **Answer**: Thanks for mentioning the design of the baselines. We introduce the four baselines in detail in the first paragraph of Appendix B. Here we would like to further discuss the details and improvement. The baseline ICRL is just running the algorithm in [D1] on one demonstration. It has bad performance because it lacks data and thus suffers from overfitting. To improve this baseline, we can either use more training data or tune a regularization term to avoid overfitting. The baseline Meta-IRL directly uses the algorithm in [D14]. To improve its performance, one way is to add certain constraint signal to augment the learned reward function. The constraint signal includes certain information about the constraint so that it can help Meta-IRL to achieve better CVR performance and thus CR performance. The baseline ICLR(pre) naively learns meta-priors across all demonstrations of all the training tasks. In specific, ICRL(pre) first solves $\min_{\theta}\frac{1}{m}\sum_{i=1}^mL_i(\theta,\omega^{\ast}(\theta),D_i), \text{s.t.}\omega^{\ast}(\theta)=\arg\min_{\omega}\frac{1}{m}\sum_{i=1}^mG_i(\omega;\theta,D_i)$ where $D_i\triangleq${$D_i^{\text{tr}},D_i^{\text{eval}},D_i^{\text{h}}$}. The obtained results of this problem (i.e., $\theta$ and $\omega^{\ast}(\theta)$) are the meta-priors of ICRL(pre). ICRL(pre) then uses these meta-priors as initializations to solve the problem $\min_{\theta}L_{m+1}(\theta,\omega^{\ast}(\theta),D_{m+1}),\text{s.t.}\omega^{\ast}(\theta)=\arg\min_{\omega}G_{m+1}(\omega;\theta,D_{m+1})$ for an arbitrary new task $\mathcal{T}_{m+1}$. To improve the performance of ICRL(pre), we can make the distance between different tasks smaller. The distance between tasks is defined in Section 4.2.
> >
> > **Q1**: Line 4, Algorithm 1: It's unclear what the second $\mathcal{D}_i^{\text{tr}}$ in the gradient is intended for. It appears that the final parenthesis doesn't correspond to any elements in the algorithm.
> >
> > **Answer**: In $\frac{\partial}{\partial \theta}\hat{L}_i(\theta(n),\hat{\eta}_i(\theta(n),\omega(n),\mathcal{D}_i^{\text{tr}},K),\mathcal{D}_i^{\text{tr}})$, the first $\mathcal{D}_i^{\text{tr}}$ is to compute the task specific adaptation $\hat{\eta}_i(\theta(n),\omega(n),\mathcal{D}_i^{\text{tr}},K)$ and the second $\mathcal{D}_i^{\text{tr}}$ is the data set used to formulate the function $\hat{L}_i(\cdot,\cdot,\mathcal{D}_i^{\text{tr}})$.
> >
> > **Q2**: Why does objective (1) incorporate a discounted log-likelihood? This suggests that the policy in later time steps has less impact on the likelihood objective. It raises the question: Would a Markov Decision Process (MDP) with a finite horizon be more consistent with the current objective?
> >
> > **Answer**: The reason that the objective (1) uses a discounted log-likelihood is that we study infinite horizon MDP and the log-likelihood may be infinite without the discount factor $\gamma$. It is interesting to study finite horizon MDPs. Intuitively, if the problem aimed to be solved has a finite horizon, finite horizon MDP is more suitable.
> >
> > **Q3**: The $\delta$ present in gradients (6) and (7) is not defined anywhere in the paper. It would be beneficial to provide an explicit definition or reference for this term.
> >
> > **Answer**: Thanks for notifying this. We already add the explanation of $\delta$ in the newly uploaded version. The $\delta$ is just a constant, which is the magnitude of perturbation.

---

> > > ### Author Response · Authors · 2023-11-17
> > > **Response to Reviewer Eje2 (continued)**
> > >
> > > **References**
> > >
> > > [D1] Shicheng Liu and Minghui Zhu. "Distributed inverse constrained reinforcement learning for multi-agent systems.“ In Advances in Neural Information Processing Systems, pp. 33444–33456, 2022.
> > >
> > > [D2] Pieter Abbeel and Andrew Y. Ng, "Apprenticeship learning via inverse reinforcement learning," International conference on Machine Learning, pp. 1--8, 2004.
> > >
> > > [D3] Nathan D. Ratliff, J. Andrew Bagnell, and Martin A. Zinkevich. "Maximum margin planning." In Proceedings of the 23rd international conference on Machine learning, pp. 729-736. 2006.
> > >
> > > [D4] Saurabh Arora, and Prashant Doshi, ``A survey of inverse reinforcement learning: Challenges, methods and progress," Artificial Intelligence, vol. 297, p. 103500, 2021.
> > >
> > > [D5] Siliang Zeng, Chenliang Li, Alfredo Garcia, and Mingyi Hong. "Maximum-likelihood inverse reinforcement learning with finite-time guarantees." Advances in Neural Information Processing Systems, pp. 10122-10135, 2022.
> > >
> > > [D6] Feiyang Wu, Jingyang Ke, and Anqi Wu. "Inverse Reinforcement Learning with the Average Reward Criterion." arXiv preprint arXiv:2305.14608, 2023.
> > >
> > > [D7] Chelsea Finn, Pieter Abbeel, and Sergey Levine. "Model-agnostic meta-learning for fast adaptation of deep networks." In International Conference on Machine Learning, pp. 1126–1135, 2017a.
> > >
> > > [D8] Aravind Rajeswaran, Chelsea Finn, Sham M Kakade, and Sergey Levine. "Meta-learning with implicit gradients.“ In Advances in Neural Information Processing Systems, pp. 113–124, 2019.
> > >
> > > [D9] Maria-Florina Balcan, Mikhail Khodak, and Ameet Talwalkar. "Provable guarantees for gradient-based meta-learning." In International Conference on Machine Learning, pp. 424-433, 2019.
> > >
> > > [D10] Alireza Fallah, Aryan Mokhtari, and Asuman Ozdaglar. "On the convergence theory of gradient-based model-agnostic meta-learning algorithms." In International Conference on Artificial Intelligence and Statistics, pp. 1082–1092, 2020.
> > >
> > > [D11] Alireza Fallah, Kristian Georgiev, Aryan Mokhtari, and Asuman Ozdaglar. "On the convergence theory of debiased model-agnostic meta-reinforcement learning." In Advances in Neural Information Processing Systems, pp. 3096–3107, 2021a.
> > >
> > > [D12] Alireza Fallah, Aryan Mokhtari, and Asuman Ozdaglar. "Generalization of model-agnostic meta-learning algorithms: Recurring and unseen tasks." In Advances in Neural Information Processing Systems, pp. 5469–5480, 2021b.
> > >
> > > [D13] Giulia Denevi, Carlo Ciliberto, Riccardo Grazzi, and Massimiliano Pontil. "Learning-to-learn stochastic gradient descent with biased regularization." In International Conference on Machine Learning, pp. 1566–1575, 2019.
> > >
> > > [D14] Kelvin Xu, Ellis Ratner, Anca Dragan, Sergey Levine, and Chelsea Finn. "Learning a prior over intent via meta-inverse reinforcement learning.” In International Conference on Machine Learning, pp. 6952–6962, 2019.
> > >
> > > [D15] Chelsea Finn,Tianhe Yu, Tianhao Zhang, Pieter Abbeel, and Sergey Levine. "One-shot visual imitation learning via meta-learning." In Conference on robot learning, pp. 357-368, 2017b.
> > >
> > > [D16] Zichuan Lin, Garrett Thomas, Guangwen Yang, and Tengyu Ma. "Model-based adversarial meta-reinforcement learning." In Advances in Neural Information Processing Systems, pp. 10161-10173, 2020.
> > >
> > > [D17] Kate Rakelly, Aurick Zhou, Chelsea Finn, Sergey Levine, and Deirdre Quillen. "Efficient off-policy meta-reinforcement learning via probabilistic context variables." In International Conference on Machine Learning, pp. 5331-5340, 2019.
> > >
> > > [D18] Seyed Kamyar Seyed Ghasemipour, Shixiang Gu, and Richard Zemel. "SMILe: scalable meta inverse reinforcement learning through context-conditional policies." In Proceedings of the 33rd International Conference on Neural Information Processing Systems, pp. 7881-7891. 2019.
> > >
> > > [D19] Ugo Rosolia, Stijn De Bruyne, and Andrew G. Alleyne. "Autonomous vehicle control: A nonconvex approach for obstacle avoidance." IEEE Transactions on Control Systems Technology 25, no. 2, pp. 469-484, 2016.
> > >
> > > [D20] Vivek Jagannath, Sahil Sanil, Prabhat Kumar, Aman Malhotra, J. Vighneswar, Advay S. Pethakar, and M. Sangeetha. "Locomotion and Path Planning for Roller Skating Dog Robot." In International Conference on Computing for Sustainable Global Development, pp. 681-684, 2021.
> > >
> > > [D21] Mingyi Hong, Hoi-To Wai, Zhaoran Wang, and Zhuoran Yang. "A two-timescale stochastic algorithm framework for bilevel optimization: Complexity analysis and application to actor-critic." SIAM Journal on Optimization 33, no. 1, pp. 147-180, 2023.
> > >
> > > [D22] Kaiyi Ji, Junjie Yang, and Yingbin Liang. "Bilevel optimization: Convergence analysis and enhanced design." In International conference on machine learning, pp. 4882-4892. PMLR, 2021.
> > >
> > > [D23] Risheng Liu, Jiaxin Gao, Jin Zhang, Deyu Meng, and Zhouchen Lin. "Investigating bi-level optimization for learning and vision from a unified perspective: A survey and beyond." IEEE Transactions on Pattern Analysis and Machine Intelligence 44, no. 12, pp. 10045-10067, 2021.

---

> ### Comment · Reviewer_Eje2 · 2023-11-20
>
> Thanks for responding. I am indeed surprised by the length of the response. It looks nice, but It takes me a while to read these paragraphs. Some of concerns are resolved, and my further questions are:
>
> - When I mentioned "However, it should be noted that this study deviates somewhat from the original goal of Meta ICRL", when I saw the title, I was expecting to see an advancement of ICRL. but from the paper and the response, most of the advancement has nothing to do with the ICRL, but about how to resolve the meta-prior's computational difficulty with three levels of approximation.
>
> - Some of the concerns (Weakness 4 and 5) about the paper are interpreted as standard practice in Meta-RL. Since my expertise lies in RL rather than meta-learning, I find this somewhat surprising, but I'm unable to comment further on it. From my perspective, this paper primarily focuses on extending the meta-Learning strategies to ICRL, rather than on how to solve ICRL problems within the context of meta-learning.
>
> - I am not sure about the responses "current IRL theoretical studies focus on the reward learning part instead of the (RL) policy learning part" and "The reward learning is a supervised learning problem where the data is pre-collected and an offline optimization problem needs to be solved.". Please refer to the following theoretical works about IRL. This line of works all talk about sample complexity or regrets.
>
> Metelli, A. M., Ramponi, G., Concetti, A., and Restelli, M. "Provably efficient learning of transferable rewards". In International Conference on Machine Learning (ICML), 2021.
>
> Lindner, David, Andreas Krause, and Giorgia Ramponi. "Active exploration for inverse reinforcement learning." Advances in Neural Information Processing Systems (NeurIPS) 2022
>
> Metelli, Alberto Maria, Filippo Lazzati, and Marcello Restelli. "Towards Theoretical Understanding of Inverse Reinforcement Learning." In International Conference on Machine Learning (ICML), 2023.
>
>
> - I have a further concern regarding the implementation. Notably, the authors of DICRL have decided not to release their code, leading to subsequent works encountering difficulties when comparing with them, which was disappointing to me for a while. I'm interested to know how you managed to compare your work with DICRL. Did you implement their algorithm independently? If so, could you provide guidance on how to locate this implementation?
>
> - Specifically, within the submitted code, it appears that the algorithm depends on SAC, which was not mentioned in the paper. Additionally, the learn.py function does not seem to include the cost inference component.
>
> - Furthermore, do the authors intend to make their code publicly available in the future? I'm raising this question because, once your paper is published, subsequent studies are likely to be required to compare their methods with yours, particularly since your paper introduces an algorithm rather than proposing a theory. Given the complexity of your paper, I foresee that most followers may encounter difficulties in implementing your method.

---

> > ### Author Response · Authors · 2023-11-20
> >
> > Thanks for your reply. We highly appreciate your feedback and the valuable references you provide. Indeed, we study the problem of M-ICRL from the angle of meta-learning where we only use ICRL as a task-specific solver and we primarily focus on dealing with the meta-learning part. Just as you have mentioned, we do not aim to advance ICRL. Instead, we aim to advance meta-learning to the context of ICRL since meta-learning will meet many new challenges when advanced to the setting of ICRL. In fact, this is a standard practice in meta-learning. For example, Meta-IRL [D14] directly uses maximum entropy IRL [D24] as the task-specific solver and focuses on meta-learning, Meta constrained RL [D25] directly uses CRPO [D26] as the constrained RL solver and focuses on the design of meta-learning. We now address your other comments.
> >
> > **Q1**: I am not sure about the responses "current IRL theoretical studies focus on the reward learning part instead of the (RL) policy learning part" and "The reward learning is a supervised learning problem where the data is pre-collected and an offline optimization problem needs to be solved.". Please refer to the following theoretical works about IRL. This line of works [D27,D28,D29] all talk about sample complexity or regrets.
> >
> > **Answer**: Thanks for providing these valuable references. We agree that these IRL works focus on the policy learning part and thus they study the regret and sample complexity. However, as mentioned at the beginning of this response, our focus is on the meta-learning part and we only treat ICRL as an existing oracle. Therefore, we focus on the properties of the meta-learning algorithm instead of the propertis inside the ICRL oracle. We agree that studying the regret and sample complexity of the policy learning part is significant to IRL research, and we would like to explore this in the future.
> >
> > Given that we have strong contributions to the meta-learning part, i.e., we are the first to study the meta-ICRL problem, we propose novel algorithm designs to solve the unique challenges of this problem, and provide convergence and (more importantly) generalization analysis of the algorithm, this paper is strong enough for a publication.
> >
> > **Q2**: I have a further concern regarding the implementation. Notably, the authors of DICRL have decided not to release their code, leading to subsequent works encountering difficulties when comparing with them, which was disappointing to me for a while. I'm interested to know how you managed to compare your work with DICRL. Did you implement their algorithm independently? If so, could you provide guidance on how to locate this implementation?
> >
> > **Answer**: The authors of DICRL actually released the code. We can find the code in their supplementary folder (https://proceedings.neurips.cc/paper_files/paper/2022/hash/d842425e4bf79ba039352da0f658a906-Abstract-Conference.html). This paper directly uses their code.
> >
> >
> > **Q3**: Specifically, within the submitted code, it appears that the algorithm depends on SAC, which was not mentioned in the paper. Additionally, the learn.py function does not seem to include the cost inference component.
> >
> > **Answer**: We mentioned using SAC in the last line of Appendix A.2. Since we treat ICRL as an oracle and ICRL treats SAC as an RL oracle, we use SAC as an RL oracle. In the newly updated zip file, we learn both reward function and cost function.
> >
> > **Q4**: Furthermore, do the authors intend to make their code publicly available in the future? I'm raising this question because, once your paper is published, subsequent studies are likely to be required to compare their methods with yours, particularly since your paper introduces an algorithm rather than proposing a theory. Given the complexity of your paper, I foresee that most followers may encounter difficulties in implementing your method.
> >
> > **Answer**: We will make the code publicly if this paper is accepted. Here we would like to mention again that this paper is strong enough for a publication since it has significant contribution where we are the first to study the M-ICRL problem and we provide novel algorithm design and threotical (both convergence and generalization) analysis.

---

> > > ### Author Response · Authors · 2023-11-20
> > >
> > > **References**
> > >
> > > [D24] Brian D. Ziebart, Andrew Maas, J. Andrew Bagnell, and Anind K. Dey. "Maximum entropy inverse reinforcement learning." In National Conference on Artificial intelligence, pp. 1433-1438. 2008.
> > >
> > > [D25] Khattar, Vanshaj, Yuhao Ding, Bilgehan Sel, Javad Lavaei, and Ming Jin. "A CMDP-within-online framework for Meta-Safe Reinforcement Learning." In International Conference on Learning Representations, 2022.
> > >
> > > [D26] Tengyu Xu, Yingbin Liang, and Guanghui Lan. "CRPO: A new approach for safe reinforcement learning with convergence guarantee." In International Conference on Machine Learning, pp. 11480-11491, 2021.
> > >
> > > [D27] Metelli, A. M., Ramponi, G., Concetti, A., and Restelli, M. "Provably efficient learning of transferable rewards". In International Conference on Machine Learning, 2021.
> > >
> > > [D28] Lindner, David, Andreas Krause, and Giorgia Ramponi. "Active exploration for inverse reinforcement learning." Advances in Neural Information Processing Systems, 2022
> > >
> > > [D29] Metelli, Alberto Maria, Filippo Lazzati, and Marcello Restelli. "Towards Theoretical Understanding of Inverse Reinforcement Learning." In International Conference on Machine Learning, 2023.

---

> > > > ### Author Response · Authors · 2023-11-22
> > > >
> > > > Dear reviewer,
> > > >
> > > > We have answered all your questions. If you have any further question, we are willing to discuss.
> > > >
> > > > Best,
> > > >
> > > > Authors.

---

### Official Review · Reviewer_byVV · 2023-10-30

**Soundness:** 3 good
**Presentation:** 3 good
**Contribution:** 3 good
**Rating:** 6
**Confidence:** 3

**Summary:**

The paper studies the meta inverse reinforcement learning problems in the constrained MDP. The paper proposes an approach that first learn meta-priors over reward functions and constrains from related tasks and then adapt the learned prior to new tasks with few expert demonstration. The problem is formulated as a bi-level optimization problem, where the upper level learns the prior on the reward functions and the lower level learns the prior on the constraints. The paper shows that the algorithm reaches the $\epsilon$-station points at $O(1\epsilon^2)$ and quantify the generalization error to new tasks. The theoretical results are supported empirically.

**Strengths:**

- The paper proposes a theoretical framework for  meta inverse constrained reinforcement learning, which extends previous study on meta
 inverse reinforcement learning and inverse constrained reinforcement learning.
- The theoretical study is solid. The paper shows the convergence guarantee of the proposed algorithm. Then, the paper studies the generalization error for a new arbitrary task.
- The paper provides empirical study on the algorithm in two settings: navigation with obstacles in AR and mojoco experiments, showing that M-ICRL performs better than other methods.
- The paper provides a clear presentation on its study, with a detailed discussion on the challenges and the approach.

**Weaknesses:**

- The paper is an extension of meta IRL and ICRL. Building upon these, the contribution of the study is not significant.
- The paper only shows the convergence to $\epsilon$-FOSP, whose optimality is not discussed. Convergence of gradient-based algorithms to stationary point is not a novel contribution.
- Assumption 1 assumes that the reward function and the cost function are bounded and smooth up to the fourth order gradient, which is a strong assumption, especially for neural networks with ReLU activation and unbounded state-action space.

**Questions:**

See weaknesses for details.

---

> ### Author Response · Authors · 2023-11-17
> **Response to Reviewer byVV**
>
> Thanks for your constructive reviews. We believe that our discussion can improve this paper in general. We address your comments below:
>
> **Weakness 1**: The paper is an extension of meta IRL and ICRL. Building upon these, the contribution of the study is not significant.
>
> **Answer**: Thanks for mentioning the novelty. Though our problem combines ICRL in [C1] and meta-learning in [C2,C4], our algorithm design and theoretic analysis are substantially different from those in the three references. We discuss the challenges of algorithm design in Section 3.1 and the challenges of theoretical analysis in the first paragraph of Section 4.1. Here we would like to further discuss our unique challenges and novelties in algorithm design and theoretical analysis compared to [C1,C2,C4].
>
> **Algorithm design**. [C1,C2,C4] and our work all formulate a bi-level optimization problem. Note that [C2] uses MAML [C3] and MAML can be regarded as a bi-level optimization problem where the lower level is one-step gradient descent from the meta-prior and the upper level aims to optimize for the meta-prior [C4]. A critical step of solving bi-level optimization problems is to compute the hyper-gradient, i.e., the gradient of the upper-level problem. The methods of computing the hyper-gradient in [C1,C2,C4] **CANNOT** be applied to our case and thus novel algorithm designs for computing the hyper-gradient are needed. In specific, there are two unique challenges in terms of algorithm design.
>
> First, our hyper-gradient includes the term $\frac{\partial^2}{\partial\theta^2}L_i(\theta,\eta_i^{\ast}(\theta,\omega))$ while [C1,C2,C4] do not have this term. As mentioned in challenge (i) in Section 3.1, computing $\frac{\partial^2}{\partial\theta^2}L_i(\theta,\eta_i^{\ast}(\theta,\omega))$ requires to compute the gradient of an inverse-of-Hessian term $[\nabla_{\eta\eta}^2G_i(\eta_i^{\ast}(\varphi_i,\omega);\varphi_i)+\lambda I]^{-1}$. Since we use neural networks as the parameterized model, computing this inverse-of-Hessian term is already challenging enough and now we need to compute the gradient of this inverse-of-Hessian term, which is intractable. To solve this issue, we propose a novel approximation design where we only need to compute first-order gradients (see (6)-(7) and Algorithm 2) and we can prove that the approximation error can be arbitrarily small (Lemma 2).
>
> Second, as mentioned in challenge (ii) in Section 3.1, our hyper-gradient includes the inverse-of-Hessian term $[\nabla_{\eta\eta}^2G_i(\eta_i^{\ast}(\varphi_i,\omega);\varphi_i)+\lambda I]^{-1}$ while [C1,C2] do not have this issue. [C4] also has this issue, however, they just use existing conjugate gradient method and assume that a good approximation can be obtained. In contrast, we propose to solve an optimization problem (10) in Algorithm 3 to approximate and more importantly, we theoretically guarantee in Lemma 1 that the approximation error can be arbitrarily small.
>
> **Theoretical analysis**. We study **BOTH** the convergence and generalization of our algorithm while [C2] do not have theoretical analysis and [C1,C4] only study convergence. Therefore, the whole Section 4.2 is unique and novel compared to [C1,C2,C4] because it studies generalization. Even for convergence, our analysis is novel compared to [C1,C4]. In specific, since we have the unique challenges for algorithm design and we provide novel algorithm designs (i.e., approximation methods) to tackle those unique challenges, we need corresponding theoretical analysis to guarantee the performance of our unique algorithm designs, i.e., Lemma 1 and Lemma 2 quantifies the approximation error of our algorithm designs.

---

> ### Author Response · Authors · 2023-11-17
> **Response to Reviewer byVV (continued)**
>
> **Weakness 2**: The paper only shows the convergence to $\epsilon$-FOSP, whose optimality is not discussed. Convergence of gradient-based algorithms to stationary point is not a novel contribution.
>
> **Answer**: Thanks for mentioning the convergence. We agree that convergence to global optimal solutions is very interesting, however, it is quite challenging in our case because our problem is highly non-convex. Note that our problem is still non-convex even if we use linear parameterized models instead of neural networks [C1]. We notice that some bi-level optimization works [C9,C10] have studied the convergence to the global optimal solutions, however, they all impose strong assumptions, e.g., the upper-level objective function is either convex or strongly convex. These assumptions are not satisfied in our case.
>
> In fact, the convergence to $\epsilon$-FOSP is not trivial in our case. Convergence of SGD to stationary points has been widely studied, however, our novelty is beyond the convergence of SGD. As mentioned in the first paragraph of Section 4.1, compared to the standard SGD, the main difficulty of guaranteeing the convergence of Algorithm 1 lies in quantifying the approximation error of the hyper-gradients and how the errors affect the convergence. Note that we do not have the exact hyper-gradient, thus our algorithm is not standard SGD. The most difficult part of solving problem (2)-(3) is how to approximate the hyper-gradient and how to ensure that the approximation error does not ruin the convergence. Therefore, our major contribution of convergence is to design novel approximation methods in Section 3.2 and theoretically guarantee that the proposed approximation method can lead to convergence in Section 4.1. Besides convergence, we also study the generalization in Section 4.2.
>
> **Weakness 3**: Assumption 1 assumes that the reward function and the cost function are bounded and smooth up to the fourth order gradient, which is a strong assumption, especially for neural networks with ReLU activation and unbounded state-action space.
>
> **Answer**: Thanks for mentioning the assumption. We agree that these assumptions could be strong when the neural networks use ReLU activation. However, the smoothness assumption can be satisfied by using some smooth activation functions such as tanh and using linear parameterized models. Moreover, as mentioned in the paragraph under Assumption 1, these assumptions are typical assumptions in RL [C5], IRL [C6,C8] and meta-RL [C7]. It is interesting to relax these assumptions, and we will explore it in the future.
>
> **References**
>
> [C1] Shicheng Liu and Minghui Zhu. "Distributed inverse constrained reinforcement learning for multi-agent systems.“ In Advances in Neural Information Processing Systems, pp. 33444–33456, 2022.
>
> [C2] Kelvin Xu, Ellis Ratner, Anca Dragan, Sergey Levine, and Chelsea Finn. "Learning a prior over intent via meta-inverse reinforcement learning.” In International Conference on Machine Learning, pp. 6952–6962, 2019.
>
> [C3] Chelsea Finn, Pieter Abbeel, and Sergey Levine. "Model-agnostic meta-learning for fast adaptation of deep networks." In International Conference on Machine Learning, pp. 1126–1135, 2017.
>
> [C4] Aravind Rajeswaran, Chelsea Finn, Sham M Kakade, and Sergey Levine. "Meta-learning with implicit gradients.“ In Advances in Neural Information Processing Systems, pp. 113–124, 2019.
>
> [C5] Kaiqing Zhang ,Alec Koppel, Hao Zhu, and Tamer Basar. "Global convergence of policy gradient methods t o(almost) locally optimal policies." SIAM Journal on Control and Optimization, 58(6): 3586–3612, 2020.
>
> [C6] Ziwei Guan, Tengyu Xu, and Yingbin Liang. "When will generative adversarial imitation learning algorithms attain global convergence." In International Conference on Artificial Intelligence and Statistics, pp. 1117–1125, 2021.
>
> [C7] Alireza Fallah, Kristian Georgiev, Aryan Mokhtari, and Asuman Ozdaglar. "On the convergence theory of debiased model-agnostic meta-reinforcement learning." In Advances in Neural Information Processing Systems, pp. 3096–3107, 2021.
>
> [C8] Siliang Zeng, Chenliang Li, Alfredo Garcia, and Mingyi Hong. "Maximum-likelihood inverse reinforcement learning with finite-time guarantees." Advances in Neural Information Processing Systems, pp. 10122-10135, 2022.
>
> [C9] Saeed Ghadimi, and Mengdi Wang. "Approximation methods for bilevel programming." arXiv preprint arXiv:1802.02246, 2018.
>
> [C10] Mingyi Hong, Hoi-To Wai, Zhaoran Wang, and Zhuoran Yang. "A two-timescale stochastic algorithm framework for bilevel optimization: Complexity analysis and application to actor-critic." SIAM Journal on Optimization 33, no. 1, pp. 147-180, 2023.

---

### Official Review · Reviewer_NXcx · 2023-11-02

**Soundness:** 3 good
**Presentation:** 3 good
**Contribution:** 3 good
**Rating:** 6
**Confidence:** 3

**Summary:**

In this work, the authors optimize the ability of inverse constrained reinforcement learning (ICRL) to learn the reward and constraint(s) for a new task by meta-learning reward and constraint function priors over multiple similar tasks. The ICRL problem is formulated as a bi-level optimization, as proposed by [1], and then the authors provide a way to meta-learn the priors efficiently through empirical approximations and iterative techniques. ICRL is a growing field, and being able to perform ICRL more efficiently across new tasks is definitely relevant and useful. The proposed approach could be slightly mis-formulated (as explained later in weaknesses), but is theoretically well-analyzed and has promising experimental results.

**Strengths:**

1. Extensive theoretical analysis about the approximation errors and (approximate) convergence result
2. The baselines clearly demonstrate the effectiveness of the approach since the proposed method can learn a good meta-prior over several tasks, and thus performs well when tested later
3. Successful real world experiments

**Weaknesses:**

1. The lower level optimization (as explained in Equation 11, Appendix A.1) maximizes the reward and causal entropy objective while matching the constraint feature expectations. On the other hand, the authors of [1] formulate the optimization as a maximization of the causal entropy while matching reward feature expectations and expected cumulative costs. The slight difference is that [1] has an additional Lagrange multiplier term $\lambda$ in the dual formulation (see [1], page 4, below remark 2, in the definition of $G$). In this work, since the constraint feature expectations are directly matched, the overall form of the lower level objective $G$ (Appendix, equation 12) is not exactly the same as [1]. This is also apparent in the constrained soft Bellman policy definition adapted from [1] (no $\lambda$ in the constrained policy for this work, whereas there is a $\lambda$ in the constrained policy as described in [1], page 2, Appendix). The outcome is that the constraint function is just treated as a negative reward term that can be just added to the original reward and thus, constrained RL just amounts to running RL with reward $r-c$ (Appendix A.2 of this work also says this). I am not sure this is representative of typical constrained RL problems, since typically it is not possible to rewrite a constrained RL problem as an unconstrained RL problem with a different reward. (maybe the authors can elaborate on this)
2. What if several demonstrations are available for each task? If more demonstrations are available, ICRL could perform better and the gap between M-ICRL & ICRL could be lesser. M-ICRL should still perform better, since it has a better meta-prior, but overall it would be useful to understand the empirical improvement of M-ICRL over ICRL as the number of demonstrations vary.

**Questions:**

1. While Equation 12 (Appendix A.1) is the dual of Equation 11, if the domain is extended from linear constraint functions to non-linear constraint functions, the equation would no longer behave as the dual of the original problem as formulated in Equation 11, right? Does it make sense to use this as the lower level problem, in that case?
2. (Suggestion) more ablations (eg. empirical effect of $\alpha$, batch size $B$, number of gradient descent steps $K$, etc.) can be performed. Also, are these values specified somewhere in the paper?
3. (Suggestion) Notation can be slightly confusing at some places, so I would suggest mentioning in the algorithms what the inputs and the outputs are, in words (eg. meta-priors, etc.). Implicit hyperparameters should also be mentioned in the beginning of the algorithm, eg. $\alpha$, $B$, etc. what do these refer to, in the algorithm?

**References**
1.  Distributed inverse constrained reinforcement learning for multi-agent systems, Liu & Zhu (2022)

**Updates**
Increased score from 5->6 (22 Nov)

**Details Of Ethics Concerns:**

_

---

> ### Author Response · Authors · 2023-11-17
> **Response to Reviewer NXcx**
>
> Thanks for your insightful reviews. We believe that our discussion can lead to a stronger paper. We address your comments as follows:
>
> **Weakness 1**: The lower level optimization (as explained in Equation 11, Appendix A.1) maximizes the reward and causal entropy objective while matching the constraint feature expectations. On the other hand, the authors of [B1] formulate the optimization as a maximization of the causal entropy while matching reward feature expectations and expected cumulative costs. The slight difference is that [B1] has an additional Lagrange multiplier term $\lambda$ in the dual formulation (see [B1], page 4, below remark 2, in the definition of $G$). In this work, since the constraint feature expectations are directly matched, the overall form of the lower level objective (Appendix, equation 12) is not exactly the same as [B1]. This is also apparent in the constrained soft Bellman policy definition adapted from [B1] (no $\lambda$ in the constrained policy for this work, whereas there is a $\lambda$ in the constrained policy as described in [B1], page 2, Appendix). The outcome is that the constraint function is just treated as a negative reward term that can be just added to the original reward and thus, constrained RL just amounts to running RL with reward $r-c$ (Appendix A.2 of this work also says this). I am not sure this is representative of typical constrained RL problems, since typically it is not possible to rewrite a constrained RL problem as an unconstrained RL problem with a different reward. (maybe the authors can elaborate on this).
>
> **Answer:** It is very insightful that you point this out. In fact, the definition of $G$ in Appendix A.1 is equivalent to the one in [B1] even if we do not have $\lambda$, and the definition of $G$ in Appendix A.1 can work for constrained RL in our case. Note that the cost function $c_{\omega}$ in our problem is not given nor fixed, but is actually learned. This point is very important for understanding why the slight difference of $\lambda$ does not matter and why the $r_{\theta}-c_{\omega}$ can work for constrained RL in our case.
>
> First, we would like to explain why the function $G$ in Appendix A.1 is equivalent to the function $G$ in [B1]. The definition of $G$ in Appendix A.1 is $G(\omega;\theta)=H(\pi_{\omega;\theta})+J_{r_{\theta}}(\pi_{\omega;\theta})+\omega^{\top}(\mu_c(\pi_i)-\mu_c(\pi_{\omega;\theta}))$ where $\pi_{\omega;\theta}(a|s)=\frac{\exp(r_{\theta}(s,a)-\omega^{\top}\phi_c(s,a)+\gamma\int_{s'\in\mathcal{S}}P(s'|s,a)V_{\omega;\theta}^{\text{soft}}(s')ds')}{\exp(V_{\omega;\theta}^{\text{soft}}(s))}$. The definition of function G in [B1] (let us call it $\bar{G}$) is $\bar{G}(\bar{\omega};\theta,\lambda)=H(\pi_{\bar{\omega};\theta,\lambda})+J_{r_{\theta}}(\pi_{\bar{\omega};\theta,\lambda})+\lambda\bar{\omega}^{\top}(\mu_c(\pi_i)-\mu_c(\pi_{\bar{\omega};\theta,\lambda}))$ where $\pi_{\bar{\omega};\theta,\lambda}(a|s)=\frac{\exp(r_{\theta}(s,a)-\lambda\bar{\omega}^{\top}\phi_c(s,a)+\gamma\int_{s'\in\mathcal{S}}P(s'|s,a)V_{\bar{\omega};\theta,\lambda}^{\text{soft}}(s')ds')}{\exp(V_{\bar{\omega};\theta,\lambda}^{\text{soft}}(s))}$ (section 8 in Appendix of [B1]). Note that $\lambda$ is just a scalar and $\bar{\omega}$ is a vector ([B1] studies linear cost). In [B1], both $\lambda$ and $\bar{\omega}$ are learned. In our problem, $\omega$ is learned. For any learned $\lambda$ and $\bar{\omega}$, we can always learn an $\omega$ such that $\omega=\lambda\bar{\omega}$. Therefore, $G$ in our case is equivalent to $\bar{G}$ in [B1] as long as we simply replace $\lambda\bar{\omega}$ in $\bar{G}$ with $\omega$. Since the characteristics of $\lambda\bar{\omega}$ can be easily captured by a single $\omega$, there is no need to learn two parameters $\lambda$ and $\bar{\omega}$.
>
> Second, we explain why the $r_{\theta}-c_{\omega}$ can work for the constrained RL in our case. We agree that it is not possible to rewrite a constrained RL problem as an unconstrained RL problem. However, we do not aim to use $r_{\theta}-c_{\omega}$ to solve a constrained RL problem where the reward function is $r_{\theta}$ and the cost function is $c_{\omega}$. What we do is to learn $r_{\theta}$ and $c_{\omega}$ such that the corresponding policy $\pi_{\omega;\theta}$ can imitate the expert and thus solve the original constrained RL problem where the reward function is $r_i$ and cost function is $c_i$. The learned cost function $c_{\omega}$ could be learned as the expert's cost function $c_i$ multiplying a very large positive constant so that the $r_{\theta}-c_{\omega}$ can enable $\pi_{\omega;\theta}$ to approximately satisfy the original constraint which is defined using $c_i$. The situation in [B1]

---

> > ### Author Response · Authors · 2023-11-17
> > **Response to Reviewer NXcx (continued)**
> >
> > is that they use $r_{\theta}-\lambda c_{\bar{\omega}}$ and they can learn both $\lambda$ and $\bar{\omega}$ to enable the policy to approximately satisfy the constraint in the original constrained RL. We have the same spirit with [B1] even if we do not have the parameter $\lambda$. Since $\lambda$ is just a scalar, as mentioned in last paragraph, we can learn a single model $c_{\omega}$ to replace $\lambda c_{\bar{\omega}}$. The $r_{\theta}-c_{\omega}$ in our case is equivalent to $r_{\theta}-\lambda c_{\bar{\omega}}$ in [B1].
> >
> > In conclusion, our approach is not mis-formulated compared to the one in [B1], but is actually equivalent to the one in [B1]. More importantly, our theoretical analysis is self-contained, i.e., we can theoretically guarantee the convergence and generalization performance of our approach.
> >
> > **Weakness 2**: What if several demonstrations are available for each task? If more demonstrations are available, ICRL could perform better and the gap between M-ICRL \& ICRL could be lesser. M-ICRL should still perform better, since it has a better meta-prior, but overall it would be useful to understand the empirical improvement of M-ICRL over ICRL as the number of demonstrations vary.
> >
> > **Answer**: Thanks for mentioning varying the number of demonstrations. We add an additional experiment in Appendix B.4 in the newly uploaded version to show how the performance of ICRL and M-ICRL improves when the number of demonstrations increases. Basically, ICRL improves faster than M-ICRL and the gap between ICRL and M-ICRL is decreasing. When there are more than $80$ demonstrations, the performance of ICRL and M-ICRL is very close because at this point, both M-ICRL and ICRL have enough information about the test tasks, and the meta-prior no longer plays a dominant role to help M-ICRL achieve good performance.
> >
> > **Q1**: While Equation 12 (Appendix A.1) is the dual of Equation 11, if the domain is extended from linear constraint functions to non-linear constraint functions, the equation would no longer behave as the dual of the original problem as formulated in Equation 11, right? Does it make sense to use this as the lower level problem, in that case?
> >
> > **Answer**: Yes, you are right! The function $G$ is no longer the dual function of (11) if we use non-linear cost functions. However, we can still use this $G$ function in the lower level.
> >
> > First, the fundamental reason that the function $G$ can be used to learn cost functions is that it serves as a negative log-likelihood function of the training data (explained in Appendix A.4) instead of being the dual of (11). When the cost function is non-linear, the function $G$ is no longer the dual function of (11) but it still serves as a negative log-likelihood function of the training data. Therefore, the function $G$ does not lose its functionality when the cost function is non-linear and thus can be used as the lower-level objective function. The derivation of $G$ from (11) only serves as explaining how we come up with the function $G$.
> >
> > Second, the whole theoretical analysis in this paper holds for non-linear cost functions, which provides strong justification that non-linear cost functions can work.
> >
> > **Suggestion**: More ablations (eg. empirical effect of $\alpha$, batch size $B$, number of gradient descent steps $K$, etc.) can be performed. Also, are these values specified somewhere in the paper?
> >
> > **Answer**: Thanks for your suggestion. The specification of the parameters is included in Table 2 in Appendix B.2 in the newly updated version. Moreover, we also include the effect of $K$ in Appendix B.4 in the newly uploaded version. Due to the time limit, we will keep adding more ablation results.
> >
> > **Suggestion**: Notation can be slightly confusing at some places, so I would suggest mentioning in the algorithms what the inputs and the outputs are, in words (eg. meta-priors, etc.). Implicit hyperparameters should also be mentioned in the beginning of the algorithm, eg. $\alpha$, $B$, etc. what do these refer to, in the algorithm?
> >
> > **Answer**: Thanks for your suggestion. We have revised the algorithms accordingly. The $\alpha$ is the step size and $B$ is the number of sampled tasks at each iteration $n$.
> >
> > **References**
> >
> > [B1] Shicheng Liu and Minghui Zhu. "Distributed inverse constrained reinforcement learning for multi-agent systems.“ In Advances in Neural Information Processing Systems, pp. 33444–33456, 2022.

---

> > > ### Comment · Reviewer_NXcx · 2023-11-22
> > > **Rebuttal response**
> > >
> > > Thanks for the clarifications. I have updated my score to 6.

---

### Official Review · Reviewer_5qhh · 2023-11-05

**Soundness:** 3 good
**Presentation:** 3 good
**Contribution:** 3 good
**Rating:** 6
**Confidence:** 3

**Summary:**

This paper extends the inverse constrained RL problem studied in (Liu & Zhu 2022) to the meta-learning setting where authors propose to first learn meta priors over reward/cost functions from similar tasks and then adapt the priors to new task via few-shot learning. This problem is then formulated using the bi-level optimization which is intractable in general. Authors propose novel approximate methods to solve the formulated bi-level optimization problem and quantify the approximation errors. Both physical and numerical experiments are conducted to demonstrate the effectiveness of the proposed method.

**Strengths:**

1. The author proposed a new setting which is based on the inverse constrained RL problem (Liu & Zhu 2022) and the meta RL/IRL (Xu et al 2019). It is a creative combination of existing ideas.

2. The paper is well-written with the main ideas, algorithms, theories and experiments well presented.

3. The demonstration using the physical experiment on drone navigation makes the proposed method more convincing.

**Weaknesses:**

See questions.

**Questions:**

1. In terms of novelty, it would be helpful for readers if authors can emphasize the unique novelty and challenges of solving meta inverse constrained RL beyond simply combining the techniques in the inverse constrained RL problem studied in (Liu & Zhu 2022) and the meta RL/ICRL studied in (Xu et al 2019; Rajeswaran et al 2019)?

2. There is a related work

"A CMDP-within-online framework for Meta-Safe Reinforcement Learning, Vanshaj Khattar, et al, ICLR, 2023"

which studies the meta learning for the constrained RL. What is the similarity and differences between authors' methodology compared with their works in terms of how to deal with constraints and meta-learning? It is important to add such comparisons in the paper.

3. Can authors further explain why the regularization term ||\eta - \omega||^2 in equation (3) is required for the M-ICRL problem?

4. The formulated M-ICRL is a complicated formulation, and the proposed approximate methods further increases the complexity of the algorithm. Can authors summarize the tricks in the implementation level to achieve the reported good results? This is helpful for the future researchers who want to extend this paper.

---

> ### Author Response · Authors · 2023-11-17
> **Response to Reviewer 5qhh**
>
> Thanks for your constructive reviews! We appreciate that you find this work creative and we believe that our discussion can contribute to a better paper in general. We address your comments as follows:
>
> **Q1**: In terms of novelty, it would be helpful for readers if authors can emphasize the unique novelty and challenges of solving meta inverse constrained RL beyond simply combining the techniques in the inverse constrained RL problem studied in [A1] and the meta RL/ICRL studied in [A2,A3]?
>
> **Answer**: Thanks for mentioning the novelty. Though our problem combines ICRL in [A1] and meta-learning in [A2,A3], our algorithm design and theoretic analysis are substantially different from those in the three references. We discuss the challenges of algorithm design in Section 3.1 and the challenges of theoretical analysis in the first paragraph of Section 4.1. Here we would like to further discuss our unique challenges and novelties in algorithm design and theoretical analysis compared to [A1,A2,A3].
>
> **Algorithm design**. [A1,A2,A3] and our work all formulate a bi-level optimization problem. Note that [A2] uses MAML [A7] and MAML can be regarded as a bi-level optimization problem where the lower level is one-step gradient descent from the meta-prior and the upper level aims to optimize for the meta-prior [A3]. A critical step of solving bi-level optimization problems is to compute the hyper-gradient, i.e., the gradient of the upper-level problem. The methods of computing the hyper-gradient in [A1,A2,A3] **CANNOT** be applied to our case and thus novel algorithm designs for computing the hyper-gradient are needed. In specific, there are two unique challenges in terms of algorithm design.
>
> First, our hyper-gradient includes the term $\frac{\partial^2}{\partial\theta^2}L_i(\theta,\eta_i^{\ast}(\theta,\omega))$ while [A1,A2,A3] do not have this term. As mentioned in challenge (i) in Section 3.1, computing $\frac{\partial^2}{\partial\theta^2}L_i(\theta,\eta_i^{\ast}(\theta,\omega))$ requires to compute the gradient of an inverse-of-Hessian term $[\nabla_{\eta\eta}^2G_i(\eta_i^{\ast}(\varphi_i,\omega);\varphi_i)+\lambda I]^{-1}$. Since we use neural networks as the parameterized model, computing this inverse-of-Hessian term is already challenging enough and now we need to compute the gradient of this inverse-of-Hessian term, which is intractable. To solve this issue, we propose a novel approximation design where we only need to compute first-order gradients (see (6)-(7) and Algorithm 2) and we can prove that the approximation error can be arbitrarily small (Lemma 2).
>
> Second, as mentioned in challenge (ii) in Section 3.1, our hyper-gradient includes the inverse-of-Hessian term $[\nabla_{\eta\eta}^2G_i(\eta_i^{\ast}(\varphi_i,\omega);\varphi_i)+\lambda I]^{-1}$ while [A1,A2] do not have this issue. [A3] also has this issue, however, they just use existing conjugate gradient method and assume that a good approximation can be obtained. In contrast, we propose to solve an optimization problem (10) in Algorithm 3 to approximate and more importantly, we theoretically guarantee in Lemma 1 that the approximation error can be arbitrarily small.
>
> **Theoretical analysis**. We study **BOTH** the convergence and generalization of our algorithm while [A2] do not have theoretical analysis and [A1,A3] only study convergence. Therefore, the whole Section 4.2 is unique and novel compared to [A1,A2,A3] because it studies generalization. Even for convergence, our analysis is novel compared to [A1,A3]. In specific, since we have the unique challenges for algorithm design and we provide novel algorithm designs (i.e., approximation methods) to tackle those unique challenges, we need corresponding theoretical analysis to guarantee the performance of our unique algorithm designs, i.e., Lemma 1 and Lemma 2 quantifies the approximation error of our algorithm designs.

---

> > ### Author Response · Authors · 2023-11-17
> > **Response to Reviewer 5qhh (continued)**
> >
> > **Q2**: There is a related work [A4] which studies the meta learning for the constrained RL. What is the similarity and differences between authors' methodology compared with their works in terms of how to deal with constraints and meta-learning? It is important to add such comparisons in the paper.
> >
> > **Answer**: Thanks for mentioning this valuable reference, we have added the comparisons in Appendix A.15 in our newly updated version. First, we would like to mention that [A4] studies a different problem. [A4] studies meta constrained RL where the constraint (cost function) is given while we study meta ICRL where the cost function needs to be learned.
> >
> > For dealing with the constraint, [A4] has a constrained RL problem to solve for each task. It uses a constrained RL algorithm called CRPO [A5] to solve the constrained RL problem and get a corresponding task-specific policy. We have a cost learning problem in the lower level and we use gradient descent to obtain a corresponding task-specific cost adaptation. One similarity between [A4] and our work is that we both do not require the exact task-specific adaptation and this makes the theoretical analysis of the meta learning performance more challenging.
> >
> > For dealing with meta-learning, [A4] studies an online setting where at each online iteration, a new task is input and a corresponding task-specific policy adaptation is computed. At each online iteration, it updates the policy meta-prior by minimizing the KL divergence between the policy meta-prior and the current task-specific policy adaptation via one or multiple online gradient descent steps. In contrast, we utilize a bi-level optimization framework where we learn the meta-priors in the upper level such that the corresponding task-specific adaptations can maximize the likelihood of the demonstrations of each task. In order to optimize for the meta-priors, we need to compute the hyper-gradient which is very challenging in our case. We propose several novel approximation methods and algorithm designs to approximate the hyper-gradient. In conclusion, the meta-prior in our case is learned such that the task-specific adaptations adapted from the meta-prior have good performance on each specific task while the meta-prior in [A4] is learned such that the meta-prior is close to the task-specific adaptations according to the metric of KL divergence.
> >
> > **Q3**: Can authors further explain why the regularization term $||\eta - \omega||^2$ in equation (3) is required for the M-ICRL problem?
> >
> > **Answer**: It is very insightful that you mention the regularization term $||\eta-\omega||^2$. Note that $\omega$ is the cost meta-prior and $\eta$ is to compute the task-specific cost adaptation $\eta_i^{\ast}$. This regularization term characterizes how we adapt the cost meta-prior $\omega$ to the task-specific cost adaptation $\eta_i^{\ast}$. Basically, we not only want the task-specific cost adaptation $\eta_i^{\ast}$ to minimize $G_i(\eta;\varphi_i)$, but also want the task-specific cost adaptation $\eta_i^{\ast}$ not to be too far from the cost meta-prior $\omega$. Since the data of training the task-specific cost adaptation $\eta_i^{\ast}$ is lacking, this regularization term can avoid overfitting. In fact, this regularization term is first introduced in iMAML [A3] and this kind of meta-learning method is called meta-regularization [A6].

---

> > > ### Author Response · Authors · 2023-11-17
> > > **Response to Reviewer 5qhh (continued)**
> > >
> > > **Q4**: The formulated M-ICRL is a complicated formulation, and the proposed approximate methods further increases the complexity of the algorithm. Can authors summarize the tricks in the implementation level to achieve the reported good results? This is helpful for the future researchers who want to extend this paper.
> > >
> > > **Answer**: Thanks for mentioning the implementation tricks and we have added the details in Appendix B.3 in the newly uploaded version. We have two major tricks, the first trick is to reduce the sample complexity of computing all the gradient and second-order terms in our algorithm, and the second trick is to reduce the iteration complexity of solving the lower-level problem and computing the partial gradients.
> > >
> > > **Reducing the sample complexity**. Note that the expressions of all the gradient and second-order terms are included in Appendix A.3. These terms all have expectation terms which need to be estimated. Take $\nabla_{\omega}G_i(\omega;\theta)=\nabla_{\omega}J_{c_{\omega}}(\pi_i)-E_{\zeta\sim P_{\pi_{\omega;\theta}}}[\sum_{t=0}^{\infty}\gamma^t\nabla_{\omega}c_{\omega}(s_t,a_t)]$ as an example, we need to roll out $\pi_{\omega;\theta}$ to get a set of trajectories in order to estimate $E_{\zeta\sim P_{\pi_{\omega;\theta}}}[\sum_{t=0}^{\infty}\gamma^t\nabla_{\omega}c_{\omega}(s_t,a_t)]$. The trick of reducing sample complexity is that for each task, we use the same sampled trajectory set to estimate the expectation terms in all gradient and second-order terms, e.g., $\nabla_{\omega}L$, $\nabla_{\omega\omega}^2G_i$, etc. By doing so, we only need to conduct the sample operation once when a new task is input, and then we just use these samples to estimate the expectation terms of all the gradient and second-order terms within this task.
> > >
> > > **Reducing the iteration complexity**. The trick we use is warm start. For solving the lower level problem, we do not initialize $\eta$ randomly. In fact, we use the obtained task-specific adaptation of last task to be the initialization of this task. Since the tasks are relevant in our case, The corresponding task-specific adaptations are not too far from each other. Therefore, this kind of initialization can significantly reduce the iteration numbers needed to compute the task-specific adaptations. We use the same trick for Algorithm 3 and it can significantly reduce the needed number of iterations, especially for computing the partial gradients of the perturbed variables in Algorithm 2. Since $\theta+\delta\Delta_{\theta,i}$ and $\theta-\delta\Delta_{\theta,i}$ are very close given that $\delta$ is chosen small, empirically, fewer than ten gradient descent steps from the warm start initialization can provide a good result.
> > >
> > > **References**
> > >
> > > [A1] Shicheng Liu and Minghui Zhu. "Distributed inverse constrained reinforcement learning for multi-agent systems.“ In Advances in Neural Information Processing Systems, pp. 33444–33456, 2022.
> > >
> > > [A2] Kelvin Xu, Ellis Ratner, Anca Dragan, Sergey Levine, and Chelsea Finn. "Learning a prior over intent via meta-inverse reinforcement learning.” In International Conference on Machine Learning, pp. 6952–6962, 2019.
> > >
> > > [A3] Aravind Rajeswaran, Chelsea Finn, Sham M Kakade, and Sergey Levine. "Meta-learning with implicit gradients." In Advances in Neural Information Processing Systems, pp. 113–124, 2019.
> > >
> > > [A4] Khattar, Vanshaj, Yuhao Ding, Bilgehan Sel, Javad Lavaei, and Ming Jin. "A CMDP-within-online framework for Meta-Safe Reinforcement Learning." In International Conference on Learning Representations, 2022.
> > >
> > > [A5] Tengyu Xu, Yingbin Liang, and Guanghui Lan. "CRPO: A new approach for safe reinforcement learning with convergence guarantee." In International Conference on Machine Learning, pp. 11480-11491, 2021.
> > >
> > > [A6] Maria-Florina Balcan, Mikhail Khodak, and Ameet Talwalkar. "Provable guarantees for gradient-based meta-learning." In International Conference on Machine Learning, pp. 424-433, 2019.
> > >
> > > [A7] Chelsea Finn, Pieter Abbeel, and Sergey Levine. "Model-agnostic meta-learning for fast adaptation of deep networks." In International Conference on Machine Learning, pp. 1126–1135, 2017.

---

### Author Response · Authors · 2023-11-17
**Looking forward to discussions**

Dear Reviewers,

Thanks for your constructive reviews. We have addressed all your comments, while some of them are misunderstandings. We also revise our paper according to your suggestions, and upload the revised version (revisions highlighted in blue).

We are really looking forward to discussions.

Best,

Authors.

---

### Meta-Review · Area_Chair_qqdM · 2023-12-09

**Metareview:**

## Overview
The paper addresses a significant extension of inverse constrained reinforcement learning (ICRL) into the meta-learning domain. It proposes an algorithm to learn meta-priors over reward functions and constraints from related tasks and adapt them to new tasks using few expert demonstrations. The problem is formulated as a bi-level optimization, and the authors introduce novel approximation methods to address the computational challenges involved. Theoretical analysis of convergence and generalization is provided, backed by empirical validation in both physical (drone navigation) and numerical experiments.

## Strengths
1. **Theoretical Contribution**: The paper extends ICRL to the challenging domain of meta-learning. The theoretical groundwork for convergence guarantees and generalization analysis is a significant addition to the literature.
2. **Novel Algorithmic Approach**: The proposal of a bi-level optimization framework and novel approximation methods to compute hyper-gradients showcase a deep understanding of the complexities in meta ICRL.
3. **Empirical Validation**: The physical experiment with drone navigation adds substantial credibility to the proposed methods, demonstrating their practical applicability and effectiveness.

## Weaknesses
1. **Marginal Advancement over ICRL**: The advancements focus more on computational difficulties rather than directly advancing the core ICRL methodology.
2. **Audience and Relevance**: The paper's emphasis on computational and theoretical aspects might not align fully with the broader ML community's interests in practical applications of meta-learning in RL.

## Reviewer Discussions
The discussions among reviewers and authors highlight the paper’s strong theoretical foundation and novel approach. Concerns about novelty and complexity are acknowledged, but the overall sentiment leans towards the paper's significance in advancing theoretical understanding of meta-learning in the ICRL domain.

## Decision Recommendation
Given the substantial theoretical contributions, novel algorithmic approach, and successful empirical validation, the paper is recommended for acceptance. The paper represents a significant step forward in the theoretical understanding of meta-learning in the ICRL domain. Authors are encouraged to consider feedback regarding clarity and application in future work.

**Justification For Why Not Higher Score:**

- The paper, while making significant theoretical contributions, presents incremental advancements over existing frameworks in Meta IRL and ICRL.
- The complexity of the proposed methods and their theoretical underpinnings might limit practical applicability and accessibility.
- The assumptions made about function smoothness and boundedness may not align with real-world scenarios, restricting broader applicability.
- Clarity in the presentation of methodologies and theories needs improvement for wider comprehension and application.

**Justification For Why Not Lower Score:**

The paper has sufficient contribution to be accepted.

---

### Decision · Program_Chairs · 2024-01-16

Accept (poster)